# Can we trust remote sensing ET products over Africa?

Imeshi Weerasinghe[1], Wim Bastiaanssen[2], Marloes Mul[2], Li Jia[3,4], and Ann van Griensven[1,2]

[1]Vrije Universiteit Brussels, Brussels, Belgium
[2]IHE Delft Institute for Water Education, Delft, Netherlands
[3]Chinese Academy of Sciences, Beijing, China
[4]Joint Center for Global Change Studies, Beijing, China

**Correspondence:** Imeshi WEERASINGHE (Imeshi.Nadishka.Weerasinghe@vub.be)

**Abstract.** Evapotranspiration (ET) is one of the most important components in the water cycle. However, there are relatively few direct measurements of available ET (e.g. using flux towers). Nevertheless, various disciplines ranging from hydrology to agricultural and climate sciences require information on the spatial and temporal distribution of ET at regional and global scales. Due to the limited data availability, attention has turned toward satellite based products to fill observational gaps. Various remote sensing (RS) and other data products have been developed, providing a large range of ET estimations. Across Africa only a limited number of flux towers are available which are insufficient for systematic evaluation of RS derived and other available ET products. Thus, in this study, we conduct a methodological evaluation of nine existing RS derived and other ET products in order to evaluate their reliability at the basin scale. A general water balance (WB) approach is used, where ET is equal to precipitation minus discharge for long-term averages. Firstly, ET products are compared with WB inferred ET ($ET_{WB}$) for basins without long-term trends present. The ET products and calculated $ET_{WB}$ are then evaluated against the Budyko equation, used as a reference condition. The spatial characteristics of the ET products are finally assessed through the analysis of selected land cover elements, forests, irrigated areas and water bodies, across Africa. Additionally, a cluster analysis is conducted to identify similarities between individual ET products. The results show that CMRSET, SSEBop and WaPOR rank highest in terms of estimation of long-term average mean ET across basins with low biases and good spatial variability across Africa. GLEAM consistently ranks the lowest in most evaluation criteria however, has the longest available time period. Each product shows specific advantages and disadvantages. Depending on the study under question at least one product should be suitable for a particular requirement. Care should be taken to bear in mind that many products suffer from a large bias. Based on the evaluation criteria in this study the three highest ranked products, CMRSET, SSEBop and WaPOR would suit many user needs due to low biases and good spatial variability across Africa.

# 1 Introduction

Evapotranspiration (ET) or the water vapor flux is an important component in the water cycle and is widely studied due to its implications in hydrology to agricultural and climate sciences (Trambauer et al., 2014). Growing attention has been given to estimating ET fluxes at regional and global scales for a wide variety of reasons, for example, understanding the partitioning of energy and water at the earths surface and their feedbacks; how the different external drivers of ET vary regionally and; understanding the impacts of potential changes on the hydrological cycle under a changing climate, to name a few (Teuling et al., 2009; Vinukollu et al., 2011a; Mu et al., 2011). However, the estimation of ET at large scales has always been a difficult task due to direct measurement of ET being possible only at point locations, for example using flux towers (Trambauer et al., 2014). Obtaining ET observations from flux towers is challenging due to the high costs of implementation and maintenance and often studies rely on openly accessible data especially for regions in Asia, South America and Africa. Worldwide flux tower data can be openly accessed through FLUXNET [1], however there is limited coverage in many regions (Figure 1 (left panel)). For the entire African continent, for example, there are only six FLUXNET sites (Figure 1 (right panel)) with available ET data. Due to the limited data availability of in situ measurements a method of evaluating ET estimations using data other than point observations is required.

Recent advances in satellite based ET products provide promising data to fill these observational gaps (Alkema et al., 2011; Miralles et al., 2016; Guerschman et al., 2009; Zheng et al., 2016; Mu et al., 2007, 2011; Jung et al., 2011; Senay et al., 2013). ET cannot be directly measured by satellite based measurements, but can be derived from physical variables that can be observed from space, such as latent heat flux and surface heat flux using the surface energy balance. In addition, due to passing frequencies and cloud interference, interpolations in time are required. Keeping this in mind, remote sensing derived ET cannot be interpreted as direct satellite observations but as model outputs based on satellite forcing data (Miralles et al., 2016). Therefore, large-scale estimations of ET are most commonly products of remote sensing based models, hydrological models and land-surface models (Trambauer et al., 2014). More recently, ET products have also been developed using Machine Learning (ML) approaches such as Model Tree Ensemble (MTE) or Artificial Neural Networks (ANN) combined with observed flux tower data or model outputs used as training sets (Tramontana et al., 2016; Jiménez et al., 2011; Jung et al., 2017; Alemohammad et al., 2017).

Satellite observations often give useful information on the spatial variability, however many products tend to suffer from a large bias. With this range of approaches to estimate ET, large differences are observed among the products and therefore, evaluation is required. Keeping in mind limited availability of in situ measurements for evaluation, an alternate approach is to consider the water balance closure at the river basin scale. Only few studies exist comparing different satelite based and gridded ET products at the global and continental scales using this approach among others. In their study, Miralles et al. (2016) evaluated four commonly used and tested algorithms (the Surface Energy Balance System (SEBS): (Su, 2002), the Moderate Resolution Imaging Spectroradiometer (MOD16): (Mu et al., 2007, 2011), the Global Land Evaporation Amsterdam Model

---

[1]FLUXNET is a global network of micrometeorological flux measurement sites that measure the exchange of CO2, water vapor and energy between the biosphere and the atmosphere (Baldocchi et al., 2001)

(GLEAM): (Miralles et al., 2011) and the Priestly-Taylor Jet Propulsion Laboratory model (PT-JPL): (Fisher et al., 2008)) to derive ET using a range of methods including water balance closure across a broad range of catchments worldwide. They found that GLEAM and PT-JPL appear more realistic when compared with 837 globally distributed catchments, however find that all products show large dissimilarities in conditions of water stress and drought conditions (Miralles et al., 2016). Another

global evaluation of three process-based models (SEBS, Penman-Montieth algorithm (PM-Mu): (Mu et al., 2007; Penman, 1948; Montieth, 1965) and Priestly-Taylor based approach (PT-Fi): (Priestley and Taylor, 1972; Fisher et al., 2008)) in their estimation of ET was conducted by Vinukollu et al. (2011a) using the water balance approach at twenty six major basins worldwide along with other methods. A Root Mean Square Difference (RMSD) of 118 to 194 mm/year and bias of -132 to 53 mm/year were found between the estimated annual ET and water balance approximations. The LandFlux initiative, supported

by GEWEX (http://www.gewex.org/) is a framework aiming to evaluate and compare several global ET data sets (Mueller et al., 2011; Jiménez et al., 2011). With these aims, global merged bench-marking ET products were derived (Mueller et al., 2013a) using 40 datasets over a seven year period (1989-1995) and 14 datasets over a seventeen year period (1989-2005) to be used for evaluation. At the continental scale a study by Trambauer et al. (2014) compared ET estimates derived using a continental hydrological model (PCR-GLOBWB: (Van Beek and Bierkens, 2009) with other independently computed ET products (the

European Center for Medium-range Weather Forecasts (ECMWF) Re-Analysis (ERA)-Interim: (Dee et al., 2011), ERA-Land: (Balsamo et al., 2015), MOD16, GLEAM and three other versions of the PCR-GLOBWB model) using visual inspection and statistical methods. By sub-diving the continent into climatic regions, they found that the annual anomalies of ET for each of the products with respect to the multi-product mean was highest in ERA-Interim. GLEAM was in most cases lower than the multi-product mean while PCR-GLOBWB was close to the multi-product mean in nearly all cases.

To our knowledge, there are no existing studies focusing solely and entirely on the African continent that use the water balance approach for evaluating existing ET products. The water budget of a catchment implies that precipitation (P) minus river discharge (Q) equals evapotranspiration ($ET_{WB}$) when considering a long time period so that the change in water storage (soil moisture, lakes, deltas) can be neglected (Miralles et al., 2016, 2011; Vinukollu et al., 2011b). Using this general water balance to infer $ET_{WB}$, it is possible to gain understanding of the magnitude of ET within a given basin and hence to estimate

biases in ET estimation by the different products at the catchment scale. Unfortunately, the period of observation for measured discharge for certain basins is limited or do not overlap with existing ET products and thus different time periods need to be used.

    Therefore, this study focuses on evaluating nine existing, mostly open access, ET products ($ET_{RS}$) using a water balance approach over Africa. The products being analysed are CSIRO's Moderate resolution imaging spectroradiometer Reflectance Scaling Evapotranspiration (CMRSET): (Guerschman et al., 2009), ETMonitor: (Zheng et al., 2016), GLEAM, LandFlux-

30 EVAL, MOD16, FLUXNET Model Tree Ensemble (MTE): (Jung et al., 2011), the operational Simplified Surface Energy Balance model (SSEBop): (Senay et al., 2013), the Food and Agriculture Organisation's (FAO) portal to monitor **Wa**ter **P**roductivity through **O**pen access of **R**emotely sensed derived data (WaPOR): (FAO, 2018) and the Water, Energy and Carbon Cycle with Artificial Neural Networks (WECANN): (Alemohammad et al., 2017). The evaluation of the products will be con-

35 ducted using a) a comparison of their performance against calculated $ET_{WB}$, b) a robustness check of their performance against

the Budyko curve (Budyko, 1974) which provides a reference condition for the water balance assuming it correctly partitions P into Q and c) a spatial variability assessment using specific land cover elements (forests, water bodies and irrigated areas).

## 2 Data and Methods

### 2.1 Data

#### 2.1.1 Evapotranspiration products

The derived ET products being evaluated in this study include CMRSET, ETMonitor, GLEAM, LandFlux-EVAL, MOD16, MTE, SSEBop, WaPOR and WECANN. Overall there are large differences between the products which results in certain advantages and disadvantages between products. All products have a global spatial coverage (advantage) except for WaPOR (disadvantage). All products are openly accessible (advantage) except for ETMonitor (disadvantage). GLEAM and ETMonitor have a daily, CMRSET has an 8-daily and WaPOR has dekadal temporal resolution (advantage) over other products which have monthly or yearly resolutions (disadvantage). Most products are still ongoing (advantage) except for ETMonitor, LandFlux-EVAL and MTE (disadvantage). GLEAM, MTE and LandFlux-EVAL have data available prior to 1990 (advantage) with all other product data available after 1999 (disadvantage). CMRSET and WaPOR have the highest resolutions ((0.0022 $°\times$ 0.0022°)) (possible advantage), LandFlux-EVAL and WECANN have the lowest resolutions (1 $°\times1°$) (possible disadvantage) with all other products ranging in between. Table 1 summarises the different features mentioned and whether these are possible advantages or disadvantages. These different ET products give a good sample of the available data sets to choose from.

All products have been projected and gridded on a 0.0022 $°\times$ 0.0022° geographic grid and averaged at yearly temporal resolution for the purposes of this study. Table 2 summarizes the characteristics of the products being used. For details and access on each of the products please refer to the references and access section in Table 2.

#### 2.1.2 Precipitation products

The precipitation products used in this study are the EartH2Observe (E2OBS), WATCH forcing data methodology applied to ERA-Interim Reanalysis (WFDEI), ERA-Interim data Merged and Bias-corrected (EWEMBI), the Climate Hazards group Infrared Precipitation with Stations (CHIRPS) and the Multi-Source Weighted Ensemble Precipitation (MSWEP). Precipitation products were averaged at yearly temporal resolution for the purposes of this study. Table 3 summarizes the characteristics of the products being used. For details and access on each of the products please refer to the references and access section in Table 3. The ensemble of the three P products were used for all calculations requiring P.

#### 2.1.3 Discharge data

Discharge data was obtained from the Global Runoff Data Centre (GRDC) and the Vrije Universiteit Brussels (VUB) Department of Hydrology and Hydraulic Engineering (HYDR). Table 4 summarizes the characteristics of the data being used. For details and access on each of the products please refer to the references and access section in Table 4.

### 2.1.4 Reference potential evapotranspiration data

Three global reference Potential Evapotranspiration (PET) data products developed by Deltares (Sperna Weiland et al., 2015) are used based on the Hargreaves (Har) (Hargreaves and Samani, 1985), Penman-Montieth (P-M) (Montieth, 1965; Penman, 1948) and Priestly-Taylor (P-T) (Priestley and Taylor, 1972) approaches. Table 5 summarizes the characteristics of the products being used. For details and access on each of the products please refer to the references and access section in Table 5. The ensemble of the three PET products were used for all calcuations requiring PET.

### 2.2 Methods

A methodology to evaluate ET product estimations is presented next:

1. Comparison between catchment water balance evapotranspiration ($ET_{WB}$) and ET products

2. Evaluation of $ET_{WB}$ and ET product estimations using the Budyko curve ($ET_{Budyko}$) as a reference

3. Assessment of spatial variability using land cover elements

4. Assessment of similarity using a cluster analysis

### 2.2.1 Catchment water balance evapotranspiration ($ET_{WB}$)

Due to the limited availability of direct observations of ET across Africa, we infer ET estimates at the river basin level using the water balance approach assuming a negligible change in storage (discussed further in Section 5) for long time periods:

$$ET_{WB} = P - Q \tag{1}$$

$ET_{WB}$ was calculated for 27 major river basins across Africa based on discharge data (GRDC and HYDR VUB) quality and availability at the outlets of 54 major basins (Fig. 2). Catchment or basin areas were taken from the 'Major River Basins of the World' (MRBW) shapefile (World Bank, 2017). Discharge was converted from cubic meters per second to millimeters per year using the above mentioned catchment areas for all years of data availability for each basin. Since direct observations of precipitation from gauges were not used, precipitation was taken as the average of the three data products, EWEMBI, CHIRPS and MSWEP. Basin average precipitation was calculated for the years 1979-2016 according to the MRBW shapefile boundaries recording the basin mean. The performance of the precipitation products in estimating P for each of the basins were compared. Long-term $ET_{WB}$ was calculated by using the long-term average discharge and precipitation data for each catchment. The MRBW shapefile area did not differ greatly with the drainage area reported by the GRDC except in two cases. Here we found the $ET_{WB}$ calculated using the two areas only differed by 2.5 percent and 3.3 percent and thus kept these basins in the analyses.

One problem that arises when using the water balance approach is that the period of observation for measured discharge is limited or does not overlap with existing ET products in certain cases. For this reason, long-term averages of $ET_{WB}$ were used where no major trends were present in order to justify the evaluation using different time periods (discussed further in

Section 5). The Mann-Kendall (MK) (Mann, 1945; Kendall, 1948) test was used to identify whether a monotonic upward or downward trend is present in the calculated $ET_{WB}$ estimates. The MK test is non-parametric (distribution free) and best used as an exploratory analysis to identify where changes are significant or of large magnitude (Matzke et al., 2014) and should only be used where seasonal trends are not present. Considering annual averages are used in this study, the MK test was deemed appropriate.

In order to conduct our comparisons using the calculated $ET_{WB}$, all ET products being evaluated were projected to WGS 84, EPSG:4326 on a $0.0022° \times 0.0022°$ grid. This resolution represented the highest spatial resolution of the products being analysed. Products were resampled to the highest resolution in order to obtain the best approximation of basin areas when overlaid with basin boundary shapefiles. Only negligible differences were found between calculation of $ET_{WB}$ using products with original resolution compared with $ET_{WB}$ calculated using resampled products. The nearest neighbours' interpolation method was used for any resampling required from course to high resolution to limit the loss of any information. The estimations were then combined to give a single map for each product of the long-term average $ET_{RS}$ across Africa. The time periods averaged for each product can be found in Table 2. Basin average $ET_{RS}$ was calculated according to the MRBW shapefile boundaries and the basin mean was recorded. The Root Mean Square Error (RMSE), the basin area weighted RMSE ($RMSE_{aw}$), the correlation coefficient (r), bias and basin area weighted bias ($bias_{aw}$) between $ET_{WB}$ versus $ET_{RS}$ for all basins were calculated. Basin area weighting was considered when calculating bias and RMSE due to a large difference in basin areas. Therefore, basins with larger areas had more weight in the basin area weighted statistics than basins with smaller areas. Correlations were calculated based on long-term averages across all basins.

The ranking of the ET products are based on their performance on RMSE, $RMSE_{aw}$, r, bias and $bias_{aw}$.

## 2.2.2 Evaluation using the Budyko curve

The Budyko equation partitions precipitation into streamflow and $ET_{Budyko}$ by describing the relationship between mean annual ET and the long-term average water and energy balance at catchment scales (Sposito, 2017) as seen in Fig. 3. Budyko (1974) developed this approach for the physics of catchment ET by postulating on the phase transformation of green water to vapor and thus that ET reflects not only the partitioning of water but also radiant energy at the vadoze zone and atmosphere interface (Sposito, 2017; Gerrits et al., 2009) following equation 2.

$$\Big[\frac{PET}{P}tanh(\frac{1}{\frac{PET}{P}})(1-exp^{-\frac{PET}{P}})\Big]^{0.5} \tag{2}$$

The Budyko curve provides a reference condition for the water balance assuming it correctly describes the partitioning of P into Q, which can be used to see how well the ET products and calculated $ET_{WB}$ perform in estimating ET. For each of the basins under study, we calculated ET/P and PET/P and plotted these against the Budyko curve. Average PET estimates from the three products using the Hargreaves, P-M and P-T approaches were used by taking the basin mean PET according to the MRBW shapefile boundaries. The performance of the reference potential evapotranspiration products in estimating PET for each of the basins were compared. P was taken as the average of EWEMBI, CHIRPS and MSWEP precipitation products. The bias was found between the calculated $ET_{WB}$ and $ET_{RS}$ with the calculated $ET_{Budyko}$.

The ranking of the ET$_{RS}$ from each product are based on the performance of their average bias across all basins with that of the calculated ET$_{Budyko}$.

### 2.2.3   Spatial variability assessment

Three types of land cover elements were evaluated in this study, irrigated areas, water bodies and forested areas. A map with Areas Equipped for Irrigation actually irrigated (AEIai) by FAO and Rheinische Friedrich-Wilhelms-University (Siebert et al., 2013), a map of Water Bodies obtained from the Global Reservoir and Dam ($WB_{GRanD}$) database (Lehner et al., 2011) and a map of 2013 Intact Forest Landscapes (IFL) were used to evaluate how well the ET products identified spatial characteristics. Two steps were used. Firstly the ET products were evaluated visually. Using different scales and identified land cover elements (Figure 4) the ET products were evaluated on how well each type of land cover element was detected. Secondly, a quantitative assessment was conducted for forested areas and water bodies. A quantitative assessment of irrigated areas was not conducted due to not being able to find a suitable reference condition for such large pixels and long-term temporal scales. For water bodies ET should be more or less equal to the PET. Therefore, the long-term annual average ET$_{RS}$ and PET across water bodies was calculated by recording the mean according to the boundary provided by the $WB_{GRanD}$ map. The mean ET$_{RS}$ for water bodies for each ET product was then compared with the PET mean for water bodies by calculating the bias.

For forested areas, the average ET was taken from literature where estimations for the Congo forest, the forested area being evaluated, were between 1200-1500 mm/year (Otto et al., 2013; Reynolds et al., 1988). Therefore a value of 1350 mm/year as a reference for ET across the evaluated forested area was taken. Mean values of ET for the forested area were found using the IFL shapefile and recorded for each ET product. The bias between the reference ET as reported in literature and calculated mean ET for forested areas for each product was found and recorded.

Ranking was conducted in two stages. Firstly on the performance of ET products to characterise the three land cover element types through visual inspection. And secondly based on the bias of each of the ET products in relation to the used reference for water bodies and forested area.

### 2.2.4   Assessment of similarity

Lastly, a cluster analysis was performed, using the method followed by Wartenburger et al. (2018) on the ET products to find the overall level of similarity between the individual products in terms of spatial variability and magnitude. The aggregated long-term average maps for all products were used whereby the pairwise Euclidean distance between each data set for each pixel was calculated and evaluated. Each of the maps used were resampled to $0.0096° \times 0.0096°$ for computation efficiency.

## 3   Results

### 3.1   Catchment water balance

#### 3.1.1   Comparison of precipitation and potential evapotranspiration products

Precipitation and PET were taken as the average of three products. Here we compare the results of the different P and PET
products for the basins being analysed. We see that the three precipitation products show little differences in their estimations
of long-term average P across the basins. No large outliers can be seen (Figure 5). The comparison of the three PET products
showed larger differences in their estimations of long-term average PET across the basins (Figure 6). One significant outlier
can be seen for Bandama basin where the Hargreaves PET product has a much lower PET estimation than the Priestly-Taylor
product. However, as no reference PET was available for Banadama or any of the other basins we kept all basins within the
analyses and still used the average of all three products.

#### 3.1.2   Basins used in the analyses

Figure 7 (left) shows the long-term average $ET_{WB}$ estimates for the 27 basins with available discharge and precipitation data.
The spread of the ET across the basins seems to be consistent with the African climate, where basins in the semi-arid to arid
northern and southern parts of Africa show lower ET than the more centrally located basins known to be more tropical.

The MK test was then conducted on the 27 basins with calculated $ET_{WB}$ to test for trends. In order for the MK test to be
accurate a minimum of ten data points should be used which were not available for all basins. For these basins the MK test was
conducted on the collected P and Q data used to calculate ET. For the results from the MK test please see Table 6 in Appendix
A. After conducting the MK test on the 27 basins for major trends in the calculated $ET_{WB}$ and/or the precipitation and discharge
data, 20 basins remained without a monotonic trend being present (Fig. 7). The spread of the remaining 20 basins still gives
good spatial coverage for analysis across the African continent.

#### 3.1.3   Catchment water balance comparison

Table 7 shows the calculated statistics for the comparison of the long-term average $ET_{WB}$ versus $ET_{RS}$ across the average of all
basins. Three products, CMRSET, SSEBop and WaPOR, clearly stand out in terms of showing low biases ranging from 3-46
mm/year. The remainder of the products have relatively large biases ranging from 115-313 mm/year. CMRSET and WaPOR
are the only two products that overestimate ET with respect to calculated $ET_{WB}$ while all other products underestimate ET
when looking at the average bias across all basins. All products show a high RMSE, with CMRSET, SSEBop and WaPOR
showing the lowest RMSE and $RMSE_{aw}$. The $RMSE_{aw}$ for most products exceeds 300 mm/year. There is a significant positive
correlation for all products ranging from 0.89-0.97 with GLEAM and LandFlux-EVAL showing the strongest relationships
with $ET_{WB}$ across the different basins.

Delving deeper into the biases (Fig. 8) we can identify certain basins where most products have large biases, namely Awash,
Groot, Niger, Olifant and the Upper Blue Nile. The only pattern that may be seen here with the location of the basins is that they

are found in the semi-arid northern and southern regions of Africa. The majority of the products underestimate basin-average ET across most basins except for CMRSET and WaPOR where ET is mostly overestimated. While the ET is equally over and underestimated by SSEBop across the different basins.

## 3.2 Evaluation using the Budyko curve

Figure 9 shows the ability of each ET product to capture ET according to the Budyko curve. The $ET_{WB}$ follows the Budyko curve well, where we see that for each of the basins, the calculated $ET_{WB}$ falls very close to the Budyko curve. The calculated ET for most of the ET products and also for the majority of basins falls under the curve showing a tendency for products to underestimate basin ET as previously observed. Conversely, a clear tendency by the CMRSET product of overestimating basin ET can be seen. What is interesting to note here is that some ET products exceed either the water limit and/or the energy limit in their calculation of ET in certain basins. This implies water is being lost, for example through the groundwater system when the energy limit is exceeded or there is an additional input of water beyond precipitation if the water limit is exceeded. SSEBop, WECANN and CMRSET exceed the water limit in more basins relative to other products, however their ET estimations are not necessarily further from ET estimations using the Budyko approach as given by equation 2. This is confirmed in table 8 where CMRSET and SSEBop along with WaPOR have the lowest biases when compared with $ET_{Budyko}$ after $ET_{WB}$.

## 3.3 Spatial variability assessment

Figure 10 shows ET across Africa for all ET products with the specific land cover elements (forest, irrigated areas and water bodies) highlighted. Two different scales are used in order to be able to visually compare the products according to spatial variability rather than magnitude of ET. For products where large biases were found, a scale of 0-1200 mm/year was used and for the remaining products a scale of 0-1800 mm/year was used. Visually, all products capture the forested area. Irrigated areas are also captured well by most products. GLEAM and LandFlux-EVAL do not capture the majority of selected irrigated areas. CMRSET, ETMonitor, SSEBop and WaPOR capture most of the selected irrigated areas while the remaining products capture a few. GLEAM, LandFlux-EVAL, MOD16, MTE and WECANN only estimate land ET and thus do not have ET across water bodies. The remaining products capture the water bodies well, with CMRSET and ETMonitor showing larger differences in their estimations of ET across water bodies than the surrounding areas over SSEBop and WaPOR. A ranking based on visual inspection of how well each ET product captures the selected land cover element can be found in Table 9.

Figures 11 and 12 show the bias between the mean ET across the forests and water bodies estimated by the ET products and the reference ET used for each element. All ET products capture ET across the selected forested area, however some perform better than others at describing the magnitude. CMRSET, SSEBop and WaPOR have very low biases with respect to the reference found in literature, while MOD16 and WECANN have the largest biases. All products underestimate ET across the forested area with respect to the used reference. The four products which estimate ET across water bodies, show relatively low biases with the reference PET. CMRSET overestimates ET while ETMonitor, SSEBop and WaPOR underestimate ET on average across water bodies. The lowest bias for water bodies is found in ETMonitor.

### 3.4 Product similarity assessment

Two groupings or clusters are observed when looking at the similarity between individual products (Fig. 13). We see one cluster formed with three products, CMRSET, SSEBop and WaPOR, with SSEBop and WaPOR being slightly more similar than with CMRSET. And a second cluster with the remaining products. Within the second cluster, LandFlux-EVAL and WECANN show the highest level of similarity which also coincides with having the same spatial resolution.

### 3.5 Ranking of products

Table 9 shows the ranking of the ET products based on the different assessment criteria. First we look at the ranking for statistics of the catchment water balance. In terms of bias and $bias_{aw}$ CMRSET, SSEBop and WaPOR are consistently ranked the highest while GLEAM is ranked the lowest. When looking at the RMSE and $RMSE_{aw}$ the same three products along with LandFlux-EVAL are ranked the top four while again GLEAM is ranked lowest. For correlation GLEAM and LandFlux-EVAL rank highest while SSEBop is ranked the lowest. Overall for comparison of calculated $ET_{WB}$ and ET calculated by the products, CMRSET, LandFlux-EVAL, SSEBop and WaPOR rank the highest while GLEAM and MOD16 rank the lowest. Second we look at the comparison with the reference condition of the Budyko curve. Here, the same ranking pattern can be seen, with CMRSET, LandFlux-EVAL, SSEBop and WaPOR ranking highest and GLEAM and MOD16 ranking the lowest. Thirdly we look at the spatial variability rankings. For spatial variability with visual inspection, CMRSET, ETMonitor, SSEBop and WaPOR rank the highest and LandFlux-EVAL and WECANN rank the lowest. For spatial variability with quantitative inspection we see that the same four products, CMRSET, ETMonitor, SSEBop and WaPOR rank the highest with GLEAM and WECANN ranking the lowest. Overall for spatial variability, CMRSET, ETMonitor, SSEBop and WaPOR rank highest while GLEAM and WECANN rank the lowest. The final ranking was conducted with and without visual inspection. The top four products, CMRSET, LandFlux-EVAL, SSEBop and WaPOR, do not vary in the two ranking schemes. GLEAM is also ranked lowest in both ranking schemes. Interesting to note is that ETMonitor ranks higher when including visual inspection while WECANN ranks higher when excluding visual inspection.

## 4 Discussion

We make two assumptions in this paper regarding the methodology applied for evaluating the selected ET products. The first assumption is that if no trends are present in long-term average $ET_{WB}$ across a basin, then long-term average $ET_{WB}$ across basins can be compared with different time periods. This is true if long-term trends in global ET are not visibly present. However, Jung et al. (2010) claim that there have been declining trends in global ET estimates in the recent past along with the last major El Niño event in 1998 with largest regional contributions to the declining trend in Australia and Southern Africa. The exact opposite effect is reported by Zhang et al. (2016) which claims significant increases in global land ET trends especially in Australia and Southern Africa. Other studies also focus on investigating trends in long-term ET and do not come to a consensus as to the cause or direction of the trend (Miralles et al., 2014; Douville et al., 2013; Jung et al., 2010; Zhang et al., 2016). With

this in mind, it is difficult to assume there is long-term global trend in one direction or the other. For this first assumption to hold, we must also address the possibility that regardless of whether there are no trends present, the mean ET from one period may be different from another period due to precipitation variability. In this case we analysed four basins for which the calculated $ET_{WB}$ estimations had a period sufficient enough to cover the time period of the range of ET products being

evaluated. For the four basins, $ET_{WB}$ was calculated for each of the different time periods of the ET products. We then found the bias from the the calculated long-term average $ET_{WB}$. From Table 10 we see that the percentage differences relative to total basin long-term average ET ranges from 0 to a maximum of 7.4 percent for the four basins evaluated and all ET products. Thus, considering the lack of consensus of the direction of a long-term global trend in ET and very low differences in precipitation variability, in this study our assumption holds that if no significant trend can be found in annual long-term ET estimates then

different time periods can be used due to lack of overlapping data.

The second assumption is that the water balance can be simplified to equation 1 where for long-term average estimates the change in storage is negligible. Many studies make this assumption for long-term averages and basin scale averages (Du et al., 2016; Taniguchi et al., 2003; Wang and Alimohammadi, 2012; Carter, 2001; Budyko, 1974). However a recent study by Rodell et al. (2018) quantified trends in terrestrial water storage using the Gravity Recovery and Climate Experiment (GRACE) data

for the period 2002- 2016. The largest annual trend found in this study is 20 mm per year and for the African continent can be found across sections of the Congo, Zambezi, Okavango, Cunene, Save and Rufiji basins. Of these basins Okavango, Cunene and Save are not used in this study and thus are not affected. Assuming a contribution of the largest trend in storage for the other basins this represents a maximum of 2.3 percent of the long-term annual average mean basin ET. Therefore we assumed negligible change in storage for our calculations.

The comparison between the RS products was carried out at the highest spatial resolution of the different products which is $0.0022° \times 0.0022°$. As we are resampling from coarse resolution to higher resolution, the nearest neighbor method employed for completing the resampling is sufficient, as the magnitude and spatial characteristics will not be altered or lost (Porwal and Katiyar, 2014; Gurjar and Padmanabhan, 2005). It must also be kept in mind that the initial spatial resolution and the temporal period under comparison are not the same for each product and this may effect the ranking that we are considering. However,

considering there are different resolution products available, this is an important feature in considering the ranking of products in terms of accuracy in order to make an educated decision on which product to use. Also many of the products do not estimate ET across water bodies and this may therefore explain the large biases in certain products when comparing ET estimations with the $ET_{WB}$ estimations. Another aspect to bear in mind, is that WaPOR, ETMonitor and WECANN have less than 10 years in total coverage in order to calculate their long-term average.

Evaluation of the spatial characteristics is completed using two steps, the comparison of land cover elements with reference estimates and visual interpretation. There are two issues involved in this spatial comparison. Firstly, the evaluation is taking place based on products originating with different resolutions. Thus, the view that higher resolution products may outperform the coarser resolution products, which is generally the case. However, we can also see that coarser resolution products, namely LandFlux-EVAL and in certain cases MTE and WECANN, outperform the higher resolution product GLEAM. Thus higher

resolution products do not always outperform lower resolutions as can be seen. The spatial resolution of the ET estimates used

may also be a critical element in determining which product is of use for the user. Secondly, the visual interpretation can be viewed as quite arbitrary and subjective according to the evaluator's eye. However, by using land cover elements that are large and easy to visualize, such as forested areas, irrigated areas and water bodies, the relative subjectivity can be reduced.

We used the assumption that where there is ample water ET equals PET (McMahon et al., 2013) and thus applied this assumption for evaluating our ET products for water bodies. The assumption holds quite well for the products that estimate ET over water. There are several reasons why it is difficult to find a quantitative reference for irrigated areas at such large magnitudes. Firstly, it is difficult to assume there is no mixing and only irrigated areas are found in pixels of a minimum of 250m x 250m. Secondly, an irrigated area of a particular size is often in reality growing more than one crop which is difficult to measure or map. A reference that could be used in subsequent studies would be to use water productivity (biomass/water consumed (ET)) for comparison.

The overall ranking for each product was based on the average ranking of the different comparative elements. An overall ranking was performed including the visual inspection of the land cover elements, however was also performed without, due to the subjectivity of the analyst doing the visual inspection. This does not affect the ranking of the top four or the lowest ranked product but changes the order of the products ranked in the middle. WaPOR, CMRSET, SSEBop and LandFlux-EVAL are consistently ranked 1, 2, 3 and 4 respectively. CMRSET and WaPOR rank first when including a visual inspection however only WaPOR ranks first without. The lowest ranked product is GLEAM in both cases. WECANN ranks higher without visual inspection from positions 8 to 6 and ETMonitor ranks lower without visual inspection going from position 5 to 7.

Looking at the overall level of similarity between the products in Fig. 5 we can see that for the cluster between CMRSET, SSEBop and WaPOR all products use MODIS as an input. SSEBop and WaPOR both use the P-M method for the calculation of ET, while CMRSET uses the P-T method. ETMonitor and MOD16 also use MODIS as an input with MOD16 using the P-M method for ET calculation and ETMonitor using both Shuttleworth-Wallace and the P-M method, however both are found in the second cluster. The remaining products within the second cluster use different inputs and different ET estimation methods. Thus, no patterns can be inferred through the cluster analysis by looking at the input or ET calculation method. What is clear is that the first cluster contains the products which have the highest spatial resolutions and which overall rank the best in terms of ET estimation based on the evaluation criteria.

In terms of consistency in results with previous studies conducted on some of the products under evaluation we see similar tendencies. According to Miralles et al. (2016) GLEAM, MOD16 and other products in their study show divergences in conditions of water stress and drought. Considering large parts of Africa are potentially under water stress due to the semi-arid and arid climate (IPCC, 2019; World Bank, 2018), this can explain the low ranking of GLEAM and MOD16 in this study. The RMSE and biases found in our study for Africa are comparable with those found by Vinukollu et al. (2011b) at the global scale, however comparing different products to that of this study. The range is higher in this study for Africa than the range found at the global scale. In their study, Trambauer et al. (2014) found GLEAM to underestimate ET in terms of their multi-product mean. This is again consistent with our finding where biases in GLEAM showed large underestimations across the basins in Africa with respected to the calculated $ET_{WB}$. We used the LandFlux-EVAL benchmark product as an ensemble product without calculating the multi-product mean of the products being used in this study, as it was developed using a large range

of ET products. LandFlux-EVAL, with the coarsest spatial resolution, ranked fourth in the final ranking only outranked by the products with the three highest spatial resolutions in this study, CMRSET, SSEBop and WaPOR. Therefore, LandFlux-EVAL performs well overall regardless of it's coarse resolution and is interesting due to being an ensemble product. Therefore, continuation or commencement of a similar initiative to develop a benchmark product using a range of ET data sets including
high resolution products ranked within this study may improve the ensemble product for future use.

It is also important to note that the overall ranking is interesting for global or large scale regional modellers however, for catchment studies a detailed look into their basin(s) of interest and local elements should also be considered. For example, if we look at the basin level bias and area weighted bias (Fig 8) for three of the large basins in Africa, the Congo, the Nile and the Niger basins, the following products have the lowest biases in the specified order: for the Congo basin, SSEBop, CMRSET
and WaPOR; for the Nile basin, MTE, SSEBop and CMRSET; and for the Niger basin, WaPOR, SSEBop and MOD16. This shows that a detailed look into the local characteristics of a particular basin is required before selecting a product for use. Due to the limited overlap between discharge data and ET estimations by the products, temporal evaluations were not possible. It would also be interesting and valuable to see which products capture temporal trends which may also effect the choice of a product.

**5  Conclusions**

This study focuses on the question of whether or not we can trust remote sensing and other ET products over Africa. By trying to overcome the problem of the lack of data for validation and evaluation purposes the methodology used can identify which products perform well in terms of biases and spatial characteristics. Using observations of discharge and observation based precipitation products to infer long-term average mean ET estimates at the basin scale and overcoming the lack of overlapping
data for comparison by using different time periods for calculation of our long-term averages, different ET products were evaluated. According to the comparison of the $ET_{WB}$ with $ET_{Budyko}$, we see that $ET_{WB}$ follows the Budyko curve and has an overall low bias across the basins. This indicates the calculated $ET_{WB}$ is a sound reference condition to use for analyses. Based on the different elements being analysed CMRSET, WaPOR and SSEBop capture the magnitude of ET showing small biases in the long-term average mean ET across basins. The same products also capture the spatial distribution of the ET patterns well
along with ETMonitor. Apart from the visual inspection, the ensemble product LandFlux-EVAL consistently ranks fourth or higher acting as a bridge between the products with the highest spatial resolutions and others. The high correlation statistics indicate good spatial distribution in all products, especially GLEAM and LandFlux-EVAL which rank the highest. However, nearly all products show relatively large biases in ET estimations, except CMRSET, SSEBop and WaPOR. It is difficult to come to a concrete judgement as to the reasons behind the differences between the ET products. A big difference between
the top three ranked products and the others is the high spatial resolution as well as the estimation of ET as a whole rather than only land ET in most other cases. However, no pattern can be found between the product ranking and the forcing or ET calculation methods. There are also certain advantages and disadvantages of the products outside of the evaluation criteria which are important to name. Although GLEAM is ranked lowest overall, the product has the longest temporal coverage

starting from 1980 and is on-going. LandFlux-EVAL and MTE also have early starting years however only go up to 2005 and 2012, respectively. ETMonitor is also no longer being extended and is not openly accessible or available for use. WaPOR is only available for Africa and not globally compared to all other products. Therefore, if we answer our question of whether to trust remote sensing estimates of ET across Africa, the answer is not black and white. Yes, in general we can trust the products

5   under evaluation in this study. CMRSET, WaPOR and SSEBop show low biases in estimations and a good spatial distribution of ET patterns. Each of these products have relatively high resolutions and both CMRSET and SSEBop are global products. Depending on the study under question, whether an early and long time period is needed, whether a higher or lower resolution is required, whether looking at the global or regional scale or whether looking only at land evapotranspiration, a different product may be more suited than another. However, a large consideration to be kept in mind for Africa, is that the three highest

10   ranked products, CMRSET, SSEBop and WaPOR have low biases and perform well in spatial variability and will suit most needs within a given study. However, for catchment scale studies within Africa a detailed look into the characteristics of the basin should be considered along with the overall ranking.

## Appendix A: Appendix A

*Author contributions.* IW and AVG conceived and designed the alternate methodology for evaluation of large scale RS ET products. IW performed the required data analysis using scripts written by IW. IW and AVG prepared the structure of the manuscript. IW wrote the initial draft of the paper. AVG and WB supervised the research and contributed to improving the manuscript prior to submission. MM contributed to improving the manuscript prior to submission. LJ made available ETMonitor data that is not openly accessible.

"Data Availability" - data used in this analysis that is openly accessible can be accessed when requested by emailing the first author.

*Competing interests.* The authors declare no conflict of interest.

*Acknowledgements.* This research is supported by the ERA4CS CIREG and TORUS projects. Most spatial data layers can be accessed through the public domain, however we would still like to thanks the teams providing the CMRSET, GLEAM, MOD16, MTE, SSEBop, WaPOR, WECANN, CHIRPS, MSWEP, EWEMBI and PET data sets. We thank the team at Global Runoff Data Centre (GRDC), 56068 Koblenz, Germany for providing the discharge data. We thank Wim Thiery, Steven Eisenreich, Inne Vanderkelen and Graham Jewitt for their advice on improvements to the scientific content and manuscript.

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

| Basin | Variable | Data Availability | Trend | hypothesis | p-value | z-value | no. of samples |
|-------|----------|-------------------|-------|------------|---------|---------|----------------|
| Awash | ET | 1990-2004 | no trend | false | 0.2496 | -1.1514 | 14 |
| | P | 1979-2016 | MK test not conducted, no trend found in ET | | | | 38 |
| | Q | 1990-2004 | | | | | 15 |
| Bandama | ET | 1979-1996 | no trend | false | 0.7619 | -0.3030 | 18 |
| | P | 1979-2016 | MK test not conducted, no trend found in ET | | | | 38 |
| | Q | 1970-1996 | | | | | 27 |
| Blue Nile | ET | not enough ET data points to conduct MK test on calculated ET | | | | | 4 |

| Basin | | Period | Trend | Significant | p-value | z-stat | N |
|---|---|---|---|---|---|---|---|
| | P | 1979-2016 | no trend | false | 0.6875 | -0.4023 | 38 |
| | Q | 1900-1982 | decreasing | true | 0.0009 | -3.3271 | 83 |
| Buzi | ET | not enough ET data points to conduct MK test on calculated ET | | | | | 5 |
| | P | 1979-2016 | no trend | false | 0.4210 | -0.8046 | 38 |
| | Q | 1957-1983 | no trend | false | 1.0 | 0.0 | 23 |
| Cavally | ET | 1979-1996 | no trend | false | 0.54449 | -0.6060 | 18 |
| | P | 1979-2016 | MK test not conducted, no trend found in ET | | | | 38 |
| | Q | 1970-1996 | | | | | 27 |
| Congo | ET | 1979-2010 | no trend | false | 0.0830 | -1.7336 | 31 |
| | P | 1979-2016 | MK test not conducted, no trend found in ET | | | | 38 |
| | Q | 1903-2010 | | | | | 108 |
| Cunene | ET | 1980-2015 | increasing | true | 0.0003 | 3.5823 | 36 |
| | P | 1979-2016 | MK test not conducted, no trend found in ET | | | | 38 |
| | Q | 1980-2015 | | | | | 36 |
| Gambia | ET | not enough ET data points to conduct MK test on calculated ET | | | | | 5 |
| | P | 1979-2016 | no trend | false | 0.2579 | 1.1315 | 38 |
| | Q | 1979, 1981-82, 1984,1988 | no trend | false | 0.8065 | 0.2449 | 5 |
| Groot | ET | 1979-2015 | no trend | false | 0.1697 | 1.3733 | 37 |
| | P | 1979-2016 | MK test not conducted, no trend found in ET | | | | 38 |
| | Q | 1964-2014 | | | | | 51 |
| Kamoe | ET | 1979-1996 | no trend | false | 0.3633 | -0.9091 | 18 |
| | P | 1979-2016 | MK test not conducted, no trend found in ET | | | | 38 |
| | Q | 1970-1996 | | | | | 27 |
| Lake Chad | ET | not enough ET data points to conduct MK test on calculated ET | | | | | 4 |
| | P | 1979-2016 | increasing | true | 0.0194 | 2.3384 | 38 |
| | Q | 1983-1986 | no trend | false | 0.3081 | -1.0190 | 4 |
| Maputo | ET | not enough ET data points to conduct MK test on calculated ET | | | | | 5 |
| | P | 1979-2016 | no trend | false | 0.3393 | -0.9555 | 38 |
| | Q | 1953-1983 | no trend | false | 0.1261 | -1.5297 | 31 |
| Mono | ET | 1979-2007 | no trend | false | 0.5115 | -0.6565 | 29 |
| | P | 1979-2016 | MK test not conducted, no trend found in ET | | | | 38 |
| | Q | 1944-2007 | | | | | 64 |
| Niger | ET | 1979-2006 | no trend | false | 0.6214 | 0.4939 | 28 |
| | P | 1979-2016 | MK test not conducted, no trend found in ET | | | | 38 |
| | Q | 1970-2006 | | | | | 37 |
| Nile | ET | not enough ET data points to conduct MK test on calculated ET | | | | | 6 |
| | P | 1979-2016 | no trend | false | 0.2909 | 1.0560 | 38 |
| | Q | 1912-1984 | no trend | false | 0.0693 | 1.8164 | 56 |
| Okavango | ET | 1979-2014 | increasing | true | 0.0127 | 2.4926 | 36 |
| | P | 1979-2016 | MK test not conducted, no trend found in ET | | | | 38 |
| | Q | 1950-2014 | | | | | 65 |
| Olifant | ET | 1979-2014 | no trend | false | 0.9457 | 0.0681 | 36 |
| | P | 1979-2016 | MK test not conducted, no trend found in ET | | | | 38 |
| | Q | 1927-2014 | | | | | 88 |
| Orange | ET | 1979-2016 | no trend | false | 0.6691 | 0.4274 | 38 |
| | P | 1979-2016 | MK test not conducted, no trend found in ET | | | | 38 |
| | Q | 1936-2014 | | | | | 79 |
| Queme | ET | 1979-80, 1982-84, 1990-2005, 2007 | no trend | false | 0.3377 | 0.9587 | 22 |
| | P | 1979-2016 | MK test not conducted, no trend found in ET | | | | 38 |
| | Q | 1948-2007 | | | | | 60 |
| Rufiji | ET | not enough ET data points to conduct MK test on calculated ET | | | | | 0 |
| | P | 1979-2016 | no trend | False | 0.6508 | -0.4526 | 38 |
| | Q | 1954-1978 | no trend | False | 0.9741 | -0.0324 | 20 |

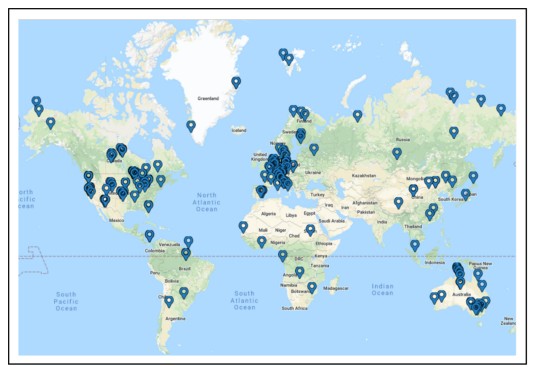 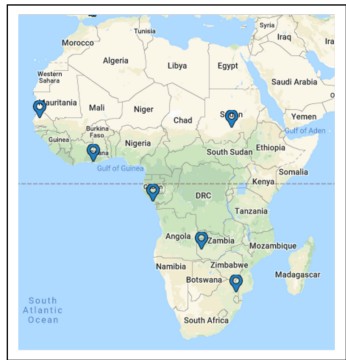

**Figure 1.** (left) distribution of flux towers worldwide. (right) distribution of flux towers across Africa (Google)

| | | | | | | | |
|---|---|---|---|---|---|---|---|
| Sassandra | ET | 1979-1996 | no trend | false | 0.8796 | 0.1515 | 18 |
| | P | 1979-2016 | MK test not conducted, no trend found in ET | | | | 38 |
| | Q | 1970-1996 | | | | | 27 |
| Save | ET | not enough ET data points to conduct MK test on calculated ET | | | | | 3 |
| | P | 1979-2016 | no trend | False | 0.8801 | 0.1509 | 38 |
| | Q | 1968-1981 | increasing | True | 0.0118 | 2.5183 | 14 |
| Senegal | ET | 1979-1989 | no trend | false | 0.2129 | 1.2456 | 11 |
| | P | 1979-2016 | MK test not conducted, no trend found in ET | | | | 38 |
| | Q | 1979-1989 | | | | | 11 |
| Tana | ET | not enough ET data points to conduct MK test on calculated ET | | | | | 0 |
| | P | 1979-2016 | decreasing | True | 0.0006 | -3.4447 | 38 |
| | Q | 1975-1978 | no trend | False | 0.7341 | -0.3397 | 4 |
| Upper Blue Nile | ET | not enough ET data points to conduct MK test on calculated ET | | | | | 8 |
| | P | 1979-2016 | no trend | False | 0.6875 | -0.4023 | 38 |
| | Q | 1961-1983 | no trend | False | 0.1339 | -1.4988 | 26 |
| Void | ET | not enough ET data points to conduct MK test on calculated ET | | | | | 3 |
| | P | 1979-2016 | no trend | False | 0.1251 | -1.5338 | 38 |
| | Q | 1979-1981 | increasing | True | 0.0483 | 1.9748 | 7 |
| Zambezi | ET | 1979-1990 | no trend | false | 0.5371 | 0.6172 | 12 |
| | P | 1979-2016 | MK test not conducted, no trend found in ET | | | | 38 |
| | Q | 1960-1990 | | | | | 31 |

**Table 1.** Characteristics of evapotranspiration products

| Feature | Global Spatial Coverage | Openly Accessible | Dekaadal or higher temporal resolution | Product ongoing | Available from 1990 or earlier | highest resolution | Lowest resolution |
|---|---|---|---|---|---|---|---|
| Possible advantage or disadvantage | Advantage in general. Possible disadvantage in losing features if coarse resolution. | Advantage as accessible for everyone | Advantage as captures more temporal features | Advantage as can still be accessed for the present | Advantage as available for a longer time period | Possible advantage as may capture more features | Possible disadvantage as may capture fewer features |
| CMRSET | Yes | Yes | Yes | Yes | No | Yes | No |
| ETMonitor | Yes | No | Yes | No | No | No | No |
| GLEAM | Yes | Yes | Yes | Yes | Yes | No | No |
| LandFlux-EVAL | Yes | Yes | No | No | Yes | No | Yes |
| MOD16 | Yes | Yes | No | Yes | No | No | No |
| MTE | Yes | Yes | No | No | Yes | No | No |
| SSEBop | Yes | Yes | No | Yes | No | No | No |
| WaPOR | No | Yes | Yes | Yes | No | Yes | No |
| WECANN | Yes | Yes | No | Yes | No | No | Yes |

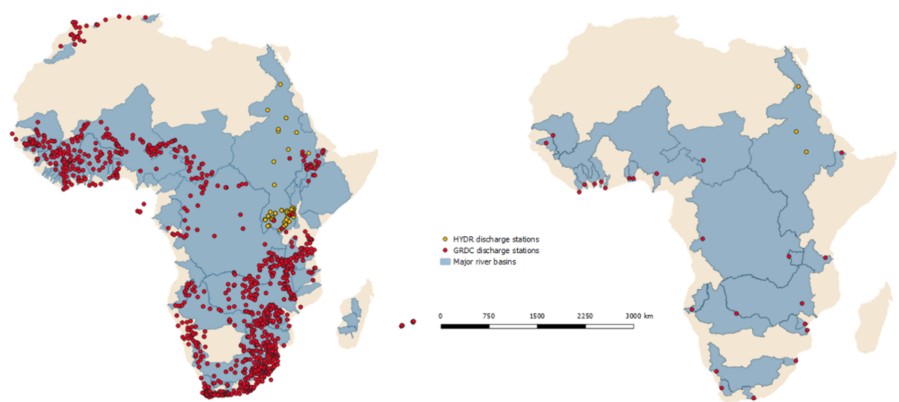

**Figure 2.** (left) All major basins in Africa and all available discharge stations; (right) Major basins in Africa with available discharge data at outlet

**Table 2.** Characteristics of remotely sensed ET products

| Product | Temporal Coverage | Spatial Coverage | Temporal Resolution | Spatial Resolution | Estimation Approach | Input Data Source | Reference |
|---|---|---|---|---|---|---|---|
| CMRSET (v20140423) | 2000-2013 | Global | 8-daily | $0.0022° \times 0.0022°$ | P-T Equation, relationship between EVI and GVMI | MODIS | (Guerschman et al., 2009) |
| Access: http://remote-sensing.nci.org.au/u39/public/html/wirada/index.shtml | | | | | | | |
| ETMonitor | 2008-2013 | Global | daily | $0.005° \times 0.005°$ | P-M, Gash model, Shuttleworth-Wallace | MODIS | Zheng et al. (2016) |
| Access: email first author in reference | | | | | | | |
| GLEAM (v3.2a) | 1980-2016 | Global | Daily | $0.25° \times 0.25°$ | P-T Equation, soil stress factor | AMSR-E, LPRM, TRMM | Martens et al. (2017); Miralles et al. (2011) |
| Access: www.gleam.eu | | | | | | | |
| LandFlux-EVAL | 1989-2005 | Global | Monthly | $1° \times 1°$ | Ensemble Approach | See reference | Mueller et al. (2013b) |
| Access: https://iac.ethz.ch/group/land-climate-dynamics/research/landflux-eval.html | | | | | | | |
| MOD16 (vA3) | 2000-2014 | Global | Monthly | $0.0083° \times 0.0083°$ | P-M Equation, surface conductance model | MODIS | Mu et al. (2011, 2007) |
| Access: https://modis.gsfc.nasa.gov/data/dataprod/mod16.php | | | | | | | |
| MTE (vMay12) | 1982-2012 | Global | Monthly | $0.5° \times 0.5°$ | MTE approach, training using in-situ observations, flux tower data | Eddy Covariance, in-situ | Jung et al. (2011) |
| Access: https://climatedataguide.ucar.edu/climate-data/fluxnet-mte-multi-tree-ensemble | | | | | | | |
| SSEBop (v4) | 2003-2017 | Global | Monthly | $0.0096° \times 0.0096°$ | P-M Equation, ET fractions from $T_s$ estimates | MODIS | Senay et al. (2013) |
| Access: https://earlywarning.usgs.gov/fews/search | | | | | | | |
| WaPOR (v1.1) | 2009-2017 | Africa | Dekadal | $0.0022° \times 0.0022°$ | P-M Equation, calculates E, T and I separately | MODIS, GEOS-5/MERRA | FAO (2018) |
| Access: https://wapor.apps.fao.org/home/1 | | | | | | | |
| WECANN (v1.0) | 2007-2015 | Global | Monthly | $1° \times 1°$ | ANN approach, training using observations and model based LE | GOME-2 | Alemohammad et al. (2017) |
| Access: https://avdc.gsfc.nasa.gov/pub/data/project/WECANN/ | | | | | | | |

**Table 3.** Characteristics of precipitation products

| Product | Temporal Coverage | Spatial Coverage | Temporal Resolution | Spatial Resolution | Input Data Source | Reference |
|---------|-------------------|------------------|---------------------|--------------------|-------------------|-----------|
| EWEMBI (v1.1) | 1979-2016 | Global | daily | $0.5° \times 0.5°$ | ERA-Interim, WFDEI: (Weedon et al., 2014), E2OBS: (Calton et al., 2016) | (Lange, 2019) |
| | Access: http://doi.org/10.5880/pik.2019.004 | | | | | |
| CHIRPS (v2.0) | 1981-2019 | quasi-Global | daily | $0.05° \times 0.05°$ | in situ precipitation gauges, TRMM: (Huffman et al., 2007), CMORPH: (NCAR, 2017) | Funk et al. (2015) |
| | Access: https://chc.ucsb.edu/data/chirps | | | | | |
| MSWEP (v2.2) | 1979-2017 | Global | 3-hourly | $0.1° \times 0.1°$ | in situ precipitation gauges, CMORPH, TRMM, GSMaP: (JAXA, 2009), IRA-Interim | Bai and Liu (2018) |
| | Access: http://www.gloh2o.org/ | | | | | |

**Table 4.** Characteristics of discharge data

| Product | Temporal Coverage | Spatial Coverage | Temporal Resolution | Spatial Resolution | Input Data Source | Reference |
|---------|-------------------|------------------|---------------------|--------------------|-------------------|-----------|
| GRDC (v1.1) | 1806-2019 | Global | daily | point data | in situ discharge gauges | |
| | Access: https://www.bafg.de/GRDC/EN/Home/ | | | | | |
| HYDR-VUB | 1932-2018 | Global | daily | point data | in situ discharge gauges | |
| | Access: on request to http://www.hydr.vub.ac.be/ | | | | | |

**Table 5.** Characteristics of potential evapotranspiration products

| Product | Temporal Coverage | Spatial Coverage | Temporal Resolution | Spatial Resolution | Input Data Source | | Reference |
|---------|-------------------|------------------|---------------------|--------------------|-------------------|---|-----------|
| Hargreaves | 1979-2012 | Global | daily | $0.05° \times 0.05°$ | WFDEI, DEM | SRTM | (Sperna Weiland et al., 2015) |
| | Access: https://wci.earth2observe.eu/ | | | | | | |
| Penman-Montieth | 1979-2012 | Global | daily | $0.05° \times 0.05°$ | WFDEI, DEM | SRTM | (Sperna Weiland et al., 2015) |
| | Access: https://wci.earth2observe.eu/ | | | | | | |
| Priestly-Taylor | 1979-2012 | Global | daily | $0.05° \times 0.05°$ | WFDEI, DEM | SRTM | (Sperna Weiland et al., 2015) |
| | Access: https://wci.earth2observe.eu/ | | | | | | |

**Table 7.** Calculated statistics, bias, bias$_{aw}$, RMSE, RMSE$_{aw}$ and r, for the comparison of the long-term annual average ET$_{WB}$ versus ET$_{RS}$

| | CMRSET | ETMonitor | GLEAM | LandFlux-EVAL | MOD16 | MTE | SSEBop | WaPOR | WECANN |
|---|--------|-----------|-------|---------------|-------|-----|--------|-------|--------|
| bias | -19 | 156 | 254 | 115 | 131 | 146 | 12 | -3 | 139 |
| bias$_{aw}$ | -18 | 237 | 313 | 148 | 266 | 183 | 30 | -46 | 223 |
| RMSE | 113 | 211 | 273 | 152 | 199 | 184 | 163 | 104 | 189 |
| RMSE$_{aw}$ | 187 | 502 | 594 | 304 | 590 | 424 | 123 | 165 | 520 |
| r | 0.94 | 0.91 | 0.97 | 0.97 | 0.91 | 0.95 | 0.89 | 0.96 | 0.95 |

**Table 8.** Bias between the ET$_{Budyko}$ and, ET$_{WB}$ and ET$_{RS}$

| mm/year | ET$_{WB}$ | CMRSET | ETMonitor | GLEAM | LandFlux-EVAL | MOD16 | MTE | SSEBop | WaPOR | WECANN |
|---------|-----------|--------|-----------|-------|---------------|-------|-----|--------|-------|--------|
| bias | 42 | 101 | 202 | 284 | 152 | 185 | 177 | 140 | 86 | 180 |

**Table 9.** Ranking of the RS products based on the different evaluation steps of the proposed methodology

| | CMRSET | ETMonitor | GLEAM | LandFlux-EVAL | MOD16 | MTE | SSEBop | WaPOR | WECANN |
|---|---|---|---|---|---|---|---|---|---|
| **Catchment water balance ranking (CWB)** | | | | | | | | | |
| bias | 3 | 8 | 9 | 4 | 5 | 7 | 2 | 1 | 6 |
| bias$_{aw}$ | 1 | 7 | 9 | 4 | 8 | 5 | 2 | 3 | 6 |
| RMSE | 2 | 8 | 9 | 3 | 7 | 5 | 4 | 1 | 6 |
| RMSE$_{aw}$ | 3 | 6 | 9 | 4 | 8 | 5 | 1 | 2 | 7 |
| r | 6 | 7 | 1 | 1 | 7 | 4 | 9 | 3 | 4 |
| **Overall CWB ranking** | 2 | 8 | 9 | 3 | 7 | 5 | 4 | 1 | 6 |
| **Budyko ranking** | | | | | | | | | |
| Budyko | 2 | 8 | 9 | 4 | 7 | 5 | 3 | 1 | 6 |
| **Spatial variability ranking** | | | | | | | | | |
| **Visual Inspection (VI) - land cover elements** | | | | | | | | | |
| Forest | 2 | 2 | 6 | 8 | 1 | 7 | 2 | 2 | 9 |
| Irrigated Area | 1 | 4 | 8 | 9 | 5 | 6 | 3 | 2 | 7 |
| Water Bodies | 1 | 1 | n/a | n/a | n/a | n/a | 3 | 3 | n/a |
| **Overall VI spatial ranking** | 1 | 2 | 7 | 9 | 5 | 6 | 4 | 3 | 8 |
| **Quantitative Inspection (QI) - land cover elements** | | | | | | | | | |
| Forest | 1 | 6 | 7 | 4 | 8 | 5 | 3 | 2 | 8 |
| Water Bodies | 2 | 1 | n/a | n/a | n/a | n/a | 3 | 4 | n/a |
| **Overall QI spatial ranking** | 1 | 4 | 9 | 5 | 7 | 6 | 2 | 2 | 7 |
| **Overall spatial ranking** | 1 | 3 | 8 | 7 | 5 | 6 | 3 | 2 | 9 |
| **Final ranking** | | | | | | | | | |
| With visual inspection | 1 | 5 | 9 | 4 | 7 | 6 | 3 | 1 | 8 |
| Without visual inspection | 2 | 7 | 9 | 4 | 8 | 5 | 3 | 1 | 6 |

**Table 10.** Differences in mean WB ET estimations for varying RS product periods

| | Total | CMRSET | ETMonitor | GLEAM | Period MOD16 | MTE | SSEBop | WaPOR | WECANN | Average |
|---|---|---|---|---|---|---|---|---|---|---|
| **Congo** | | | | | | | | | | |
| | 1979-2010 | 2000-2010 | 2008-2010 | 1980-2010 | 2000-2010 | 1982-2010 | 2003-2010 | 2009-2010 | 2007-2010 | |
| ET mm/year | 1186 | 1203 | 1159 | 1196 | 1203 | 1194 | 1194 | 1168 | 1193 | 1189 |
| Bias mm/year | | 17 | 27 | 10 | 17 | 8 | 8 | 18 | 7 | 14 |
| % bias | | 1.4 | 2.3 | 0.8 | 1.4 | 0.7 | 0.7 | 1.5 | 0.6 | 1.2 |
| **Groot** | | | | | | | | | | |
| | 1979-2015 | 2000-2013 | 2008-2013 | 1980-2015 | 2000-2014 | 1982-2012 | 2003-2015 | 2009-2015 | 2007-2015 | |
| ET mm/year | 373 | 390 | 381 | 377 | 390 | 371 | 387 | 396 | 392 | 386 |
| Bias mm/year | | 17 | 8 | 4 | 17 | 2 | 14 | 23 | 19 | 13 |
| % bias | | 4.4 | 2.1 | 1.1 | 4.4 | 0.5 | 3.6 | 5.8 | 4.9 | 3.4 |
| **Olifant** | | | | | | | | | | |
| | 1979-2014 | 2000-2013 | 2008-2013 | 1980-2014 | 2000-2014 | 1982-2012 | 2003-2014 | 2009-2014 | 2007-2014 | |
| ET mm/year | 278 | 279 | 293 | 284 | 278 | 286 | 272 | 275 | 296 | 283 |
| Bias mm/year | | 1 | 15 | 6 | 0 | 8 | 6 | 3 | 18 | 7 |
| % bias | | 0.4 | 5.1 | 2.1 | 0.0 | 2.8 | 2.2 | 1.1 | 6.1 | 2.5 |
| **Orange** | | | | | | | | | | |
| | 1979-2015 | 2000-2013 | 2008-2013 | 1980-2015 | 2000-2014 | 1982-2012 | 2003-2015 | 2009-2015 | 2007-2015 | |
| ET mm/year | 349 | 377 | 376 | 356 | 374 | 362 | 351 | 350 | 351 | 362 |
| Bias mm/year | | 28 | 27 | 7 | 25 | 13 | 2 | 1 | 2 | 13 |
| % bias | | 7.4 | 7.2 | 2.0 | 6.7 | 3.6 | 0.6 | 0.3 | 0.6 | 3.6 |

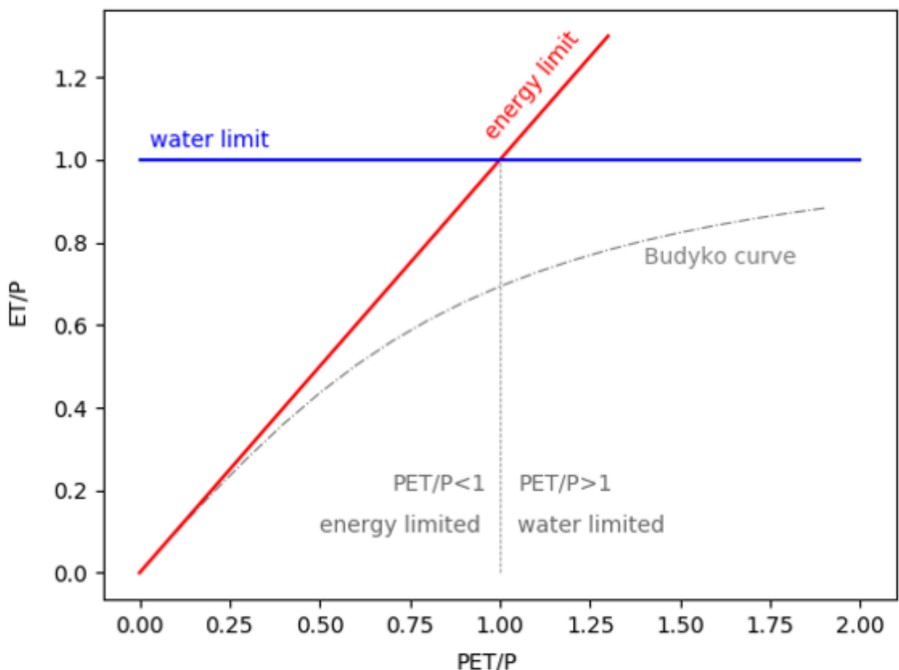

**Figure 3.** Budyko curve showing the energy limit and water limit

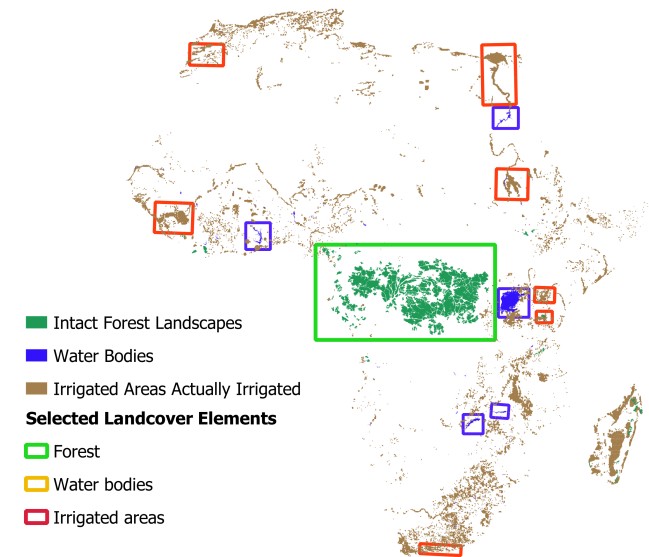

**Figure 4.** IFL, WB GRanD and AEIai land cover element maps and areas selected for visual inspection

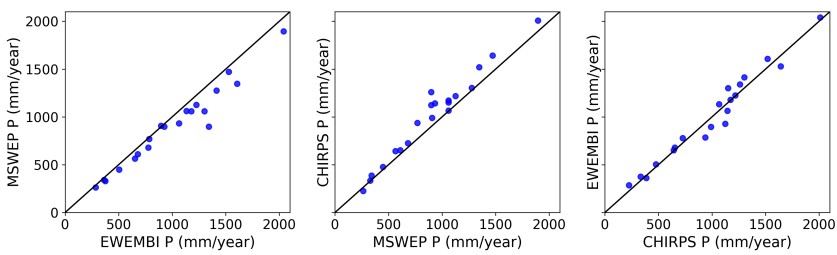

**Figure 5.** Comparison of the EWEMBI, MSWEP and CHIRPS precipitation products on their prediction of mean P across the basins

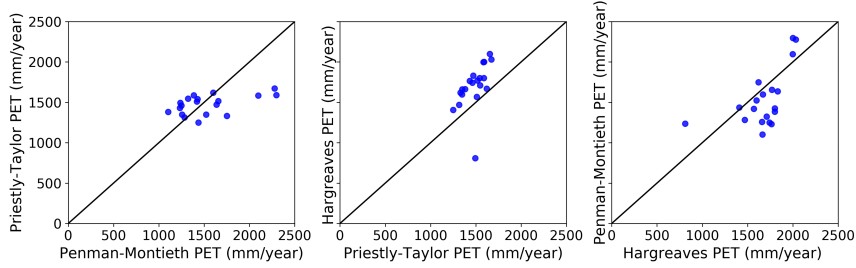

**Figure 6.** Comparison of the P-M, P-T and Hargreaves potential evapotranspiration products on their prediction of mean PET across the basins

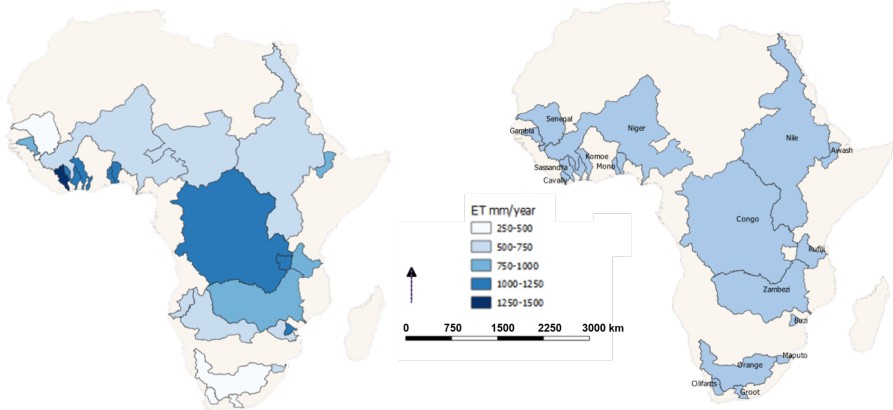

**Figure 7.** (right) ET$_{WB}$ estimation for 28 major basins in Africa using P-Q (left) Final basins being analysed after analyses to discount basins with trends in ET$_{WB}$, P and/or Q.

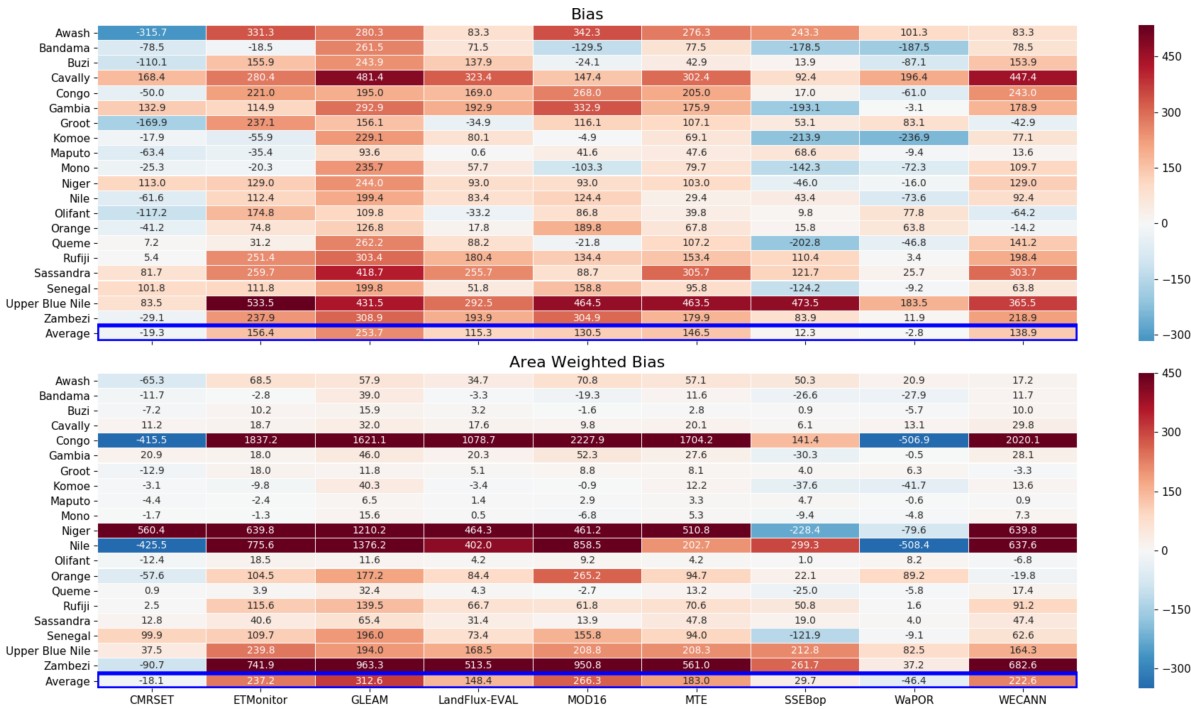

**Figure 8.** Bias and basin area weighted bias between the long-term annual average calculated $ET_{WB}$ and $ET_{RS}$ for all basins and the average of the 20 basins

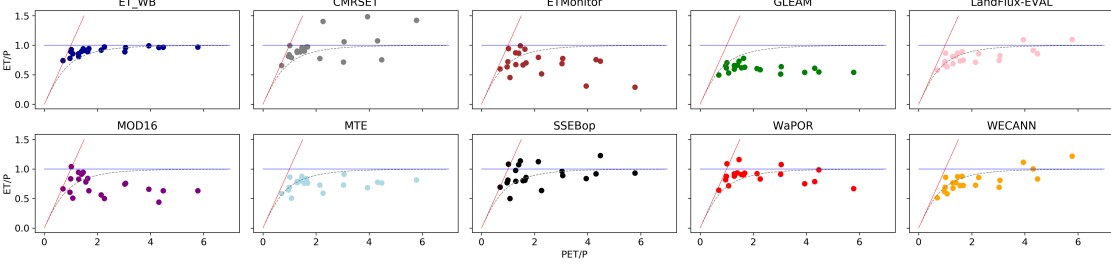

**Figure 9.** Evaluation of the calculated $ET_{WB}$ and $ET_{RS}$ from products using the Budyko curve calculated using average P and PET from three products

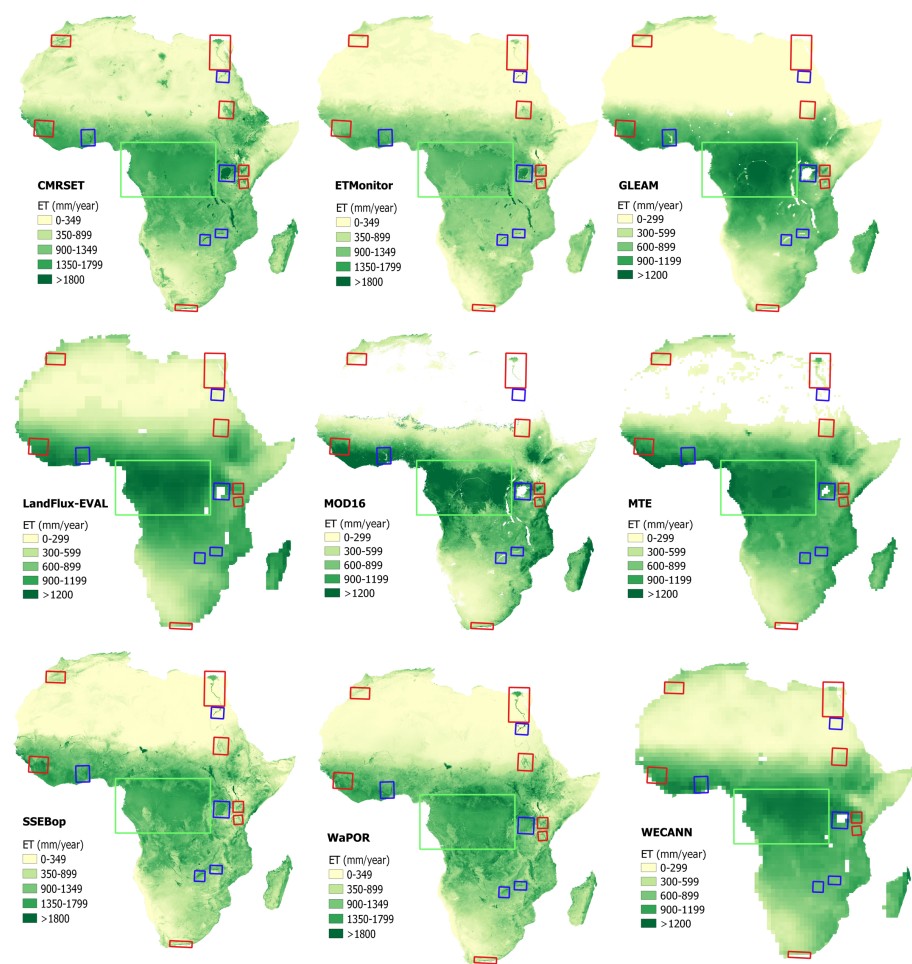

**Figure 10.** Spatial assessment across Africa of each ET product based on selected land cover elements, forest, irrigated areas and water bodies. Red boxes indicate irrigated areas, blue boxes indicate water bodies and green boxes indicate forested areas

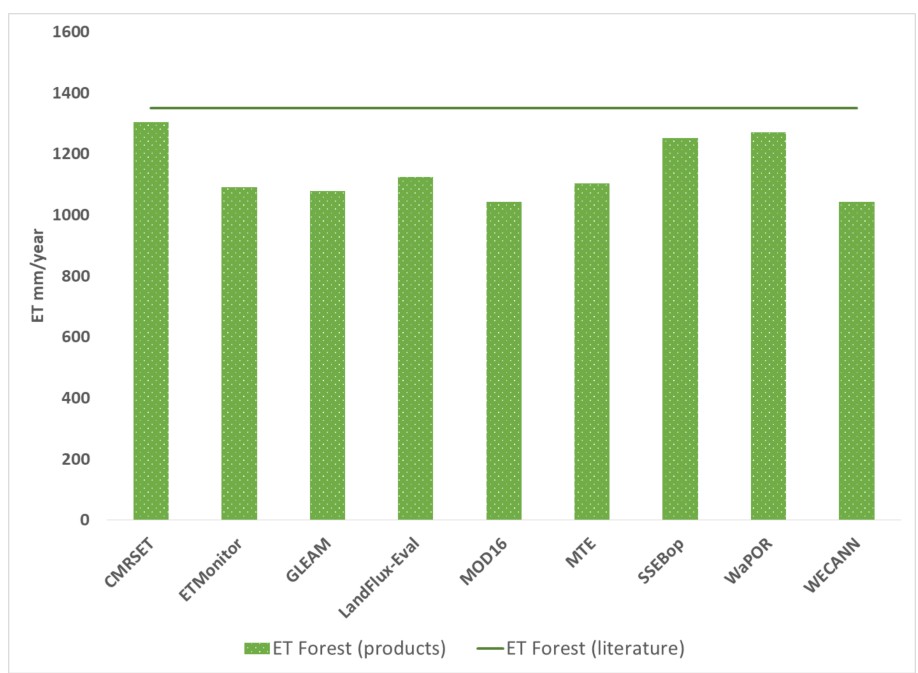

**Figure 11.** Comparison of mean ET across the selected forested area for each product versus mean ET found from literature

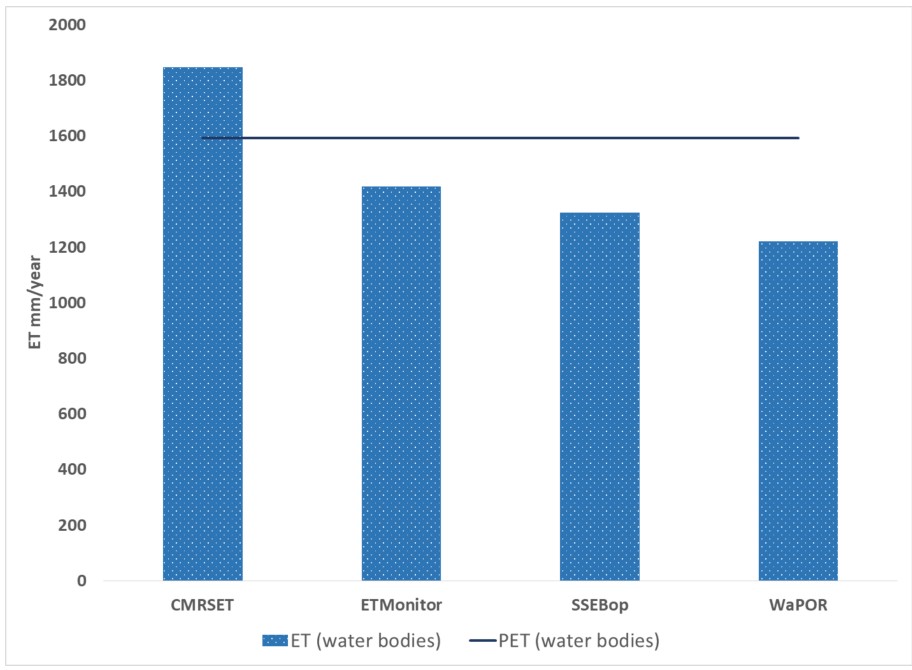

**Figure 12.** Comparison of mean ET across water bodies estimated by each ET product and PET using the average of three PET products

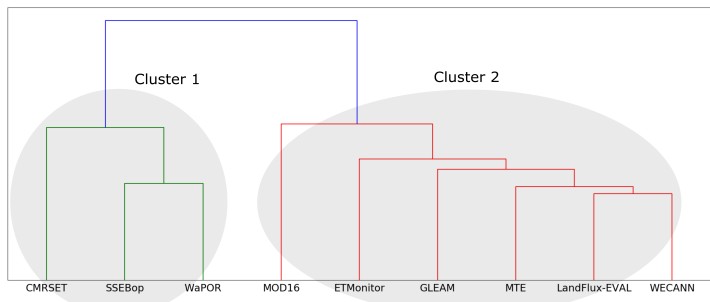

**Figure 13.** Cluster analysis based on the pairwise Euclidean distance between each pixel for each ET product to assess overall similarity between data sets