# Peer review of "Can we trust remote sensing ET products over Africa?"

_Hydrology and Earth System Sciences, 2019_

## Referee Comment (RC1) · Anonymous Referee #1 · 11 Jul 2019

Summary The manuscript has followed a commendable approach to evaluate eight diverse ET products and presented a ranking of the different products on data sparse region. The method evaluated 8 products using a basin water balance ET and Budyko curve over several basins across Africa using the average of three precipitation products along with observed runoff data. Care was taken to ensure the assumption of negligible storage change over several years by removing basins that showed trends using the MK test.

The manuscript is well-written with a useful application and contribution to the remote sensing community. I have a few general and specific comments that could improve the manuscript.

General: Considering Figure 6, 7 and 8 are key results for the ranking shown in Table

3, the method needs to flesh out how the data points are generated. For example, in Figure 6 seems to show correlation (r) across basins using the mean value for RS and WB ET. As indicated the correlation values are strong for all, but a root mean square error (RMSE) may have been a more useful metrics to compare the different models as that includes bias information. Also, it is not clear if the r difference between adjacent models is significantly different to rank them in a different order. I would think assigning a different rank order when the "r" are not significantly different may inflate the order. But the use of RMSE in the ranking may be more robust and it is not clear why this are not used. Similarly, Figures 7 and 8 could benefit from statement that the table values represent one data point for each basin and the average is the average of all basins, if that is correct? But unless the values in Figures 7 and 8 are missing negative biases, it is not clear how the average becomes so small when the percentage difference in each basin is much higher, as much as 73%. The difference between Figures 7 and 8, i.e., average and weighted average is not clear. Are the weights (basin area) assigned only to the RS ET or to both RS and WB ET and in that case does this mean volumetric ET difference? Again, a more detailed description is required in the methods section.

Landcover: it is not clear why the study did not include more land cover types, especially knowing the chosen two landcovers (water and irrigated lands) may not be handled well by some of the models

Specific comments; 1) Tables and figures would need improved captions and header names that would help them stand alone. 2) Figures 7 and 8 may benefit from one more panel which shows the average of the three precipitation products as the ranking is based on the average the three. 3) Zoom-in maps: it is hard to see the differences in Figures 9 and 10 among models. Maybe it is better to show deviations from the MPM data, i.e., show MPI in mm but the rest of the models as differences from MPM. Also remove the grid lines—hard to read the maps. A better color ramp will help readability. 4) Revisit carefully the description and citation of some products. For example, SSEB vs SSEBop. As far as I know the global product is from SSEBop with a different citation

with a 10-day (dekad) time scale, not monthly. Model's pre-defined boundary limits are described in SSEBop's work and not in the indicated citations. 5) It will be useful to include data source (website link) of the different models for access and discuss why the different models appear to discontinued. 6) Include some discussion on the performance of MTP in relation to the WB ET (rank 5) and the value for MTP or ensemble products for future use. 7) Table 2: not clear what "not enough data" is referring to.

---

## Referee Comment (RC2) · Anonymous Referee #2 · 10 Aug 2019

The article "Can we trust remote sensing ET products over Africa?" by Imeshi Weeras-inghe et al. presents an evaluation analysis of the eight satellite - based evapotranspi-ration (ET) products over selected African river basins against the ET estimates derived from the water balance equation. The main conclusion of the study ranks the selected ET products in accordance with the results of the comparison analyses.

The topicality and scientific relevance of the research question addressed in this study is high considering the sparseness of the in situ ET data in the region as well as the urgency of having a high quality ET estimates for the climate related problems in Africa.

However, at this point I cannot recommend publication of this article as it (i) – contains a number of significant methodological inaccuracies and (ii) – gives poor explanation and presentation of the performed analyses and graphics. Also, stylistically and structurally

the manuscript needs a substantial improvement. I highly recommend a major revision of the manuscript followed up by an internal review prior to the re-submission. Details follow.

Major comments:

=================

Generally, the presentation style of the paper makes it often hard to understand the correctness and hence the added value of the illustrated results. The lack of accompanying relevant information in a well-written form along with the multitude of presented data combinations in a variety of forms and at different scales in many cases confuses an understanding of (i) which data sets were used for this concrete calculation, (ii) in which form the data went into the following graphic, (iii) what the estimates were compared to, (iv) how many and which basins were used this time, (v) when data mean was used and over which scales the averaging was done? In my opinion, the paper did not succeed in wrapping up the results in a clear manner. Usage of multiple data levels, i.e. 3 reference rain datasets, 8 ET products, 27/20 basins with/without trends, different temporal resolutions (from one value to time-series), two spatial levels (from basin integral to pixel-basis) comprises a fairly large number of levels of information which the authors should unwrap and present in a very simple, consequent and logical manner. In the present version of the paper this have not been archived.

The presentation style needs a thorough improvement, including restructuring of the manuscript, improvement of English grammar and scientific wiring style itself. One of the major remarks is that the whole manuscript text is written in a very intermittent and superficial manner. The explanations throughout the whole paper are significantly lacking concreteness. Also, confusion and replacement of some terms used throughout the study (e.g. trends as trends or trends as tendency to show certain value, among others) together with the multitude of abbreviations used in the text makes it very hard to follow the presentation (details given below).

Finally, some data descriptions and methodological assumptions (in particular, down-scaling to the smallest grid and usage of different time periods) raises a number of questions related to their correctness and validity. The details are given further below.

More specifically:

==============

* Scientific relevance:

1. From the abstract, introduction and methodology sections it remains unclear how new is the water balance (WB) method, how accurate is the method and which other studies already used it for similar tasks. The abstract even makes a false impression that the authors developed a method and not used the existing one (p1, L6). The introduction in turn makes an impression that the focus of the study is the methodology (p3, L9) and not the evaluation of the existing products.

In this view, the authors must provide an overarching literature review of the studies that already used the WB ET estimates for satellite products evaluation, and also studies which evaluated the same satellite products over Africa using the same or other techniques. One of such study examples is the Miralles et al. (2016), which is also referenced in the present manuscript. Note, that Miralles' study also involved the African river basins. This has to be explicitly mentioned in the introduction. The authors should also then place their results into the findings of others. This is not done at the present state of the paper.

2. The authors should be more careful in formulating their scientific conclusions. The following sentence in the abstract: "However our recommendation would be the three highest ranked products being CMRSET, SSEBop and WaPOR." sounds rather subjective and needs further motivation (especially considering the huge differences in spatial-temporal scales between the products, as well as the manipulations on interpolation, then vice versa - integration - done within the study). Why first three? The same

remark applies for the conclusions.

3. The study does not mention anything at all about the quality of the reference precipitation data sets, nor about the quality of the final WB- based ET product. The study should also provide or mention some quantitative assessment to the magnitude of the differences which arise only due to application of different rainfall products. This will (i) - substantiate introduction of three different rainfall products in the paper and (ii) – justify better the obtained differences in ET between the products.

* Data products:

1. The data section needs a major elaboration. Many paragraphs appear rather like a snippets of information with lack of logical sequence, and hence they often fail to deliver main message of the paragraph or peculiarity of the concrete product. My suggestion would be: i) to either extend the data product descriptions to make them more complete and understandable or vice versa, provide only a reference links to the web sources and main papers of the products, and use instead the data section to discuss / group the products by their similarities and differences, advantages and disadvantages which can further help interpreting the paper results. ii) to omit the repetition of the time period and resolution information since they are already given in the table; iii) to place all products into the tables for consistency and clarity, i.e. also precipitation products, discharge data and reference data should be summarised in the same or separate table.

2. Many product descriptions and references miss version numbers. Those must be included, since depending on the product version there might be some already known issues related to a parameter derivations.

3. Check carefully the correct citations, once you add the product version numbers. I am more familiar with the GLEAM product, and I know that for GLEAM v3 (if you used that version) the correct references are Miralles 2011 (HESS) and Martens 2016 (GMD).

4. It is also a rule of a good scientific practice to provide/cite the data source: a web-page, ftp or a personal communication. No data sources are mentioned in the current manuscript version. For GLEAM, for example it should be the web portal: www.gleam.eu; For MSWEP: http://www.gloh2o.org/ (?), etc.

5. One of my major remarks here concerns inaccurate or sometimes even false information in data set descriptions. That is unacceptable. Please, check carefully all product descriptions you are giving!

On the example of GLEAM: - GLEAM is not a physically-based model, Prietsley-Taylor, the interception loss model, the stress module, and the water-balance model in GLEAM which form the core of GLEAM are all empirical! One can call it a process-based model, as it empirically describes the process needed to estimate E from satellites; - Table 1: GLEAM does not use CMORPH at all! That is simply wrong information. - Alemohammad et al., 2017 is the reference to a paper where they describe another method of deriving E. It is not clear why this is included in this GLEAM section?

* Methodology and results:

The methodological and result parts (i) - generally raise many questions as a result of incomplete and poor description of the calculation steps, applied quality assumptions and figures, and (ii) - casts doubt on the validity of some methodological steps and hence, on the accuracy of the final study results. Details follow.

1. As it was already mentioned earlier, the presentation of the calculation steps is done in a rather superficial manner. Lots of information is not given or remains unclear. E.g.: - which concrete quality control steps were involved in the selection of the basins and, which additional analyses were done and by whom? (e.g. p8, L7-10) - how were the basin boundaries defined? - what is the time-period of available discharge data? - how the integration over the basins is exactly done? were the simple mean or the area-weighted mean of ET or P fields used when averaging over the basin area? - which manipulations were done with the precipitation data prior averaging it over the basins?

Were the data also re-scaled to the 0.0022 deg resolution and then averaged over the basins? Never mentioned. - In their paper Miralles et al., 2016 applied additional quality control check on the difference between the GRDC-reported area and the area calculated from basin boundaries. Would not it be also relevant for the present study? - How was the MPM calculated? The products have big differences in resolution. The averaging of the products to get the MPM without applying corresponding weights can be a source of errors.

2. Correlation analysis:

(i) – It remains not very clear from the text over which values the correlation analysis is performed? Over time-series of annual means? Over multi-year averages of different basins? Should be put more clear. (ii) - Units of the correlations are not common, and confuse the interpretation. Correlation should rank from -1 to 1. Besides, it is never clear from all the graphs with percentages, by what value the normalisation was done. (iii) – Correlations should always provide significance measure, or the latter should be mentioned in the text.

3. The choice of the highest resolution is one of the two major remarks that I have to the methodological part: (i) – Generally, it is not common to interpolate products to the highest resolution, especially when the difference between the highest and the coarsest resolution is that high. It would be more correct to upscale the higher resolved data to the coarser estimates to minimise the bias. (ii) – Besides, the fact that the comparison of the products is mostly done at the basin level, the downscaling does not seem to make sense at all. First you interpolate the coarse data to the very high resolution, and then, you integrate it back again over the river basin. This clearly can be a source of additional biases and errors, which also raises my doubts about the validity of the ranking results. (iii) – All the above inter alia also raises a question of what is the minimal area of the smallest basin you have, and whether it is resolved by the products with the coarsest resolution at all?

4. The second and the most major remark of mine is related to the application of the analyses at different time-periods.

(i) - First of all it never comes clear what is the time period of available discharge data for every basin; (ii) – From the Tabel 2 it appears that for most of the basins discharge data does not extend the whole period of available precipitation data at all, and the spread of data periods is huge among the basins. In this view I do not understand at all how the analyses tests were done?

(ii) – The test for the effect of temporal variability on annual means mentioned in the discussion section was done only for the four basins, while 20 basins are analyzed throughout the study. Moreover, Congo - one of the four tested basins - has data only till 2010, while remote sensing ET products span up to 2017. In this view, I would not be able to call it a fair validity test!

(iii) – Clearly, the exclusion of periods with trends does not account for the temporal variability of data which can still result in the pretty different annual means. So, the effect of temporal variability on annual means must be done for all the basins, which are used for the evaluation of the satellite ET products in order to draw a fair conclusions.

(iv) – Calculating trends only for the WB ET reference data set is not a complete analysis. If a satellite data product has a trend, this also has to be mentioned, and maybe even that product should not participate in the validation (?)

To conclude, if the tests will show that the variability indeed matters, then none of the performed analyses is valid since they will all be affected by the differences due to variability.

5. My advice would be to not use percentages for all the figure results. This only confuses the interpretation. Use -1 to 1 scale for correlation and differences.

6. The raking of products based on visual inspection is rather speculative for me. For ex on Fig 9 it was impossible to follow the text conclusions: I did not see where irrigation

area is, to which reference product other products are compared, why MPM and some other products have no data and why GLEAM is concluded to perform worst? The same for Fig 10.

6. The analyses of comparing products over one irrigation area, and one lake, where some products have no data, and others do not even resolve the region does not make much sense to me, nor it is complete enough to make a serious conclusion on which product is better or worst.

8. It maybe due to the presentation style, but I could not follow the result section presenting the crop coefficients very well. It has to be structured and presented in a more clear manner: objective and reasoning for location, crop types, etc, data used, hypothesis to prove, which products in which form are tested, and what do results show.

* Stylistic and structural comments:

I highly recommend to look into the papers of Zeng et al., 2012 (ERL) and Miralles et al., 2016 (HESS) as an example of a good presentation style, and especially of the methodological part, as well as their choice of graphs.

Few comments:

1. Usage of term throughout the paper changes and confuses the reader. For ex, the term trend is first used as trend itself, but also to indicate tendencies if I understood it correctly (p13, L5 or p16,L9)

2. Typos are also present throughout the paper (e.g. p3 L2, p7 L17, p15 L7, p20 L3). A proper internal resew is required.

3. Discussion section rather reads like methods and should be incorporated to methods. Instead, the discussion section should place the paper findings into the existing knowledge as was already mentioned earlier.

4. Nothing is mentioned about the Budyko result in Discussion or Conclusions

5. Nothing is mentioned about the differences between using three precipitation products

6. Many abbreviations are not opened. Add abbreviation table to the paper. Use less abbreviations, i.e. if possible leave it open. Very hard to follow

7. Explanation of the results in Figures are often not complete. Not clear which products were used, which reference, etc. Figures are too small.

8. Lots of sentences are too wake, "Based on the elements being analysed... p21, L30" What is meant? Be more concrete.

9 . Titles of the sections should be reconsidered

---

## Author Comment (AC1) · 23 Sep 2019

Can we trust remote sensing ET products across Africa? Imeshi WEERASINGHEa,*, Ann van GRIENSVENab, Wim BASTIAANSSENb, Marloes MULb, Li JIAc celray.chawanda@vub.be

aVrije Universiteit Brussel (VUB) bIHE Delft Institute for Water Education cChinese Academy of Sciences dJoint Center for Global Change Studies

Overview We want to thank the reviewers for their dedication in reviewing the manuscript. We also are thankful for their thoughtful and constructive suggestions and comments. We have addressed all the comments raised by the reviewers and the manuscript has improved from the proposed changes.

[Figure]

Reviewer 1 Summary The manuscript has followed a commendable approach to eval-
uate eight diverse ET products and presented a ranking of the different products on
data sparse region. The method evaluated 8 products using a basin water balance
ET and Budyko curve over several basins across Africa using the average of three
precipitation products along with observed runoff data. Care was taken to ensure the
assumption of negligible storage change over several years by removing basins that
showed trends using the MK test. The manuscript is well-written with a useful appli-
cation and contribution to the remote sensing community. I have a few general and
specific comments that could improve the manuscript. Authors Response We thank
Reviewer 1 for his/her overall support for our study and the constructive suggestions
and feedback that he/she has given for the improvement of the manuscript. Below, we
address the issues that were raised for the improvement of the manuscript

Reviewer 1, General Comment

Considering Figure 6, 7 and 8 are key results for the ranking shown in Table 3, the
method needs to flesh out how the data points are generated. For example, in Figure
6 seems to show correlation (r) across basins using the mean value for RS and WB
ET. As indicated the correlation values are strong for all, but a root mean square error
(RMSE) may have been a more useful metrics to compare the different models as that
includes bias information. Also, it is not clear if the r difference between adjacent mod-
els is significantly different to rank them in a different order. I would think assigning a
different rank order when the "r" are not significantly different may inflate the order. But
the use of RMSE in the ranking may be more robust and it is not clear why this are
not used. Similarly, Figures 7 and 8 could benefit from statement that the table values
represent one data point for each basin and the average is the average of all basins, if
that is correct? But unless the values in Figures 7 and 8 are missing negative biases,
it is not clear how the average becomes so small when the percentage difference in
each basin is much higher, as much as 73%. The difference between Figures 7 and
8, i.e., average and weighted average is not clear. Are the weights (basin area) as-

signed only to the RS ET or to both RS and WB ET and in that case does this mean volumetric ET difference? Again, a more detailed description is required in the methods section. Landcover: it is not clear why the study did not include more land cover types, especially knowing the chosen two landcovers (water and irrigated lands) may not be handled well by some of the models . Authors Response 1. We have included a description how the data points are generated in the manuscript. (P5,L20), (P5,L26), (P6,L11) 2. We have also included both RMSE and basin area weighted RMSE along with correlation (r) in the study. We kept r in the study due to the correlation assessing solely the patterns of ET variability between basins rather than also the magnitude. There is not a significant difference between the correlations between the products. But what we do see here is that different products rank higher in this statistical criteria and feel it may be an interesting statistic for a reader depending on their study of interest. In terms of inflating the statistics, due to there being several statistical criteria included in the catchment water balance ranking, we feel this should not drastically change the results. (P6,L12) and Table 5. 3. We have included in the methodology the calculation steps for ET, including where basin areas where taken from and that the mean ET from the basin was recorded. Thss indicates one data point (mean) for each basin. (P6,L11) 4. The difference between average and weighted average is dependent on the area of each basin. This is now explained in the manuscript. It is not volumetric ET difference but difference in mm/year between calculated WB ET and ET estimated by products when considering the basin area weights. Therefore basins with larger areas have a stronger weight than basins with a smaller area as given by the used shapefile areas. (P5,L20), (P6,L12), (P6,L14) 5. Land cover: Since we have to use large land cover types in order to visually see difference at the African scale, we only used large irrigated areas and large water bodies. We have now also included large forested areas, the Congo forest, in our study and have not zoomed into a particular section of Africa but looked at it as a whole. (P8,L2)

Authors Changes in Manuscript

1. "Catchment or basin areas were taken from the 'Major River Basins of the World' (MRBW) shapefile. Discharge was converted from cubic meters per second to millimetres per year using the above mentioned catchment areas for all years of data availability for each basin." "Long-term ETWB was calculated by using the long-term average discharge and precipitation data for each catchment." "Basin average ETWB was calculated according to the MRBW shapefile boundaries and the basin mean was recorded." 2. "The Root Mean Square Error (RMSE), the basin area weighted RMSEaw, the correlation coefficient (r), bias and basin area weighted biasaw with calculated long-term average ETWB versus ETRS for all basins were found. "

3. "Basin average ETWB was calculated according to the MRBW shapefile boundaries and the basin mean was recorded." 4. "Catchment or basin areas were taken from the 'Major River Basins of the World' (MRBW) shapefile." "The Root Mean Square Error (RMSE), the basin area weighted RMSEaw, the correlation coefficient (r), bias and basin area weighted biasaw with calculated long-term average ETWB versus ETRS for all basins were found." "Basin area weighted bias and RMSE were found due to a large difference in basin areas. Therefore, basins with larger areas had more weight in the basin area weighted statistics than basin with smaller areas." 5. "Three types of land cover elements were evaluated in this study, irrigated areas, water bodies and forested areas."

Reviewer 1, Specific Comment 1

Tables and figures would need improved captions and header names that would help them stand alone. Authors Response Many of the section headers and figures in the paper have changed and the captions added or updated to reflect the content and to be able to stand alone. (Throughout the manuscript)

Previous headers and figure captions:

1. Introduction 2. Data 2.1. Remotely Sensed ET products 2.1.1. GLEAM 2.1.2. WaPOR 2.1.3. MOD16 2.1.4. SSEBop 2.1.5. WECANN 2.1.6. FLUXNET-MTE 2.1.7.

Fig 1: left) distribution of flux towers with LE data across Africa. (right) Number of years of available data at the six flux tower sites across Africa for both gap filled and bias corrected LE Fig 2: (left) All major basins in Africa and all discharge stations; (right) Major basins in Africa with available discharge data at outlet Fig 3: Budyko curve showing the energy limit and water limit Fig 4: (right) ET estimation for 28 major basins in Africa using P-Q (left) Final basins being analysed after trend analyses Fig 5: Dendogram after performing a cluster analysis showing the overall level of similarity between the RS products and MPM Fig 6: Correlation between long-term mean WB inferred ET and RS derived ET across basins using three different precipitation products (EWEMBI (left), CHIRPS (middle) and MSWEP (right)) Fig 7: Percentage difference between long-term mean WB inferred ET and RS derived ET across basins using three different precipitation products (EWEMBI (left), CHIRPS (middle) and MSWEP (right)) Fig 8: Weighted average (based on area) percentage difference between long-term mean WB inferred ET and RS derived ET across basins using three different precipitation products (EWEMBI (left), CHIRPS (middle) and MSWEP (right)) Fig 9: Comparison of RS products in representing irrigated areas. Zoomed to part of the Nile basin. Fig 10: Comparison of RS products in representing water bodies. Zoomed to part of the Nile basin. Fig 11: Average difference across long-term ET and PET estimates using (top) P-M (middle) P-T and (bottom) Hargreaves approaches for irrigated areas Fig 12: Average difference across long-term ET and PET estimates using (top) P-M (middle) P-T and (bottom) Hargreaves approaches for water bodies Fig 13: Evaluation of EWEMBI

WB and RS derived ET estimates using the Budyko curve with PET estimates from Hargreaves, PM and PT approaches. Figure (a) WECANN ET estimations (smallest difference with Budyko curve), Fig. (b) WB ET estimations and Fig. (c) GLEAM ET estimations (largest difference with Budyko curve) plotted on the Budyko curve. Fig 14: Evaluation of CHIRPS WB and RS derived ET estimates using the Budyko curve with PET estimates from Hargreaves, PM and PT approaches. Figure (a) WECANN ET estimations (smallest difference with Budyko curve), Fig. (b) WB ET estimations and Fig. (c) GLEAM ET estimations (largest difference with Budyko curve) plotted on the Budyko curve. Fig 15: Evaluation of MSWEP WB and RS derived ET estimates using the Budyko curve with PET estimates from Hargreaves, PM and PT approaches. Figure (a) WECANN ET estimations (smallest difference with Budyko curve), Fig. (b) WB ET estimations and Fig. (c) CMRSET ET estimations (largest difference with Budyko curve) plotted on the Budyko curve.

Authors Changes in Manuscript "... 7. Introduction 8. Data and Methods 8.1. Data 8.1.1. Evapotranspiration products 8.1.2. Precipitation products 8.1.3. Discharge data 8.1.4. Reference potential evapotranspiration products 8.2. Methods 8.2.1. Catchment water balance evapotranspiration (ETWB) 8.2.2. Evaluation using the Budyko curve 8.2.3. Spatial variability assessment 8.2.4. Assessment of similarity 9. Results 9.1. Catchment water balance 9.1.1. Comparison of precipitation and potential evapotranspiration products 9.1.2. Basins used in analyses 9.1.3. Catchment water balance comparison 9.2. Evaluation using the Budyko curve 9.3. Spatial variability assessment 9.4. Product similarity assessment 9.5. Ranking of products 10. Discussion 11. Conclusion ..."

Fig 1: "(left) distribution of flux towers worldwide. (right) distribution of flux towers across Africa" Fig 2: "(left) All major basins in Africa and all available discharge stations; (right) Major basins in Africa with available discharge data at outlet" Fig 3: "Budyko curve showing the energy limit and water limit for reference ET condition by partitioning precipitation into discharge and evapotranspiration" Fig 4: "Selected land cover

elements represented by the IFL, WB GRanD and AEIai maps with areas selected for visual assessment highlighted" Fig 5: "Comparison of the EWEMBI, MSWEP and CHIRPS precipitation products on their prediction of mean P across the basins" Fig 6: "Comparison of the P-M, P-T and Hargreaves potential evapotranspiration products on their prediction of mean PET across the basins" Fig 7: "(right) ETWB estimation for 28 major basins in Africa using P-Q (left) Final basins being analysed after analyses to discount basins with trends in ETWB, P and/or Q." Fig 8: "Percentage bias and basin area weighted percentage bias between the long-term annual average calculated ETWB and ETRS for all basins and the average of the 20 basins" Fig 9: "Evaluation of the calculated ETWB and ETRS from products using the Budyko curve calculated using average P and PET from three products" Fig 10: "Spatial assessment across Africa of each ET product based on selected land cover elements, forest, irrigated areas and water bodies" Fig 11: "Comparison of mean ET across the selected forested area for each product versus mean ET found from literature" Fig 12: "Comparison of the calculated kc for each product using average of the three PET products versus the average kc from maize, wheat and sugarcane from FAO." Fig 13: "Comparison of mean ET across water bodies estimated by each ET product and PET using the average of three PET products" Fig 14: "Cluster analysis based on the pairwise Euclidean distance between each pixel for each ET product to assess overall similarity between data sets"

Reviewer 1, Specific Comment 2 Figures 7 and 8 may benefit from one more panel which shows the average of the three precipitation products as the ranking is based on the average the three.

Authors Response We have taken out the statistics based on each different precipitation product and have used the average of the three products after a comparative analysis of the three precipitation products. In this way, instead of including a separate column with the average of the three products in figures 7 and 8, we have included one figure (figure 8) which shows the percentage bias and percentage basin area weighted

bias between calculated water balance ET and ET estimates from products across the basins. This includes positive and negative biases giving an idea of under or over estimation by each product across different basins or the average of all basins. (P5,L23) and Figure 8.

Authors Changes in Manuscript

"Since direct observations of precipitation from gauges were not used, precipitation was taken as the average of the three data products EWEMBI, CHIRPS and MSWEP."

Reviewer 1, Specific Comment 3

Zoom-in maps: it is hard to see the differences in Figures 9 and 10 among models. Maybe it is better to show deviations from the MPM data, i.e., show MPI in mm but the rest of the models as differences from MPM. Also remove the grid lines, hard to read the maps. A better color ramp will help readability. Authors Response We have not zoomed into a particular area but have looked at the land cover elements with respect to the entire continent. The colour ramp used is now clearer between the selected land cover elements and each element has been highlight with a box. The multi-product mean (MPM) was not used anymore as for the multi-product we used a different existing product. This was used instead of calculated the MPM from our products due to a comment by another review which asked to look at a weighting system for its calculation which was not in the scope of this study. (Figure 10) Authors Changes in Manuscript

Reviewer 1, Specific Comment 4

Revisit carefully the description and citation of some products. For example, SSEB vs SSEBop. As far as I know the global product is from SSEBop with a different citation with a 10-day (dekad) time scale, not monthly. Model's pre-defined boundary limits are described in SSEBop's work and not in the indicated citations. Authors Response We have carefully revisited the descriptions and citations of the products used in this study.

The SSEBop product as far as I have found in the USGS FEWS NET data portal the product is only available at monthly and yearly time scales when looking at the global scale and decadal for continental Africa. For this reason I used the monthly products. The citation was updated. (Table 1) Authors Changes in Manuscript

Reviewer 1, Specific Comment 5

It will be useful to include data source (website link) of the different models for access and discuss why the different models appear to discontinued Authors Response We have included an access link in Table 1. We have not discussed why different models have been discontinued but have mentioned which models are discontinued to take into consideration when selecting a product to use. (Table 1), (P20,L20) Authors Changes in Manuscript

Refer to Table 1 under Reviewer 1, Specific Comment 4

"LandFlux-EVAL and MTE also have early starting years however only go up to 2005 and 2012, respectively. ETMonitor is also no longer being extended and is not openly accessible or available for use."

Reviewer 1, Specific Comment 6

Include some discussion on the performance of MTP in relation to the WB ET (rank 5) and the value for MTP or ensemble products for future use.

Authors Response We have used an existing ensemble product, LandFlux-EVAL as the multi-product within study due to questions regarding the calculation of the MPM. This product was used also due to the initiative to create a benchmark product with the ET datasets using a range of different products for two long periods. We found this product ranked well even considering the coarseness in spatial resolution, which showed promise for ensemble products in the future. (P19,L29)

Authors Changes in Manuscript

"LandFlux-EVAL, with the coarsest spatial resolution, ranked fourth in the final ranking only outranked by the products with the three highest spatial resolutions in this study, CMRSET, SSEBop and WaPOR. Therefore, LandFlux-EVAL performs well overall regardless of its coarse resolution and is interesting due to being an ensemble product. Therefore, continuation or commencement of a similar initiative to develop a benchmark product using a range of ET data sets including high resolution products ranked within this study may improve the ensemble product for future use."

Reviewer 1, Specific Comment 7

Table 2: not clear what "not enough data" is referring to. Authors Response This means that from the calculated ETWB, there are less than 10 years of data available to calculate the MK test. This has been amended in the table to be clearer it is regarding the ET data points. (Table A1) Authors Changes in Manuscript

  Reviewer 2 Summary The article "Can we trust remote sensing ET products over Africa?" by Imeshi Weerasinghe et al. presents an evaluation analysis of the eight satellite - based evapotranspiration (ET) products over selected African river basins against the ET estimates derived from the water balance equation. The main conclusion of the study ranks the selected ET products in accordance with the results of the comparison analyses. The topicality and scientific relevance of the research question addressed in this study is high considering the sparseness of the in situ ET data in the region as well as the urgency of having a high quality ET estimates for the climate related problems in Africa. However, at this point I cannot recommend publication of this article as it (i) – contains a number of significant methodological inaccuracies and (ii) – gives poor explanation and presentation of the performed analyses and graphics. Also, stylistically and structurally the manuscript needs a substantial improvement. I highly recommend a major revision of the manuscript followed up by an internal review prior to the re-submission. Details follow. Authors Response We thank Reviewer 2 for his/her time spent on the review and constructive suggestions and feedback that he/she has given to improve the manuscript. Below, we address the issues that were

raised for the improvement of the manuscript.

Reviewer 2, General Comment

Generally, the presentation style of the paper makes it often hard to understand the correctness and hence the added value of the illustrated results. The lack of accompanying relevant information in a well-written form along with the multitude of presented data combinations in a variety of forms and at different scales in many cases confuses an understanding of (i) which data sets were used for this concrete calculation, (ii) in which form the data went into the following graphic, (iii) what the estimates were compared to, (iv) how many and which basins were used this time, (v) when data mean was used and over which scales the averaging was done? In my opinion, the paper did not succeed in wrapping up the results in a clear manner. Usage of multiple data levels, i.e. 3 reference rain datasets, 8 ET products, 27/20 basins with/without trends, different temporal resolutions (from one value to time-series), two spatial levels (from basin integral to pixel-basis) comprises a fairly large number of levels of information which the authors should unwrap and present in a very simple, consequent and logical manner. In the present version of the paper this have not been archived. The presentation style needs a thorough improvement, including restructuring of the manuscript, improvement of English grammar and scientific wiring style itself. One of the major remarks is that the whole manuscript text is written in a very intermittent and superficial manner. The explanations throughout the whole paper are significantly lacking concreteness. Also, confusion and replacement of some terms used throughout the study (e.g. trends as trends or trends as tendency to show certain value, among others) together with the multitude of abbreviations used in the text makes it very hard to follow the presentation (details given below). Finally, some data descriptions and methodological assumptions (in particular, downscaling to the smallest grid and usage of different time periods) raises a number of questions related to their correctness and validity. The details are given further below. Authors Response The paper has gone through a substantial restructuring in terms of the presented figures and different sections and sub-sections.

One of the main reasons behind this paper was to use different products with varying temporal and spatial scales for evaluation due to the fact that these products are a sample of the products available for use. In terms of which data sets were used for the concrete calculation, this was mentioned in the introduction and methodology sections. The 9 products being evaluated have all been used in the calculation. They have been resampled to the same spatial and temporal resolution. The average of the three precipitation products have been used and the average of the three PET products have been used. The discharge data has been obtained from GRDC and HYDR and have been converted to yearly averages. For each graphic and section, the methodology has been described. Each section described clearly what the ET estimates are being compared to both in the methodology section, in the results section and in the figure captions. According to how many basins are being used in the study. This is clearly identified in the results section of the paper including with graphics. The data mean for basins and land cover elements was used and stated. Long-term averaging was found after finding yearly averages for the different basins and land cover elements. The manuscript is written by an English native speaker and has been checked by a native English professor for structure and grammar. More concrete explanations and statements have been corrected to be clearer. The confusion between certain terms in their use in this paper such as trends and tendencies has been corrected. All abbreviations have been opened and no abbreviation has been used without explanation. As to the questions raised due to downscaling of products, these are addressed further into the specific questions raised. In short, this was done as to not lose any information from the high spatial resolution products at the same time not losing information from the coarser scale products. It would have been simpler to resample all products to the coarsest resolution in terms of computational efficiency and storage space. However, this was not done in order to ensure all features of ET products such as, spatial resolution, were evaluated according to the individual products as would not be the case if resampled from high to coarse. The paper addresses the use of different time periods as to the reasons why they are need and the possible reasons why they can be used.

Authors Changes in Manuscript

The entire manuscript should be shown here as most parts have been adapted and changed. Therefore the comments to the changes in the manuscript to the specific comments have been shown.

Reviewer 2, Specific Comment 1 – Scientific Relevance From the abstract, introduction and methodology sections it remains unclear how new is the water balance (WB) method, how accurate is the method and which other studies already used it for similar tasks. The abstract even makes a false impression that the authors developed a method and not used the existing one (p1, L6). The introduction in turn makes an impression that the focus of the study is the methodology (p3, L9) and not the evaluation of the existing products. In this view, the authors must provide an overarching literature review of the studies that already used the WB ET estimates for satellite products evaluation, and also studies which evaluated the same satellite products over Africa using the same or other techniques. One of such study examples is the Miralles et al. (2016), which is also referenced in the present manuscript. Note, that Miralles' study also involved the African river basins. This has to be explicitly mentioned in the introduction. The authors should also then place their results into the findings of others. This is not done at the present state of the paper. Authors Response The intention was not to give the impression that the catchment water balance methodology was developed within the scope of this study. The paper has been amended so that this false impression is not given. The introduction has been changed to show focus on the evaluation of the products and not the methodology. A literature review of similar comparisons using some products under evaluation in this study among others has been conducted. (P1,L7), (P4,L4), (P4,L10), (P3,L5), (P19,L19) Authors Changes in Manuscript

[revised manuscript text omitted]

Reviewer 2, Specific Comment 2 – Scientific Relevance The authors should be more careful in formulating their scientific conclusions. The following sentence in the abstract: "However our recommendation would be the three highest ranked products being CMRSET, SSEBop and WaPOR." sounds rather subjective and needs further motivation (especially considering the huge differences in spatial-temporal scales between the products, as well as the manipulations on interpolation, then vice versa - integration - done within the study). Why first three? The same remark applies for the conclusions. Authors Response We have changed subjective recommendations within the study to be more conclusive based on the findings. (P1,L18), (P20,L23) Authors Changes in Manuscript

Abstract "Based on the evaluation criteria in this study the three highest ranked products, CMRSET, SSEBop and WaPOR would suit many of the needs of readers due to low biases and good spatial variability."

[revised manuscript text omitted]

Reviewer 2, Specific Comment 2 – Data Products Many product descriptions and references miss version numbers. Those must be included, since depending on the product version there might be some already known issues related to a parameter derivations. Authors Response We have now included version numbers of the products in Table 1. (Table 1) Authors Changes in Manuscript

Refer to Table 1 under Reviewer 2, Specific Comment 1 – Data Products

Reviewer 2, Specific Comment 3 – Data Products Check carefully the correct citations, once you add the product version numbers. I am more familiar with the GLEAM product, and I know that for GLEAM v3 (if you used that version) the correct references are Miralles 2011 (HESS) and Martens 2016 (GMD). Authors Response We have carefully checked the references and have included and amended these in Table 1. (Table 1) Authors Changes in Manuscript

Refer to Table 1 under Reviewer 2, Specific Comment 1 – Data Products

Reviewer 2, Specific Comment 4 – Data Products It is also a rule of a good scientific practice to provide/cite the data source: a web-page, ftp or a personal communication. No data sources are mentioned in the current manuscript version. For GLEAM, for example it should be the web portal: www.gleam.eu; For MSWEP: http://www.gloh2o.org/ (?), etc. Authors Response We have now also added data sources to the manuscript in Table 1. (Table 1) Authors Changes in Manuscript

Refer to Table 1 under Reviewer 2, Specific Comment 1 – Data Products

Reviewer 2, Specific Comment 5 – Data Products One of my major remarks here con-

cerns inaccurate or sometimes even false information in data set descriptions. That is unacceptable. Please, check carefully all product descriptions you are giving! On the example of GLEAM: - GLEAM is not a physically-based model, Prietsley-Taylor, the interception loss model, the stress module, and the water-balance model in GLEAM which form the core of GLEAM are all empirical! One can call it a process-based model, as it empirically describes the process needed to estimate E from satellites; - Table 1: GLEAM does not use CMORPH at all! That is simply wrong information. - Alemohammad et al., 2017 is the reference to a paper where they describe another method of deriving E. It is not clear why this is included in this GLEAM section?. Authors Response We apologise for the incorrect description of some of the models and have made sure to change or take out the descriptions. The incorrect information has been deleted with the reconstruction of the data section. This should be reflected in Table 1. (Table 1) Authors Changes in Manuscript

Refer to Table 1 under Reviewer 2, Specific Comment 1 – Data Products

Reviewer 2, Specific Comment 1 – Methodology and results As it was already mentioned earlier, the presentation of the calculation steps is done in a rather superficial manner. Lots of information is not given or remains unclear. E.g.: - which concrete quality control steps were involved in the selection of the basins and, which additional analyses were done and by whom? (e.g. p8, L7-10) - how were the basin boundaries defined? - what is the time-period of available discharge data? - how the integration over the basins is exactly done? were the simple mean or the areaweighted mean of ET or P fields used when averaging over the basin area? - which manipulations were done with the precipitation data prior averaging it over the basins? Were the data also re-scaled to the 0.0022 deg resolution and then averaged over the basins? Never mentioned. - In their paper Miralles et al., 2016 applied additional quality control check on the difference between the GRDC-reported area and the area calculated from basin boundaries. Would not it be also relevant for the present study? - How was the MPM calculated? The products have big differences in resolution. The averaging of the

products to get the MPM without applying corresponding weights can be a source of errors. Authors Response We mention in the paper that we select the basins considering the availability of the discharge data at the outlet of a particular basin. This was how the initial 27 basins were selected. This was mentioned in the previous version and has not changed in this current version of the manuscript. A complete restructure of the methodology section has been done in order to incorporate the suggestions from this particular comment. Specifically, we now mention how the basin boundaries are defined (from the 'Major River Basins of the World' (MRBW) shapefile (World Bank 2017)). The available time period of the discharge data has been added and can be found in Table A1 in Appendix A. Integration over the basin has been done according to the MRBW shapefile boundaries for each basin with the mean of each basin within the shapefile recorded. This description has been added to the manuscript. The basin area weighted mean for averaging over a basin was only used in the statistics for RMSE and bias, otherwise the simple mean was used as stated in the manuscript previously and in this new version. No manipulations were done to the precipitation data prior to averaging over the basins except for obtaining yearly averages. This is now mentioned. Yes the data was rescaled to 0.0022 deg resolution and then averaged over the basins. This is now mentioned. We have also conducted an analysis on the difference between the MRBW shapefile areas taken for the basins and the area reported by the GRDC and mentioned only the potential problematic cases. We have now not used the MPM but an existing benchmark ensemble product, LandFlux-EVAL. (P5,L20), (Table A1), (P5,L24), (P5,L1), (P4,L26), Authors Changes in Manuscript

"Catchment or basin areas were taken from the 'Major River Basins of the World' (MRBW) shapefile (World Bank 2017). Discharge was converted from cubic meters per second to millimetres per year using the above mentioned catchment areas for all years of data availability for each basin."

"Basin average precipitation was calculated for the years 1979-2016 according to the MRBW shapefile boundaries recording the basin mean."

"Precipitation products were averaged at yearly temporal resolution for the purposes of this study."

"All products have been projected and gridded on a 0.0022 deg resolution geographic grid and averaged at yearly temporal resolution for the purposes of this study."

"The MRBW shapefile area did not differ greatly with the drainage area reported by the GRDC except in two cases. Here we found the ETWB calculated using the two areas only differed by 2.5 percent and 3.3 percent and thus kept these basins in the analyses."

Reviewer 2, Specific Comment 2 – Methodology and results (i) - It remains not very clear from the text over which values the correlation analysis is performed? Over time-series of annual means? Over multi-year averages of different basins? Should be put more clear. (ii) - Units of the correlations are not common, and confuse the interpretation. Correlation should rank from -1 to 1. Besides, it is never clear from all the graphs with percentages, by what value the normalisation was done. (iii) – Correlations should always provide significance measure, or the latter should be mentioned in the text.

Authors Response (i) - The correlation was performed over multi-year averages (long-term averages) across all basins under evaluation (20 basins). This has been mentioned in the manuscript. (P6,L16) (ii) – Correlations have been changed to -1 to 1 without units. The normalisation for each basin was conducted based on the calculated ETWB for each basin. These have been adapted in the manuscript. (Table 5), (P11,L15) (iii) – This has been adapted in the manuscript to reflect the significant measure. (P11,L11)

Authors Changes in Manuscript

(i) - "Correlations were calculated based on long-term averages across all basins."

(ii) -

"Percentage biases were normalised based on the calculated ETWB for each basin."

[Figure]

(iii) – "There is a significant positive correlation for all products ranging from 0.89-0.97 with GLEAM and LandFlux-EVAL showing the strongest relationships with ETWB"

Reviewer 2, Specific Comment 3 – Methodology and results The choice of the highest resolution is one of the two major remarks that I have to the methodological part: (i) – Generally, it is not common to interpolate products to the highest resolution, especially when the difference between the highest and the coarsest resolution is that high. It would be more correct to upscale the higher resolved data to the coarser estimates to minimise the bias. (ii) – Besides, the fact that the comparison of the products is mostly done at the basin level, the downscaling does not seem to make sense at all. First you interpolate the coarse data to the very high resolution, and then, you integrate it back again over the river basin. This clearly can be a source of additional biases and errors, which also raises my doubts about the validity of the ranking results. (iii) – All the above inter alia also raises a question of what is the minimal area of the smallest basin you have, and whether it is resolved by the products with the coarsest resolution at all?.

Authors Response (i) - We did not upscale the higher resolution products to coarser resolution products as our aim was not to minimise bias between products but to evaluate each product independently of the other according to their features (e.g. spatial resolution) regardless of whether they were advantageous or disadvantageous. I believe if we were trying to find the effects of ET calculation methods or forcing data and looking at the different products for comparison this would make sense that we would need to minimise other biases between the different products. However, as the goal of our manuscript is to evaluate different offerings of ET products regardless of temporal and spatial resolutions or coverage, forcing data, ET calculation method, etc, it does not make sense for us to minimise bias between products. We want to calculate the bias of each product with respect to a reference (ETWB). In this respect we do not lose any information by downscaling from coarse to high resolution and in fact we found that for basin boundaries to be as close to the shapefiles as possible, the data set of GLEAM, for example, showed very little difference (approximately 0-5mm/year) in the estimation of mean ET for different basins when using the coarse resolution or the resampled high resolution dataset. Whereas a slightly larger difference range was found for the CMRSET product (0-50mm/year) when comparing the high resolution product to the resampled coarse resolution for a sample of basins. In this regards we did not believe that there was a disadvantage by resampling to the highest resolution of the products. Also, since we were trying compare individual products on their own merits of each feature, this enable a more accurate comparison without minimising bias between the products. (ii) – As with point (i) above, we did not find that there were any additional biases and errors from this method of evaluation and found very small differences when looking at the two different methods of up or downscaling resolutions. In fact, the biases were greater when resolving from high resolution to coarse resolution and the biases slightly smaller when resolving from low resolution to high resolution when looking at certain basin means according to the reference used (ETWB). Although since very small differences were seen, they were almost negligible. Therefore the validity of the ranking results still hold. (ii) – The smallest basin is >30000km2 which for certain products such as LandFlux-EVAL and WECANN, this would not be fully resolved. However the intent was to also include basins that were smaller in our analysis to: 1. Have a good spatial coverage across Africa and 2. Have a range of basin sizes to evaluate the products on. Even though we do not see any spatial variability in products such as LandFlux-EVAL and WECANN in the smaller basins. Their prediction of long-term average ET for smaller basins, especially LandFlux-EVAL showed lower biases than higher resolution products such as ETMonitor. Authors Changes in Manuscript

For the above reasons, nothing with regard to this point was changed within the manuscript.

Reviewer 2, Specific Comment 4 – Methodology and results The second and the most major remark of mine is related to the application of the analyses at different time-periods. (i) - First of all it never comes clear what is the time period of available discharge data for every basin; (ii) – From the Tabel 2 it appears that for most of the basins

discharge data does not extend the whole period of available precipitation data at all, and the spread of data periods is huge among the basins. In this view I do not understand at all how the analyses tests were done? (ii) – The test for the effect of temporal variability on annual means mentioned in the discussion section was done only for the four basins, while 20 basins are analyzed throughout the study. Moreover, Congo - one of the four tested basins - has data only till 2010, while remote sensing ET products span up to 2017. In this view, I would not be able to call it a fair validity test! (iii) – Clearly, the exclusion of periods with trends does not account for the temporal variability of data which can still result in the pretty different annual means. So, the effect of temporal variability on annual means must be done for all the basins, which are used for the evaluation of the satellite ET products in order to draw a fair conclusions. (iv) – Calculating trends only for the WB ET reference data set is not a complete analysis. If a satellite data product has a trend, this also has to be mentioned, and maybe even that product should not participate in the validation (?) To conclude, if the tests will show that the variability indeed matters, then none of the performed analyses is valid since they will all be affected by the differences due to variability.

Authors Response (i) – The period of available discharge data for each of the 27 basins where ETWB was initially calculated for has been added and can be found in Table A1 within the manuscript. (ii) – Here we used the long-term average precipitation data minus the long-term average discharge data to calculate the ET. This is mentioned in the manuscript. We also calculated the ET based on the average precipitation and discharge based on the overlapping periods and found a maximum of 5% difference in ET from both methods. In most cases 0% difference was found. (P5,L26) (ii) – the test for temporal variability was conducted on the selected four basins as they were the only basins with long enough time periods to conduct this test. We used these as samples to show that in all cases tested the difference in calculated ET was minimal. Congo was also used, although it only had data until 2010 so that we could test more than just three basins. The corresponding periods were used in the remote sensing products so we made sure the periods tested overlapped. In this regard we believe this was a

fair validity test. (iii) – We agree that if possible the test for temporal variability should have been conducted on all basins, but we are unable to do this based on the available time periods of the data. Therefore we took the four basins as a sample and surmised that due to finding minimal differences in calculated ET, we could compare long-term averages from different time periods. (iv) – Considering the point is evaluating the prediction of ET estimates by different ET products, whether they do or do not have trends is not a basis for inclusion or not in this analysis. It would be interesting to find out whether the different products do show trends for particular basins but this was not in the scope of this study. Our goal was to evaluate based on a reference which we needed to calculate in an accurate manner, which would mean no trends when looking at long-term averages of different time periods. Therefore our 20 basins under analysis without trends in their calculated long-term ETWB was used as a reference to compare with ET product estimations. If an ET product had a trend in their estimation of ETWB for one of the 20 basins under analysis, then this would most likely have a higher bias when compared with the reference ETWB. It was not relevant for us to see if ET products had trends or not, only to evaluate their ET estimates. Authors Changes in Manuscript

"Long-term ETWB was calculated by using the long-term average discharge and precipitation data for each catchment."

Nothing else for this point was changed in the manuscript.

Reviewer 2, Specific Comment 5 – Methodology and results My advice would be to not use percentages for all the figure results. This only confuses the interpretation. Use -1 to 1 scale for correlation and differences.

Authors Response We did not use percentages for the overall statistics in Table 5. However, we still used percentages when looking at bias and basin area weighted bias as subjectively we found these results easier to interpret and was given this advice from different advisers of the manuscript. (Table 5) Authors Changes in Manuscript

Refer to Table 5 under Reviewer 2, Specific Comment 2 – Methodology and results

Reviewer 2, Specific Comment 6 – Methodology and results The raking of products based on visual inspection is rather speculative for me. For ex on Fig 9 it was impossible to follow the text conclusions: I did not see where irrigation C7 HESSD Interactive comment Printer-friendly version Discussion paper area is, to which reference product other products are compared, why MPM and some other products have no data and why GLEAM is concluded to perform worst? The same for Fig 10..

Authors Response We re-did this section to include forest as an additional land cover type for inspection as well as using highlighted areas across the entire African continent rather than zooming into selected areas shown in figure 5. We agree that this is a rather subjective method, however, we also believe it is relatively visible to see the difference between the products especially when using different scales to be able to visually interpret the results. We believe these maps are an added advantage for reader interested in spatial characteristics of ET. However, due to its potential subjectivity in ranking, we do two final rankings, with and without visual inspection to minimise this subjectivity. (P13,L4), (Figure 10) Authors Changes in Manuscript "Figure 10 shows ET across Africa for all ET products with the specific land cover elements (forest, irrigated areas and water bodies) highlighted. Two different scales are used in order to be able to visually compare the products according to spatial variability rather than magnitude of ET. For products where large biases were found, a scale of 0-1200 mm/year was used and for the remaining products a scale of 0-1800 mm/year was used. Visually, all products capture the forested area. Irrigated areas are also captured well by most products. GLEAM and LandFlux-EVAL do not capture the majority of selected irrigated areas. CMRSET, ETMonitor, SSEBop and WaPOR capture most of the selected irrigated areas while the remaining products capture a few. GLEAM, LandFlux-EVAL, MOD16, MTE and WECANN only estimate land ET and thus do not have ET across water bodies. The remaining products capture the water bodies well, with CMRSET and ETMonitor showing larger differences in their estimations of ET across water bodies than the surrounding areas over SSEBop and WaPOR."

Reviewer 2, Specific Comment 7 – Methodology and results The analyses of comparing products over one irrigation area, and one lake, where some products have no data, and others do not even resolve the region does not make much sense to me, nor it is complete enough to make a serious conclusion on which product is better or worst.

Authors Response Please refer to answer from section Reviewer 2, Specific Comment 6 – Methodology and results. Reviewer 2, Specific Comment 8 – Methodology and results It maybe due to the presentation style, but I could not follow the result section presenting the crop coefficients very well. It has to be structured and presented in a more clear manner: objective and reasoning for location, crop types, etc, data used, hypothesis to prove, which products in which form are tested, and what do results show.

Authors Response This section has been re-written to try and present this concept in a more structured and clear manner in the manuscript. (P8,L11)

Authors Changes in Manuscript

"For irrigated areas, the crop coefficient (kc) was used. The crop coefficient is a property of a plant that aids in determining ET and can be calculated using equation 3.

$$kc = ET/PET$$

Where kc is the coefficient for crops growing under conditions of optimum fertility and soil moisture and achieving full production potential (Allen et al. 1998). In reality optimal conditions are rarely met, however this measure was used to evaluate how well the ET products determined ET across irrigated areas. Average crop coefficients for maize, wheat and sugarcane estimated by FAO were used as a reference. The long-term annual average mean ET estimates across irrigated areas were divided by the long-term annual average mean PET estimates across irrigated areas to find the average crop coefficient (kc) across irrigated areas. These estimates were found by looking at the mean of the area according to the AEIai shapefile. The bias between the reference

kc from FAO and estimated kc using individual ET product estimates and PET derived using the mean of the three PET products was recorded."

Reviewer 2, Specific Comment – Stylistic and structural I highly recommend to look into the papers of Zeng et al., 2012 (ERL) and Miralles et al., 2016 (HESS) as an example of a good presentation style, and especially of the methodological part, as well as their choice of graphs.

Authors Response The mentioned papers and others as well as internal discussion was conducted and the paper revised along with the figures and graphs. There is a substantial change in the entire manuscript as well as most figures and graphs to be more clear and structured.

Authors Changes in Manuscript Changes are found throughout the entire manuscript.

Reviewer 2, Specific Comment 1 – Stylistic and structural Usage of term throughout the paper changes and confuses the reader. For ex, the term trend is first used as trend itself, but also to indicate tendencies if I understood it correctly (p13, L5 or p16,L9). Authors Response The term trend has been looked into and changed according to trend and tendency. As have other terms which we found to be confusing or arbitrary. (P12,L2), (P19,L19) Authors Changes in Manuscript "The calculated ET for most of the ET products and also for the majority of basins falls under the curve showing a tendency for products to underestimate basin ET." "In terms of consistency in results with previous studies conducted on some of the products under evaluation we see similar tendencies." Reviewer 2, Specific Comment 2 – Stylistic and structural Typos are also present throughout the paper (e.g. p3 L2, p7 L17, p15 L7, p20 L3). A proper internal resew is required.

Authors Response Any typos we have found have been corrected in the manuscript. A spell check and internal review has been conducted again with the new version of this manuscript. Reviewer 2, Specific Comment 3 – Stylistic and structural Discussion section rather reads like methods and should be incorporated to methods. Instead, the

discussion section should place the paper findings into the existing knowledge as was already mentioned earlier.

Authors Response We agree that the discussion should place the paper findings within existing knowledge and have added this to the manuscript. However, we also feel that the discussion section discusses certain assumptions and findings that requires further explanations. We found this style quite useful and interesting to read and did not incorporate these findings into the methods section. (P19,L19) Authors Changes in Manuscript "In terms of consistency in results with previous studies conducted on some of the products under evaluation we see similar tendencies. According to (Miralles et al. 2016) GLEAM, MOD16 and other products in their study show divergences in conditions of water stress and drought. Considering large parts of Africa are potentially under water stress due to the semi-arid and arid climate (IPCC 2019; World Bank 2018), this can explain the low ranking of GLEAM and MOD16 in this study. The RMSE and biases found in our study for Africa are comparable with those found by (Vinukollu, Wood, et al. 2011) at the global scale, however comparing different products to that of this study. The range is higher in this study for Africa than the range found at the global scale. In their study, (Trambauer et al. 2014) found GLEAM to underestimate ET in terms of their multi-product mean. This is again consistent with our finding where biases in GLEAM showed large underestimations across the basins in Africa with respected to the calculated ETWB. We used the LandFlux-EVAL benchmark product as an ensemble product without calculating the multi-product mean of the products being used in this study, as it was developed using a large range of ET products. LandFlux-EVAL, with the coarsest spatial resolution, ranked fourth in the final ranking only outranked by the products with the three highest spatial resolutions in this study, CMRSET, SSEBop and WaPOR. Therefore, LandFlux-EVAL performs well overall regardless of its coarse resolution and is interesting due to being an ensemble product. Therefore, continuation or commencement of a similar initiative to develop a benchmark product using a range of ET data sets including high resolution products ranked within this study may improve the ensemble product for future use."

Reviewer 2, Specific Comment 4 – Stylistic and structural Nothing is mentioned about the Budyko result in Discussion or Conclusions.

Authors Response We have now included conclusions regarding the Budyko analyses in the conclusion section. (P20,L7) Authors Changes in Manuscript "According to the comparison of the ETWB with ETBudyko, we see that ETWB follows the Budyko curve and has an overall low bias across the basins. This indicates the calculated ETWB is a sound reference condition to use for analyses."

Reviewer 2, Specific Comment 5 – Stylistic and structural Nothing is mentioned about the differences between using three precipitation products.

Authors Response We now use the average of the three precipitation products and not evaluations based on the individual products and thus do not mention this. Reviewer 2, Specific Comment 6 – Stylistic and structural Many abbreviations are not opened. Add abbreviation table to the paper. Use less abbreviations, i.e. if possible leave it open. Very hard to follow.

Authors Response We have now opened all abbreviations but have not added an abbreviation table to the paper as all abbreviations are opened. Reviewer 2, Specific Comment 7 – Stylistic and structural Explanation of the results in Figures are often not complete. Not clear which products were used, which reference, etc. Figures are too small.. Authors Response Figures have mostly all been changed being high resolution and easier to read. Explanation of results of these figures tries to be more complete. Authors Changes in Manuscript Changes throughout the entirety of the manuscript Reviewer 2, Specific Comment 8 – Stylistic and structural Lots of sentences are too wake, "Based on the elements being analysed. . . p21, L30" What is meant? Be more concrete. Authors Response We have tried to be stronger in our sentences. Instead of using 'based on the elements being analysed' we have used 'based on the selected land cover elements being analysed' for example. Authors Changes in Manuscript Changes throughout the entirety of the manuscript Reviewer 2, Specific

Comment 9 – Stylistic and structural Titles of the sections should be reconsidered. Authors Response Titles of the sections have been reconsidered and changed. Authors Changes in Manuscript "... 1. Introduction 2. Data and Methods 2.1. Data 2.1.1. Evapotranspiration products 2.1.2. Precipitation products 2.1.3. Discharge data 2.1.4. Reference potential evapotranspiration products 2.2. Methods 2.2.1. Catchment water balance evapotranspiration (ETWB) 2.2.2. Evaluation using the Budyko curve 2.2.3. Spatial variability assessment 2.2.4. Assessment of similarity 3. Results 3.1. Catchment water balance 3.1.1. Comparison of precipitaton and potential evapotranspiration products 3.1.2. Basins used in analyses 3.1.3. Catchment water balance comparison 3.2. Evaluation using the Budyko curve 3.3. Spatial variability assessment 3.4. Product similarity assessment 3.5. Ranking of products 4. Discussion 5. Conclusion ..."

Allen, Richard G, Luis S Pereira, Dirk Raes, Martin Smith, and W Ab. 1998. "Guidelines for Computing Crip Water Requeriments-FAO Irrigation and Drainage Paper 56." Crop Evapotranspiration. Rome. doi:10.1016/j.eja.2010.12.001. Balsamo, G, C Albergel, A Beljaars, S Boussetta, E Brun, H Cloke, D Dee, et al. 2015. "ERA-Interim/Land: A Global Land Surface Reanalysis Data Set." Hydrol. Earth Syst. Sci 19: 389–407. doi:10.5194/hess-19-389-2015. Beek, L.P.H. (Rens) Van, and Marc F P Bierkens. 2009. "The Global Hydrological Model PCR-GLOBWB: Conceptualization, Parameterization and Verification Report." Utrecht, Netherlands. http://vanbeek.geo.uu.nl/suppinfo/vanbeekbierkens2009.pdf. Dee, D. P., S. M. Uppala, A. J. Simmons, P. Berrisford, P. Poli, S. Kobayashi, U. Andrae, et al. 2011. "The ERA-Interim Reanalysis: Configuration and Performance Ofthe Data Assimilation System." Royal Meteorology Society 137: 553–97. Fisher, Joshua B., Kevin P. Tu, and Dennis D. Baldocchi. 2008. "Global Estimates of the Land–atmosphere Water Flux Based on Monthly AVHRR and ISLSCP-II Data, Validated at 16 FLUXNET Sites." Remote Sensing of Environment 112 (3): 901–19. doi:10.1016/j.rse.2007.06.025. IPCC, Intergovernmental Panel on Climate Change. 2019. "IPCC Special Report on Climate Change, Desertification, Land Degradation, Sustainable Land Management, Food Security." https://www.ipcc.ch/site/assets/uploads/2019/08/Fullreport-1.pdf. Jiménez, C.,

C. Prigent, B. Mueller, S. I. Seneviratne, M. F. McCabe, E. F. Wood, W. B. Rossow, et al. 2011. "Global Intercomparison of 12 Land Surface Heat Flux Estimates." Journal of Geophysical Research Atmospheres 116 (2): 1–27. doi:10.1029/2010JD014545. Martens, Brecht, Diego G Miralles, Hans Lievens, Robin Van Der Schalie, Richard A M De Jeu, Diego Fernández-Prieto, Hylke E Beck, Wouter A Dorigo, and Niko E C Verhoest. 2017. "GLEAM v3: Satellite-Based Land Evaporation and Root-Zone Soil Moisture." Geosci. Model Dev 10: 1903–1925. doi:10.5194/gmd-10-1903-2017. Miralles, D G, C Jiménez, M Jung, D Michel, A Ershadi, M F Mccabe, M Hirschi, et al. 2016. "The WACMOS-ET Project-Part 2: Evaluation of Global Terrestrial Evaporation Data Sets." Hydrol. Earth Syst. Sci 20: 823–42. doi:10.5194/hess-20-823-2016. Montieth, L. J. 1965. "State and Movement of Water in Living Organisms." In 19th Symposium of Evaporation and the Environment. Swansea, London: Cambridge University Press. Mu, Q, F A Heinsch, M Zhao, and S W Running. 2011. "Improvements to a MODIS Global Terrestrial Evapotranspiration Algorith." Remote Sensing of Environment 115: 1781–1800. doi:10.1016/j.rse.2007.04.015. Mu, Qiaozhen, Faith Ann Heinsch, Maosheng Zhao, and Steven W Running. 2007. "Development of a Global Evapotranspiration Algorithm Based on MODIS and Global Meteorology Data." Remote Sensing of Environment 111: 519–36. doi:10.1016/j.rse.2006.07.007. Mueller, B., S. I. Seneviratne, C. Jimenez, T. Corti, M. Hirschi, G. Balsamo, P. Ciais, et al. 2011. "Evaluation of Global Observations-Based Evapotranspiration Datasets and IPCC AR4 Simulations." Geophysical Research Letters 38 (6): 1–7. doi:10.1029/2010GL046230. Mueller, B, M Hirschi, C Jimenez, P Ciais, P A Dirmeyer, A J Dolman, J B Fisher, et al. 2013. "Benchmark Products for Land Evapotranspiration: LandFlux-EVAL Multi-Data Set Synthesis." Hydrol. Earth Syst. Sci 17: 3707–20. doi:10.5194/hess-17-3707-2013. Penman, H L. 1948. "Natural Evaporation from Open Water, Bare Soil and Grass." Source: Proceedings of the Royal Society of London. Series A, Mathematical and Physical Sciences. Vol. 193. Priestley, B., and R. Taylor. 1972. "On the Assessment of Surface Heat Flux and Exporation Using Large-Scale Parameters." Monthly Weather Review 100 (2): 81–92. Su, Z. 2002. "The Surface Energy Balance System (SEBS)

for Estimation of Turbulent Heat Fluxes." Hydrology and Earth System Sciences. Vol. 6. https://www.hydrol-earth-syst-sci.net/6/85/2002/hess-6-85-2002.pdf. Trambauer, P, E Dutra, S Maskey, M Werner, F Pappenberger, L P H Van Beek, and S Uhlenbrook. 2014. "Comparison of Different Evaporation Estimates over the African Continent." Hydrol. Earth Syst. Sci 18: 193–212. doi:10.5194/hess-18-193-2014. Vinukollu, Raghuveer K., Remi Meynadier, Justin Sheffield, and Eric F. Wood. 2011. "Multi-Model, Multi-Sensor Estimates of Global Evapotranspiration: Climatology, Uncertainties and Trends." Hydrological Processes 25 (26): 3993–4010. doi:10.1002/hyp.8393. Vinukollu, Raghuveer K, Eric F Wood, Craig R Ferguson, and Joshua B Fisher. 2011. "Global Estimates of Evapotranspiration for Climate Studies Using Multi-Sensor Remote Sensing Data: Evaluation of Three Process-Based Approaches." doi:10.1016/j.rse.2010.11.006. World Bank. 2017. "Major River Basins Of The World." World Bank Data Catalogue. https://datacatalog.worldbank.org/dataset/major-river-basins-world. World Bank, 2018. Beyond Scarcity Water Security in the Middle East and North Africa. Edited by World Bank. MENA Development Report: Washington DC: MENA Development Report: doi:10.1192/bjp.111.479.1009-a.

**Table 5.** Calculated statistics, bias, $bias_{aw}$, RMSE, $RMSE_{aw}$ and r, for the comparison of the long-term annual average $ET_{WB}$ versus $ET_{RS}$

|  | CMRSET | ETMonitor | GLEAM | LandFlux-EVAL | MOD16 | MTE | SSEBop | WaPOR | WECANN |
|---|---|---|---|---|---|---|---|---|---|
| bias | -19 | 156 | 254 | 115 | 131 | 146 | 12 | -3 | 139 |
| $bias_{aw}$ | -18 | 237 | 313 | 148 | 266 | 183 | 30 | -46 | 223 |
| RMSE | 113 | 211 | 273 | 152 | 199 | 184 | 163 | 104 | 189 |
| $RMSE_{aw}$ | 187 | 502 | 594 | 304 | 590 | 424 | 123 | 165 | 520 |
| r | 0.94 | 0.91 | 0.97 | 0.97 | 0.91 | 0.95 | 0.89 | 0.96 | 0.95 |

**Fig. 1.** Calculated statistics, bias, bias_aw, RMSE, RMSE_aw and r for the comparison of the long-term annual average ET_WB versus ET_RS

**Bias**

| | CMRSET | ETMonitor | GLEAM | LandFlux-EVAL | MOD16 | MTE | SSEBop | WaPOR | WECANN |
|---|---|---|---|---|---|---|---|---|---|
| Awash | -44.6 | 46.8 | 39.6 | 11.8 | 48.4 | 39.1 | 34.4 | 14.3 | 11.8 |
| Bandama | -7.2 | -1.7 | 24.0 | 6.6 | -11.9 | 7.1 | -16.4 | -17.2 | 7.2 |
| Buzi | -11.7 | 16.6 | 25.9 | 14.7 | -2.6 | 4.6 | 1.5 | -9.3 | 16.4 |
| Cavally | 11.5 | 19.1 | 32.8 | 22.0 | 10.0 | 20.6 | 6.3 | 13.4 | 30.5 |
| Congo | -4.1 | 18.3 | 16.2 | 14.0 | 22.2 | 17.0 | 1.4 | -5.1 | 20.2 |
| Gambia | 15.5 | 13.4 | 34.2 | 22.5 | 38.9 | 20.5 | -22.6 | -0.4 | 20.9 |
| Groot | -49.4 | 68.9 | 45.4 | -10.1 | 33.7 | 31.1 | 15.4 | 24.2 | -12.5 |
| Komoe | -1.7 | -5.5 | 22.4 | 7.8 | -0.5 | 6.8 | -20.9 | -23.2 | 7.5 |
| Maputo | -8.6 | -4.8 | 12.6 | 0.1 | 5.6 | 6.4 | 9.3 | -1.3 | 1.8 |
| Mono | -2.5 | -2.0 | 23.1 | 5.7 | -10.1 | 7.8 | -14.0 | -7.1 | 10.8 |
| Niger | 19.6 | 22.4 | 42.4 | 16.2 | 16.2 | 17.9 | -8.0 | -2.8 | 22.4 |
| Nile | -10.4 | 18.9 | 33.5 | 14.0 | 20.9 | 4.9 | 7.3 | -12.4 | 15.5 |
| Olifant | -46.9 | 70.0 | 43.9 | -13.3 | 34.7 | 15.9 | 3.9 | 31.1 | -25.7 |
| Orange | -11.8 | 21.4 | 36.2 | 5.1 | 54.3 | 19.4 | 4.5 | 18.2 | -4.1 |
| Queme | 0.7 | 3.1 | 25.8 | 8.7 | -2.1 | 10.5 | -19.9 | -4.6 | 13.9 |
| Rufiji | 0.6 | 26.5 | 31.9 | 19.0 | 14.1 | 16.1 | 11.6 | 0.4 | 20.9 |
| Sassandra | 6.1 | 19.5 | 31.4 | 19.2 | 6.6 | 22.9 | 9.1 | 1.9 | 22.8 |
| Senegal | 22.1 | 24.3 | 43.4 | 11.2 | 34.5 | 20.8 | -27.0 | -2.0 | 13.8 |
| Upper Blue Nile | 7.3 | 46.9 | 37.9 | 25.7 | 40.8 | 40.7 | 41.6 | 16.1 | 32.1 |
| Zambezi | -3.1 | 25.6 | 33.3 | 20.9 | 32.9 | 19.4 | 9.0 | 1.3 | 23.6 |
| Average | -2.3 | 18.4 | 29.9 | 13.6 | 15.4 | 17.2 | 1.4 | -0.3 | 16.3 |

**Area Weighted Bias**

| | CMRSET | ETMonitor | GLEAM | LandFlux-EVAL | MOD16 | MTE | SSEBop | WaPOR | WECANN |
|---|---|---|---|---|---|---|---|---|---|
| Awash | -44.6 | 46.8 | 39.6 | 23.7 | 48.4 | 39.1 | 34.4 | 14.3 | 11.8 |
| Bandama | -7.2 | -1.7 | 24.0 | -2.0 | -11.9 | 7.1 | -16.4 | -17.2 | 7.2 |
| Buzi | -11.7 | 16.6 | 25.9 | 5.2 | -2.6 | 4.6 | 1.5 | -9.3 | 16.4 |
| Cavally | 11.5 | 19.1 | 32.8 | 18.0 | 10.0 | 20.6 | 6.3 | 13.4 | 30.5 |
| Congo | -4.1 | 18.3 | 16.2 | 10.8 | 22.2 | 17.0 | 1.4 | -5.1 | 20.2 |
| Gambia | 15.5 | 13.4 | 34.2 | 15.1 | 38.9 | 20.5 | -22.6 | -0.4 | 20.9 |
| Groot | -49.4 | 68.9 | 45.4 | 19.6 | 33.7 | 31.1 | 15.4 | 24.2 | -12.5 |
| Komoe | -1.7 | -5.5 | 22.4 | -1.9 | -0.5 | 6.8 | -20.9 | -23.2 | 7.5 |
| Maputo | -8.6 | -4.8 | 12.6 | 2.6 | 5.6 | 6.4 | 9.3 | -1.3 | 1.8 |
| Mono | -2.5 | -2.0 | 23.1 | 0.8 | -10.1 | 7.8 | -14.0 | -7.1 | 10.8 |
| Niger | 19.6 | 22.4 | 42.4 | 16.3 | 16.2 | 17.9 | -8.0 | -2.8 | 22.4 |
| Nile | -10.4 | 18.9 | 33.5 | 9.8 | 20.9 | 4.9 | 7.3 | -12.4 | 15.5 |
| Olifant | -46.9 | 70.0 | 43.9 | 15.9 | 34.7 | 15.9 | 3.9 | 31.1 | -25.7 |
| Orange | -11.8 | 21.4 | 36.2 | 17.3 | 54.3 | 19.4 | 4.5 | 18.2 | -4.1 |
| Queme | 0.7 | 3.1 | 25.8 | 3.4 | -2.1 | 10.5 | -19.9 | -4.6 | 13.9 |
| Rufiji | 0.6 | 26.5 | 31.9 | 15.3 | 14.1 | 16.1 | 11.6 | 0.4 | 20.9 |
| Sassandra | 6.1 | 19.5 | 31.4 | 15.0 | 6.6 | 22.9 | 9.1 | 1.9 | 22.8 |
| Senegal | 22.1 | 24.3 | 43.4 | 16.2 | 34.5 | 20.8 | -27.0 | -2.0 | 13.8 |
| Upper Blue Nile | 7.3 | 46.9 | 37.9 | 32.9 | 40.8 | 40.7 | 41.6 | 16.1 | 32.1 |
| Zambezi | -3.1 | 25.6 | 33.3 | 17.7 | 32.9 | 19.4 | 9.0 | 1.3 | 23.6 |
| Average | -1.6 | 20.6 | 27.1 | 12.9 | 23.1 | 15.9 | 2.6 | -4.0 | 19.3 |

**Fig. 2.** Percentage bias and basin area weighted bias between the long-term annual average calculated ET_WB and ET_RS for all basins and the average of the 20 basins.

**Fig. 3.** Spatial assessment across Africa of each ET product based on selected land cover elements, forest, irrigated areas and water bodies

**Table 1.** Characteristics of remotely sensed ET products

| Product | Temporal Coverage | Spatial Coverage | Temporal Resolution | Spatial Resolution | Estimation Approach | Input Data Source | Reference |
|---|---|---|---|---|---|---|---|
| CMRSET (v20140423) | 2000-2013 | Global | 8-daily | 0.0022° × 0.0022° | P-T Equation, relationship between EVI and GVMI | MODIS | (Lange, 2019) |
| Access: http://remote-sensing.nci.org.au/u39/public/html/wirada/index.shtml | | | | | | | |
| ETMonitor | 2008-2013 | Global | daily | 0.005° × 0.005° | P-M, Gash model, Shuttleworth-Wallace | MODIS | Zheng et al. (2016) |
| Access: email first author in reference | | | | | | | |
| GLEAM (v3.2a) | 1980-2016 | Global | Daily | 0.25° × 0.25° | P-T Equation, soil stress factor | AMSR-E, LPRM, TRMM | Martens et al. (2017); Miralles et al. (2011) |
| Access: www.gleam.eu | | | | | | | |
| LandFlux-EVAL | 1989-2005 | Global | Monthly | 1° × 1° | Ensemble Approach | See reference | Mueller et al. (2013b) |
| Access: https://iac.ethz.ch/group/land-climate-dynamics/research/landflux-eval.html | | | | | | | |
| MOD16 (vA3) | 2000-2014 | Global | Monthly | 0.0083° × 0.0083° | P-M Equation, surface conductance model | MODIS | Mu et al. (2011, 2007) |
| Access: https://modis.gsfc.nasa.gov/data/dataprod/mod16.php | | | | | | | |
| MTE (vMay12) | 1982-2012 | Global | Monthly | 0.5° × 0.5° | MTE approach, training using in-situ observations, flux tower data | Eddy Co-variance, in-situ | Jung et al. (2011) |
| Access: https://climatedataguide.ucar.edu/climate-data/fluxnet-mte-multi-tree-ensemble | | | | | | | |
| SSEBop (v4) | 2003-2017 | Global | Monthly | 0.0096° × 0.0096° | P-M Equation, ET fractions from $T_s$ estimates | MODIS | Senay et al. (2013) |
| Access: https://earlywarning.usgs.gov/fews/search | | | | | | | |
| WaPOR (v1.1) | 2009-2017 | Africa | Dekadal | 0.0022° × 0.0022° | P-M Equation, calculates E, T and I separately | MODIS, GEOS-5/MERRA | FAO (2018) |
| Access: https://wapor.apps.fao.org/home/1 | | | | | | | |
| WECANN (v1.0) | 2007-2015 | Global | Monthly | 1° × 1° | ANN approach, training using observations and model based LE | GOME-2 | Alemohammad et al. (2017) |
| Access: https://avdc.gsfc.nasa.gov/pub/data/project/WECANN/ | | | | | | | |

**Fig. 4.** Characteristics of remotely sensed ET products

| Basin | Variable | Data Availability | Trend | hypothesis | p-value | z-value | no. of samples |
|---|---|---|---|---|---|---|---|
| Awash | ET | 1990-2004 | no trend | false | 0.2496 | -1.1514 | 14 |
| | P | 1979-2016 | | | | | 38 |
| | Q | 1990-2004 | MK test not conducted, no trend found in ET | | | | 15 |
| Bandama | ET | 1979-1996 | no trend | false | 0.7619 | -0.3030 | 18 |
| | P | 1979-2016 | | | | | 38 |
| | Q | 1970-1996 | MK test not conducted, no trend found in ET | | | | 27 |
| Blue Nile | ET | not enough ET data points to conduct MK test on calculated ET | | | | | 4 |
| | P | 1979-2016 | no trend | false | 0.6875 | -0.4023 | 38 |
| | Q | 1900-1982 | decreasing | true | 0.0009 | -3.3271 | 83 |
| Buzi | ET | not enough ET data points to conduct MK test on calculated ET | | | | | 5 |
| | P | 1979-2016 | no trend | false | 0.4210 | -0.8046 | 38 |
| | Q | 1957-1983 | no trend | false | 1.0 | 0.0 | 23 |
| Cavally | ET | 1979-1996 | no trend | false | 0.54449 | -0.6060 | 18 |
| | P | 1979-2016 | | | | | 38 |
| | Q | 1970-1996 | MK test not conducted, no trend found in ET | | | | 27 |
| Congo | ET | 1979-2010 | no trend | false | 0.0830 | -1.7336 | 31 |
| | P | 1979-2016 | | | | | 38 |
| | Q | 1903-2010 | MK test not conducted, no trend found in ET | | | | 108 |
| Cunene | ET | 1980-2015 | increasing | true | 0.0003 | 3.5823 | 36 |
| | P | 1979-2016 | | | | | 38 |
| | Q | 1980-2015 | MK test not conducted, no trend found in ET | | | | 36 |
| Gambia | ET | not enough ET data points to conduct MK test on calculated ET | | | | | 5 |
| | P | 1979-2016 | no trend | false | 0.2579 | 1.1315 | 38 |
| | Q | 1979, 1981-82, 1984,1988 | no trend | false | 0.8065 | 0.2449 | 5 |
| Groot | ET | 1979-2015 | no trend | false | 0.1697 | 1.3733 | 37 |
| | P | 1979-2016 | | | | | 38 |
| | Q | 1964-2014 | MK test not conducted, no trend found in ET | | | | 51 |
| Kamoe | ET | 1979-1996 | no trend | false | 0.3633 | -0.9091 | 18 |
| | P | 1979-2016 | | | | | 38 |
| | Q | 1970-1996 | MK test not conducted, no trend found in ET | | | | 27 |
| Lake Chad | ET | not enough ET data points to conduct MK test on calculated ET | | | | | 4 |
| | P | 1979-2016 | increasing | true | 0.0194 | 2.3384 | 38 |
| | Q | 1983-1986 | no trend | false | 0.3081 | -1.0190 | 4 |
| Maputo | ET | not enough ET data points to conduct MK test on calculated ET | | | | | 5 |
| | P | 1979-2016 | no trend | false | 0.3393 | -0.9555 | 38 |
| | Q | 1953-1983 | no trend | false | 0.1261 | -1.5297 | 31 |
| Mono | ET | 1979-2007 | no trend | false | 0.5115 | -0.6565 | 29 |
| | P | 1979-2016 | | | | | 38 |
| | Q | 1944-2007 | MK test not conducted, no trend found in ET | | | | 64 |
| Niger | ET | 1979-2006 | no trend | false | 0.6214 | 0.4939 | 28 |
| | P | 1979-2016 | | | | | 38 |
| | Q | 1970-2006 | MK test not conducted, no trend found in ET | | | | 37 |
| Nile | ET | not enough ET data points to conduct MK test on calculated ET | | | | | 6 |
| | P | 1979-2016 | no trend | false | 0.2909 | 1.0560 | 38 |
| | Q | 1912-1984 | no trend | false | 0.0693 | 1.8164 | 56 |
| Okavango | ET | 1979-2014 | increasing | true | 0.0127 | 2.4926 | 36 |
| | P | 1979-2016 | | | | | 38 |
| | Q | 1950-2014 | MK test not conducted, no trend found in ET | | | | 65 |
| Olifant | ET | 1979-2014 | no trend | false | 0.9457 | 0.0681 | 36 |
| | P | 1979-2016 | | | | | 38 |
| | | | MK test not conducted, no trend found in ET | | | | |
| | Q | 1927-2014 | | | | | 88 |
| Orange | ET | 1979-2016 | no trend | false | 0.6691 | 0.4274 | 38 |
| | P | 1979-2016 | | | | | 38 |
| | Q | 1936-2014 | MK test not conducted, no trend found in ET | | | | 79 |
| Queme | ET | 1979-80, 1982-84, 1990-2005, 2007 | no trend | false | 0.3377 | 0.9587 | 22 |
| | P | 1979-2016 | | | | | 38 |
| | Q | 1948-2007 | MK test not conducted, no trend found in ET | | | | 60 |
| Rufiji | ET | not enough ET data points to conduct MK test on calculated ET | | | | | 0 |
| | P | 1979-2016 | no trend | False | 0.6508 | -0.4526 | 38 |
| | Q | 1954-1978 | no trend | False | 0.9741 | -0.0324 | 20 |
| Sassandra | ET | 1979-1996 | no trend | false | 0.8796 | 0.1515 | 18 |
| | P | 1979-2016 | | | | | 38 |
| | Q | 1970-1996 | MK test not conducted, no trend found in ET | | | | 27 |
| Save | ET | not enough ET data points to conduct MK test on calculated ET | | | | | 3 |
| | P | 1979-2016 | no trend | False | 0.8801 | 0.1509 | 38 |
| | Q | 1968-1981 | increasing | True | 0.0118 | 2.5183 | 14 |
| Senegal | ET | 1979-1989 | no trend | false | 0.2129 | 1.2456 | 11 |
| | P | 1979-2016 | | | | | 38 |
| | Q | 1979-1989 | MK test not conducted, no trend found in ET | | | | 11 |
| Tana | ET | not enough ET data points to conduct MK test on calculated ET | | | | | 0 |
| | P | 1979-2016 | decreasing | True | 0.0006 | -3.4447 | 38 |
| | Q | 1975-1978 | no trend | False | 0.7341 | -0.3397 | 4 |
| Upper Blue Nile | ET | not enough ET data points to conduct MK test on calculated ET | | | | | 8 |
| | P | 1979-2016 | no trend | False | 0.6875 | -0.4023 | 38 |
| | Q | 1961-1983 | no trend | False | 0.1339 | -1.4988 | 26 |
| Void | ET | not enough ET data points to conduct MK test on calculated ET | | | | | 3 |
| | P | 1979-2016 | no trend | False | 0.1251 | -1.5338 | 38 |
| | Q | 1979-1981 | increasing | True | 0.0483 | 1.9748 | 7 |
| Zambezi | ET | 1979-1990 | no trend | false | 0.5371 | 0.6172 | 12 |
| | P | 1979-2016 | | | | | 38 |
| | Q | 1960-1990 | MK test not conducted, no trend found in ET | | | | 31 |

**Fig. 5.** Mann-Kendall test results for all basins based on evapotranspiration, precipitation and discharge as well as data details

[Figure]

**Fig. 6.** Comparison of the EWEMBI, CHIRPS and MSWEP precipitation products on their prediction of mean P across the basins

**Table 2.** Characteristics of precipitation products

| Product | Temporal Coverage | Spatial Coverage | Temporal Resolution | Spatial Resolution | Input Data Source | Reference |
|---|---|---|---|---|---|---|
| EWEMBI (v1.1) | 1979-2016 | Global | daily | 0.5° × 0.5° | ERA-Interim, WFDEI: (Weedon et al., 2014), E2OBS: (Calton et al., 2016) | (Lange, 2019) |
| | Access: http://doi.org/10.5880/pik.2019.004 | | | | | |
| CHIRPS (v2.0) | 1981-2019 | quasi-Global | daily | 0.05° × 0.05° | in situ precipitation gauges, TRMM: (Huffman et al., 2007), CMORPH: (NCAR, 2017) | Funk et al. (2015) |
| | Access: https://chc.ucsb.edu/data/chirps | | | | | |
| MSWEP (v2.2) | 1979-2017 | Global | 3-hourly | 0.1° × 0.1° | in situ precipitation gauges, CMORPH, TRMM, GSMaP: (JAXA, 2009), IRA-Interim | Bai and Liu (2018) |
| | Access: http://www.gloh2o.org/ | | | | | |

**Fig. 7.** Characteristics of precipitation products
**Table 3.** Characteristics of discharge data

| Product | Temporal Coverage | Spatial Coverage | Temporal Resolution | Spatial Resolution | Input Data Source | Reference |
|---|---|---|---|---|---|---|
| GRDC (v1.1) | 1806-2019 | Global | daily | point data | in situ discharge gauges | |
| | Access: https://www.bafg.de/GRDC/EN/Home/ | | | | | |
| HYDR-VUB | 1932-2018 | Global | daily | point data | in situ discharge gauges | |
| | Access: on request to http://www.hydr.vub.ac.be/ | | | | | |

**Fig. 8.** Characteristics of discharge data

**Table 4.** Characteristics of potential evapotranspiration products

| Product | Temporal Coverage | Spatial Coverage | Temporal Resolution | Spatial Resolution | Input Data Source | | Reference | |
|---|---|---|---|---|---|---|---|---|
| Hargreaves | 1979-2012 | Global | daily | 0.05° × 0.05° | WFDEI, DEM | SRTM | (Sperna et al., 2015) | Weiland |
| | Access: https://wci.earth2observe.eu/ | | | | | | | |
| Penman-Montieth | 1979-2012 | Global | daily | 0.05° × 0.05° | WFDEI, DEM | SRTM | (Sperna et al., 2015) | Weiland |
| | Access: https://wci.earth2observe.eu/ | | | | | | | |
| Priestly-Taylor | 1979-2012 | Global | daily | 0.05° × 0.05° | WFDEI, DEM | SRTM | (Sperna et al., 2015) | Weiland |
| | Access: https://wci.earth2observe.eu/ | | | | | | | |

**Fig. 9.** Characteristics of potential evapotranspiration products

---

## Author Comment (AC4) · 8 Oct 2019

Overview

We want to thank the reviewers for their dedication in reviewing the manuscript. We also are thankful for their thoughtful and constructive suggestions and comments. We have addressed all the comments raised by the reviewers and the manuscript has improved from the proposed changes.

Reviewer 2

Summary

The article "Can we trust remote sensing ET products over Africa?" by Imeshi Weeras-

inghe et al. presents an evaluation analysis of the eight satellite - based evapotranspiration (ET) products over selected African river basins against the ET estimates derived from the water balance equation. The main conclusion of the study ranks the selected ET products in accordance with the results of the comparison analyses. The topicality and scientific relevance of the research question addressed in this study is high considering the sparseness of the in situ ET data in the region as well as the urgency of having a high quality ET estimates for the climate related problems in Africa. However, at this point I cannot recommend publication of this article as it (i) – contains a number of significant methodological inaccuracies and (ii) – gives poor explanation and presentation of the performed analyses and graphics. Also, stylistically and structurally the manuscript needs a substantial improvement. I highly recommend a major revision of the manuscript followed up by an internal review prior to the re-submission. Details follow.

Authors Response

We thank Reviewer 2 for his/her time spent on the review and constructive suggestions and feedback that he/she has given to improve the manuscript. Below, we address the issues that were raised for the improvement of the manuscript.

Reviewer 2, General Comment

Generally, the presentation style of the paper makes it often hard to understand the correctness and hence the added value of the illustrated results. The lack of accompanying relevant information in a well-written form along with the multitude of presented data combinations in a variety of forms and at different scales in many cases confuses an understanding of (i) which data sets were used for this concrete calculation, (ii) in which form the data went into the following graphic, (iii) what the estimates were compared to, (iv) how many and which basins were used this time, (v) when data mean was used and over which scales the averaging was done? In my opinion, the paper did not succeed in wrapping up the results in a clear manner. Usage of multiple data

levels, i.e. 3 reference rain datasets, 8 ET products, 27/20 basins with/without trends, different temporal resolutions (from one value to time-series), two spatial levels (from basin integral to pixel-basis) comprises a fairly large number of levels of information which the authors should unwrap and present in a very simple, consequent and logical manner. In the present version of the paper this have not been archived. The presentation style needs a thorough improvement, including restructuring of the manuscript, improvement of English grammar and scientific wiring style itself. One of the major remarks is that the whole manuscript text is written in a very intermittent and superficial manner. The explanations throughout the whole paper are significantly lacking concreteness. Also, confusion and replacement of some terms used throughout the study (e.g. trends as trends or trends as tendency to show certain value, among others) together with the multitude of abbreviations used in the text makes it very hard to follow the presentation (details given below). Finally, some data descriptions and methodological assumptions (in particular, downscaling to the smallest grid and usage of different time periods) raises a number of questions related to their correctness and validity. The details are given further below.

Authors Response

The paper has gone through a substantial restructuring in terms of the presented figures and different sections and sub-sections. One of the main reasons behind this paper was to use different products with varying temporal and spatial scales for evaluation due to the fact that these products are a sample of the products available for use. In terms of which data sets were used for the concrete calculation, this was mentioned in the introduction and methodology sections. The 9 products being evaluated have all been used in the calculation. They have been resampled to the same spatial and temporal resolution. The average of the three precipitation products have been used and the average of the three PET products have been used. The discharge data has been obtained from GRDC and HYDR and have been converted to yearly averages. For each graphic and section, the methodology has been described. Each section described clearly what the ET estimates are being compared to both in the methodology section, in the results section and in the figure captions. According to how many basins are being used in the study. This is clearly identified in the results section of the paper including with graphics. The data mean for basins and land cover elements was used and stated. Long-term averaging was found after finding yearly averages for the different basins and land cover elements. The manuscript is written by an English native speaker and has been checked by a native English professor for structure and grammar. More concrete explanations and statements have been corrected to be clearer. The confusion between certain terms in their use in this paper such as trends and tendencies has been corrected. All abbreviations have been opened and no abbreviation has been used without explanation. As to the questions raised due to downscaling of products, these are addressed further into the specific questions raised. In short, this was done as to not lose any information from the high spatial resolution products at the same time not losing information from the coarser scale products. It would have been simpler to resample all products to the coarsest resolution in terms of computational efficiency and storage space. However, this was not done in order to ensure all features of ET products such as, spatial resolution, were evaluated according to the individual products as would not be the case if resampled from high to coarse. The paper addresses the use of different time periods as to the reasons why they are need and the possible reasons why they can be used.

Authors Changes in Manuscript

The entire manuscript should be shown here as most parts have been adapted and changed. Therefore the comments to the changes in the manuscript to the specific comments have been shown.

Reviewer 2, Specific Comment 1 – Scientific Relevance

From the abstract, introduction and methodology sections it remains unclear how new is the water balance (WB) method, how accurate is the method and which other studies

already used it for similar tasks. The abstract even makes a false impression that the authors developed a method and not used the existing one (p1, L6). The introduction in turn makes an impression that the focus of the study is the methodology (p3, L9) and not the evaluation of the existing products. In this view, the authors must provide an overarching literature review of the studies that already used the WB ET estimates for satellite products evaluation, and also studies which evaluated the same satellite products over Africa using the same or other techniques. One of such study examples is the Miralles et al. (2016), which is also referenced in the present manuscript. Note, that Miralles' study also involved the African river basins. This has to be explicitly mentioned in the introduction. The authors should also then place their results into the findings of others. This is not done at the present state of the paper.

Authors Response

The intention was not to give the impression that the catchment water balance methodology was developed within the scope of this study. The paper has been amended so that this false impression is not given. The introduction has been changed to show focus on the evaluation of the products and not the methodology. A literature review of similar comparisons using some products under evaluation in this study among others has been conducted. (P1,L7), (P4,L4), (P4,L10), (P3,L5), (P19,L19)

Authors Changes in Manuscript

Abstract "Thus we conduct a methodological evaluation of nine existing RS derived and other ET products in order to evaluate their reliability at the basin scale."

Introduction "Therefore, this study focuses on evaluating nine existing, mostly open access, ET products using a water balance approach over Africa." "The evaluation of the products will be conducted using a) a comparison of their performance against calculated ETWB, b) a robustness check of their performance against the Budyko curve which provides a reference condition for the water balance assuming it correctly partitions P into Q and c) a spatial variability assessment using specific land cover elements

[revised manuscript text omitted]

Reviewer 2, Specific Comment 2 – Scientific Relevance

The authors should be more careful in formulating their scientific conclusions. The following sentence in the abstract: "However our recommendation would be the three highest ranked products being CMRSET, SSEBop and WaPOR." sounds rather subjective and needs further motivation (especially considering the huge differences in spatial-temporal scales between the products, as well as the manipulations on interpolation, then vice versa - integration - done within the study). Why first three? The same remark applies for the conclusions.

Authors Response

We have changed subjective recommendations within the study to be more conclusive based on the findings. (P1,L18), (P20,L23)

Authors Changes in Manuscript

Abstract "Based on the evaluation criteria in this study the three highest ranked products, CMRSET, SSEBop and WaPOR would suit many of the needs of readers due to low biases and good spatial variability."

Conclusion "Therefore, if we answer our question of whether to trust remote sensing estimates of ET across Africa, the answer is not black and white. Yes, in general we can trust the products under evaluation in this study. CMRSET, WaPOR and SSEBop show low biases in estimations and a good spatial distribution of ET patterns. Each of these products have relatively high resolutions and both CMRSET and SSEBop are global products. Depending on the study under question, whether an early and long time period is needed, whether a higher or lower resolution is required, whether looking at the global or regional scale or whether looking only at land evapotranspiration, a

different product may be more suited than another. However, a large consideration to be kept in mind for Africa, is that the three highest ranked products, CMRSET, SSEBop and WaPOR have low biases and perform well in spatial variability and will suit most needs within a given study."

Reviewer 2, Specific Comment 3 – Scientific Relevance

The study does not mention anything at all about the quality of the reference precipitation data sets, nor about the quality of the final WB- based ET product. The study should also provide or mention some quantitative assessment to the magnitude of the differences which arise only due to application of different rainfall products. This will (i) - substantiate introduction of three different rainfall products in the paper and (ii) – justify better the obtained differences in ET between the products.

Authors Response

We have now used the average of the three precipitation products due to only slight differences found in the calculated mean precipitation between the three products also considering that we use the average of the three products for ranking purposes. The comparison between the three precipitation products have been included in the paper. We have also tried to evaluate the reasons behind the differences between the ET products within the conclusion. This was already mentioned somewhat in the discussion (P5,L23), (P9,L25), (Figure 5), (P20,L15).

Authors Changes in Manuscript

"Since direct observations of precipitation from gauges were not used, precipitation was taken as the average of the three data products EWEMBI, CHIRPS and MSWEP."

"Precipitation and potential evapotranspiration were taken as the average of three products. Here we compare the results of the different P and PET products for the basins being analysed. We see that the three precipitation products show little differences in their estimations of long-term average P across the basins. No large outliers can be

seen (Figure 5)."

Conclusion "A big difference between the top three ranked products and the others is the high spatial resolution as well as the estimation of ET as a whole rather than only land ET in most other cases. However, no pattern can be found between the product ranking and the forcing or ET calculation methods."

Reviewer 2, Specific Comment 1 – Data Products

The data section needs a major elaboration. Many paragraphs appear rather like a snippets of information with lack of logical sequence, and hence they often fail to deliver main message of the paragraph or peculiarity of the concrete product. My suggestion would be: i) to either extend the data product descriptions to make them more complete and understandable or vice versa, provide only a reference links to the web sources and main papers of the products, and use instead the data section to discuss / group the products by their similarities and differences, advantages and disadvantages which can further help interpreting the paper results. ii) to omit the repetition of the time period and resolution information since they are already given in the table; iii) to place all products into the tables for consistency and clarity, i.e. also precipitation products, discharge data and reference data should be summarised in the same or separate table.

Authors Response

We have decided to take the advice of the reviewer to provide only reference links to the sources and papers of the products with a brief description on the similarities, differences, advantages and disadvantages. All other products have also been summarised in tables consistent with the ET products. (P4,L17), (Table 1), (Table 2), (Table 3), (Table 4)

Authors Changes in Manuscript

"The derived ET products being evaluated in this study include CMRSET, ETMonitor, GLEAM, LandFlux-EVAL, MOD16, MTE, SSEBop, WaPOR and WECANN. Overall there are large differences between the products which results in certain advantages and disadvantages between products. All products have a global spatial coverage (advantage) except for WaPOR (disadvantage). All products are openly accessible (advantage) except for ETMonitor (disadvantage). GLEAM and ETMonitor have a daily, CMRSET has an 8-daily and WaPOR has dekadal temporal resolution (advantage) over other products which have monthly or yearly resolutions(disadvantage). Most products are still ongoing (advantage) except for ETMonitor, LandFlux-EVAL and MTE (disadvantage). GLEAM, MTE and LandFlux-EVAL have data available prior to 1990 (advantage) with all other product data available after 1999 (disadvantage). These different ET products give a good sample of the available data sets to choose from with their many advantages and disadvantages. "

Reviewer 2, Specific Comment 2 – Data Products

Many product descriptions and references miss version numbers. Those must be included, since depending on the product version there might be some already known issues related to a parameter derivations.

Authors Response

We have now included version numbers of the products in Table 1. (Table 1)

Authors Changes in Manuscript

Refer to Table 1 under Reviewer 2, Specific Comment 1 – Data Products

Reviewer 2, Specific Comment 3 – Data Products

Check carefully the correct citations, once you add the product version numbers. I am more familiar with the GLEAM product, and I know that for GLEAM v3 (if you used that version) the correct references are Miralles 2011 (HESS) and Martens 2016 (GMD).

Authors Response
We have carefully checked the references and have included and amended these in Table 1. (Table 1)

Authors Changes in Manuscript

Refer to Table 1 under Reviewer 2, Specific Comment 1 – Data Products

Reviewer 2, Specific Comment 4 – Data Products

It is also a rule of a good scientific practice to provide/cite the data source: a webpage, ftp or a personal communication. No data sources are mentioned in the current manuscript version. For GLEAM, for example it should be the web portal: www.gleam.eu; For MSWEP: http://www.gloh2o.org/ (?), etc.

Authors Response

We have now also added data sources to the manuscript in Table 1. (Table 1)

Authors Changes in Manuscript

Refer to Table 1 under Reviewer 2, Specific Comment 1 – Data Products

Reviewer 2, Specific Comment 5 – Data Products

One of my major remarks here concerns inaccurate or sometimes even false information in data set descriptions. That is unacceptable. Please, check carefully all product descriptions you are giving! On the example of GLEAM: - GLEAM is not a physically-based model, Prietsley-Taylor, the interception loss model, the stress module, and the water-balance model in GLEAM which form the core of GLEAM are all empirical! One can call it a process-based model, as it empirically describes the process needed to estimate E from satellites; - Table 1: GLEAM does not use CMORPH at all! That is simply wrong information. - Alemohammad et al., 2017 is the reference to a paper where they describe another method of deriving E. It is not clear why this is included in this GLEAM section?.
Authors Response

We apologise for the incorrect description of some of the models and have made sure to change or take out the descriptions. The incorrect information has been deleted with the reconstruction of the data section. This should be reflected in Table 1. (Table 1)

Authors Changes in Manuscript

Refer to Table 1 under Reviewer 2, Specific Comment 1 – Data Products

Reviewer 2, Specific Comment 1 – Methodology and results

As it was already mentioned earlier, the presentation of the calculation steps is done in a rather superficial manner. Lots of information is not given or remains unclear. E.g.: - which concrete quality control steps were involved in the selection of the basins and, which additional analyses were done and by whom? (e.g. p8, L7-10) - how were the basin boundaries defined? - what is the time-period of available discharge data? - how the integration over the basins is exactly done? were the simple mean or the areaweighted mean of ET or P fields used when averaging over the basin area? - which manipulations were done with the precipitation data prior averaging it over the basins? Were the data also re-scaled to the 0.0022 deg resolution and then averaged over the basins? Never mentioned. - In their paper Miralles et al., 2016 applied additional quality control check on the difference between the GRDC-reported area and the area calculated from basin boundaries. Would not it be also relevant for the present study? - How was the MPM calculated? The products have big differences in resolution. The averaging of the products to get the MPM without applying corresponding weights can be a source of errors.

Authors Response

We mention in the paper that we select the basins considering the availability of the discharge data at the outlet of a particular basin. This was how the initial 27 basins were selected. This was mentioned in the previous version and has not changed in

this current version of the manuscript. A complete restructure of the methodology section has been done in order to incorporate the suggestions from this particular comment. Specifically, we now mention how the basin boundaries are defined (from the 'Major River Basins of the World' (MRBW) shapefile (World Bank 2017)). The available time period of the discharge data has been added and can be found in Table A1 in Appendix A. Integration over the basin has been done according to the MRBW shapefile boundaries for each basin with the mean of each basin within the shapefile recorded. This description has been added to the manuscript. The basin area weighted mean for averaging over a basin was only used in the statistics for RMSE and bias, otherwise the simple mean was used as stated in the manuscript previously and in this new version. No manipulations were done to the precipitation data prior to averaging over the basins except for obtaining yearly averages. This is now mentioned. Yes the data was rescaled to 0.0022 deg resolution and then averaged over the basins. This is now mentioned. We have also conducted an analysis on the difference between the MRBW shapefile areas taken for the basins and the area reported by the GRDC and mentioned only the potential problematic cases. We have now not used the MPM but an existing benchmark ensemble product, LandFlux-EVAL. (P5,L20), (Table A1), (P5,L24), (P5,L1), (P4,L26),

Authors Changes in Manuscript

"Catchment or basin areas were taken from the 'Major River Basins of the World' (MRBW) shapefile (World Bank 2017). Discharge was converted from cubic meters per second to millimeters per year using the above mentioned catchment areas for all years of data availability for each basin."

"Basin average precipitation was calculated for the years 1979-2016 according to the MRBW shapefile boundaries recording the basin mean."

"Precipitation products were averaged at yearly temporal resolution for the purposes of this study."

"All products have been projected and gridded on a 0.0022 deg resolution geographic grid and averaged at yearly temporal resolution for the purposes of this study."

"The MRBW shapefile area did not differ greatly with the drainage area reported by the GRDC except in two cases. Here we found the ETWB calculated using the two areas only differed by 2.5 percent and 3.3 percent and thus kept these basins in the analyses."

Reviewer 2, Specific Comment 2 – Methodology and results

(i) - It remains not very clear from the text over which values the correlation analysis is performed? Over time-series of annual means? Over multi-year averages of different basins? Should be put more clear. (ii) - Units of the correlations are not common, and confuse the interpretation. Correlation should rank from -1 to 1. Besides, it is never clear from all the graphs with percentages, by what value the normalisation was done. (iii) – Correlations should always provide significance measure, or the latter should be mentioned in the text.

Authors Response

(i) - The correlation was performed over multi-year averages (long-term averages) across all basins under evaluation (20 basins). This has been mentioned in the manuscript. (P6,L16) (ii) – Correlations have been changed to -1 to 1 without units. The normalisation for each basin was conducted based on the calculated ETWB for each basin. These have been adapted in the manuscript. (Table 5), (P11,L15) (iii) – This has been adapted in the manuscript to reflect the significant measure. (P11,L11)

Authors Changes in Manuscript

(i) - "Correlations were calculated based on long-term averages across all basins."

(ii) - Table 5. "Percentage biases were normalised based on the calculated ETWB for each basin."

(iii) – "There is a significant positive correlation for all products ranging from 0.89-0.97 with GLEAM and LandFlux-EVAL showing the strongest relationships with ETWB"

Reviewer 2, Specific Comment 3 – Methodology and results

The choice of the highest resolution is one of the two major remarks that I have to the methodological part: (i) – Generally, it is not common to interpolate products to the highest resolution, especially when the difference between the highest and the coarsest resolution is that high. It would be more correct to upscale the higher resolved data to the coarser estimates to minimise the bias. (ii) – Besides, the fact that the comparison of the products is mostly done at the basin level, the downscaling does not seem to make sense at all. First you interpolate the coarse data to the very high resolution, and then, you integrate it back again over the river basin. This clearly can be a source of additional biases and errors, which also raises my doubts about the validity of the ranking results. (iii) – All the above inter alia also raises a question of what is the minimal area of the smallest basin you have, and whether it is resolved by the products with the coarsest resolution at all?.

Authors Response

(i) - We did not upscale the higher resolution products to coarser resolution products as our aim was not to minimise bias between products but to evaluate each product independently of the other according to their features (e.g. spatial resolution) regardless of whether they were advantageous or disadvantageous. I believe if we were trying to find the effects of ET calculation methods or forcing data and looking at the different products for comparison this would make sense that we would need to minimise other biases between the different products. However, as the goal of our manuscript is to evaluate different offerings of ET products regardless of temporal and spatial resolutions or coverage, forcing data, ET calculation method, etc, it does not make sense for us to minimise bias between products. We want to calculate the bias of each product with respect to a reference (ETWB). In this respect we do not lose any information by

downscaling from coarse to high resolution and in fact we found that for basin boundaries to be as close to the shapefiles as possible, the data set of GLEAM, for example, showed very little difference (approximately 0-5mm/year) in the estimation of mean ET for different basins when using the coarse resolution or the resampled high resolution dataset. Whereas a slightly larger difference range was found for the CMRSET product (0-50mm/year) when comparing the high resolution product to the resampled coarse resolution for a sample of basins. In this regards we did not believe that there was a disadvantage by resampling to the highest resolution of the products. Also, since we were trying compare individual products on their own merits of each feature, this enable a more accurate comparison without minimising bias between the products. (ii) – As with point (i) above, we did not find that there were any additional biases and errors from this method of evaluation and found very small differences when looking at the two different methods of up or downscaling resolutions. In fact, the biases were greater when resolving from high resolution to coarse resolution and the biases slightly smaller when resolving from low resolution to high resolution when looking at certain basin means according to the reference used (ETWB). Although since very small differences were seen, they were almost negligible. Therefore the validity of the ranking results still hold. (ii) – The smallest basin is >30000km2 which for certain products such as LandFlux-EVAL and WECANN, this would not be fully resolved. However the intent was to also include basins that were smaller in our analysis to: 1. Have a good spatial coverage across Africa and 2. Have a range of basin sizes to evaluate the products on. Even though we do not see any spatial variability in products such as LandFlux-EVAL and WECANN in the smaller basins. Their prediction of long-term average ET for smaller basins, especially LandFlux-EVAL showed lower biases than higher resolution products such as ETMonitor.

Authors Changes in Manuscript

For the above reasons, nothing with regard to this point was changed within the manuscript.

[Figure]

Reviewer 2, Specific Comment 4 – Methodology and results

The second and the most major remark of mine is related to the application of the analyses at different time-periods. (i) - First of all it never comes clear what is the time period of available discharge data for every basin; (ii) – From the Tabel 2 it appears that for most of the basins discharge data does not extend the whole period of available precipitation data at all, and the spread of data periods is huge among the basins. In this view I do not understand at all how the analyses tests were done? (ii) – The test for the effect of temporal variability on annual means mentioned in the discussion section was done only for the four basins, while 20 basins are analyzed throughout the study. Moreover, Congo - one of the four tested basins - has data only till 2010, while remote sensing ET products span up to 2017. In this view, I would not be able to call it a fair validity test! (iii) – Clearly, the exclusion of periods with trends does not account for the temporal variability of data which can still result in the pretty different annual means. So, the effect of temporal variability on annual means must be done for all the basins, which are used for the evaluation of the satellite ET products in order to draw a fair conclusions. (iv) – Calculating trends only for the WB ET reference data set is not a complete analysis. If a satellite data product has a trend, this also has to be mentioned, and maybe even that product should not participate in the validation (?) To conclude, if the tests will show that the variability indeed matters, then none of the performed analyses is valid since they will all be affected by the differences due to variability.

Authors Response

(i) – The period of available discharge data for each of the 27 basins where ETWB was initially calculated for has been added and can be found in Table A1 within the manuscript. (ii) – Here we used the long-term average precipitation data minus the long-term average discharge data to calculate the ET. This is mentioned in the manuscript. We also calculated the ET based on the average precipitation and discharge based on the overlapping periods and found a maximum of 5% difference in ET from both methods. In most cases 0% difference was found. (P5,L26) (ii) – the test for

temporal variability was conducted on the selected four basins as they were the only basins with long enough time periods to conduct this test. We used these as samples to show that in all cases tested the difference in calculated ET was minimal. Congo was also used, although it only had data until 2010 so that we could test more than just three basins. The corresponding periods were used in the remote sensing products so we made sure the periods tested overlapped. In this regard we believe this was a fair validity test. (iii) – We agree that if possible the test for temporal variability should have been conducted on all basins, but we are unable to do this based on the available time periods of the data. Therefore we took the four basins as a sample and surmised that due to finding minimal differences in calculated ET, we could compare long-term averages from different time periods. (iv) – Considering the point is evaluating the prediction of ET estimates by different ET products, whether they do or do not have trends is not a basis for inclusion or not in this analysis. It would be interesting to find out whether the different products do show trends for particular basins but this was not in the scope of this study. Our goal was to evaluate based on a reference which we needed to calculate in an accurate manner, which would mean no trends when looking at long-term averages of different time periods. Therefore our 20 basins under analysis without trends in their calculated long-term ETWB was used as a reference to compare with ET product estimations. If an ET product had a trend in their estimation of ETWB for one of the 20 basins under analysis, then this would most likely have a higher bias when compared with the reference ETWB. It was not relevant for us to see if ET products had trends or not, only to evaluate their ET estimates.

Authors Changes in Manuscript

"Long-term ETWB was calculated by using the long-term average discharge and precipitation data for each catchment."

Nothing else for this point was changed in the manuscript.

Reviewer 2, Specific Comment 5 – Methodology and results

My advice would be to not use percentages for all the figure results. This only confuses the interpretation. Use -1 to 1 scale for correlation and differences.

Authors Response

We did not use percentages for the overall statistics in Table 5. However, we still used percentages when looking at bias and basin area weighted bias as subjectively we found these results easier to interpret and was given this advice from different advisers of the manuscript. (Table 5) Authors Changes in Manuscript

Refer to Table 5 under Reviewer 2, Specific Comment 2 – Methodology and results

Reviewer 2, Specific Comment 6 – Methodology and results

The raking of products based on visual inspection is rather speculative for me. For ex on Fig 9 it was impossible to follow the text conclusions: I did not see where irrigation area is, to which reference product other products are compared, why MPM and some other products have no data and why GLEAM is concluded to perform worst? The same for Fig 10..

Authors Response

We re-did this section to include forest as an additional land cover type for inspection as well as using highlighted areas across the entire African continent rather than zooming into selected areas shown in figure 5. We agree that this is a rather subjective method, however, we also believe it is relatively visible to see the difference between the products especially when using different scales to be able to visually interpret the results. We believe these maps are an added advantage for reader interested in spatial characteristics of ET. However, due to its potential subjectivity in ranking, we do two final rankings, with and without visual inspection to minimise this subjectivity. (P13,L4), (Figure 10)

Authors Changes in Manuscript

"Figure 10 shows ET across Africa for all ET products with the specific land cover elements (forest, irrigated areas and water bodies) highlighted. Two different scales are used in order to be able to visually compare the products according to spatial variability rather than magnitude of ET. For products where large biases were found, a scale of 0-1200 mm/year was used and for the remaining products a scale of 0-1800 mm/year was used. Visually, all products capture the forested area. Irrigated areas are also captured well by most products. GLEAM and LandFlux-EVAL do not capture the majority of selected irrigated areas. CMRSET, ETMonitor, SSEBop and WaPOR capture most of the selected irrigated areas while the remaining products capture a few. GLEAM, LandFlux-EVAL, MOD16, MTE and WECANN only estimate land ET and thus do not have ET across water bodies. The remaining products capture the water bodies well, with CMRSET and ETMonitor showing larger differences in their estimations of ET across water bodies than the surrounding areas over SSEBop and WaPOR."

Reviewer 2, Specific Comment 7 – Methodology and results

The analyses of comparing products over one irrigation area, and one lake, where some products have no data, and others do not even resolve the region does not make much sense to me, nor it is complete enough to make a serious conclusion on which product is better or worst.

Authors Response

Please refer to answer from section Reviewer 2, Specific Comment 6 – Methodology and results.

Reviewer 2, Specific Comment 8 – Methodology and results

It maybe due to the presentation style, but I could not follow the result section presenting the crop coefficients very well. It has to be structured and presented in a more clear manner: objective and reasoning for location, crop types, etc, data used, hypothesis to prove, which products in which form are tested, and what do results show.

Authors Response

This section has been re-written to try and present this concept in a more structured and clear manner in the manuscript. (P8,L11)

Authors Changes in Manuscript

"For irrigated areas, the crop coefficient (kc) was used. The crop coefficient is a property of a plant that aids in determining ET and can be calculated using equation 3.

kc = ET/PET

Where kc is the coefficient for crops growing under conditions of optimum fertility and soil moisture and achieving full production potential (Allen et al. 1998). In reality optimal conditions are rarely met, however this measure was used to evaluate how well the ET products determined ET across irrigated areas. Average crop coefficients for maize, wheat and sugarcane estimated by FAO were used as a reference. The long-term annual average mean ET estimates across irrigated areas were divided by the long-term annual average mean PET estimates across irrigated areas to find the average crop coefficient (kc) across irrigated areas. These estimates were found by looking at the mean of the area according to the AEIai shapefile. The bias between the reference kc from FAO and estimated kc using individual ET product estimates and PET derived using the mean of the three PET products was recorded."

Reviewer 2, Specific Comment – Stylistic and structural

I highly recommend to look into the papers of Zeng et al., 2012 (ERL) and Miralles et al., 2016 (HESS) as an example of a good presentation style, and especially of the methodological part, as well as their choice of graphs.

Authors Response

The mentioned papers and others as well as internal discussion was conducted and the paper revised along with the figures and graphs. There is a substantial change in the

entire manuscript as well as most figures and graphs to be more clear and structured.

Authors Changes in Manuscript

Changes are found throughout the entire manuscript.

Reviewer 2, Specific Comment 1 – Stylistic and structural

Usage of term throughout the paper changes and confuses the reader. For ex, the term trend is first used as trend itself, but also to indicate tendencies if I understood it correctly (p13, L5 or p16,L9).

Authors Response

The term trend has been looked into and changed according to trend and tendency. As have other terms which we found to be confusing or arbitrary. (P12,L2), (P19,L19)

Authors Changes in Manuscript

"The calculated ET for most of the ET products and also for the majority of basins falls under the curve showing a tendency for products to underestimate basin ET." "In terms of consistency in results with previous studies conducted on some of the products under evaluation we see similar tendencies."

Reviewer 2, Specific Comment 2 – Stylistic and structural

Typos are also present throughout the paper (e.g. p3 L2, p7 L17, p15 L7, p20 L3). A proper internal resew is required.

Authors Response

Any typos we have found have been corrected in the manuscript. A spell check and internal review has been conducted again with the new version of this manuscript.

Reviewer 2, Specific Comment 3 – Stylistic and structural

Discussion section rather reads like methods and should be incorporated to methods.

Instead, the discussion section should place the paper findings into the existing knowledge as was already mentioned earlier.

Authors Response

We agree that the discussion should place the paper findings within existing knowledge and have added this to the manuscript. However, we also feel that the discussion section discusses certain assumptions and findings that requires further explanations. We found this style quite useful and interesting to read and did not incorporate these findings into the methods section. (P19,L19)

Authors Changes in Manuscript

"In terms of consistency in results with previous studies conducted on some of the products under evaluation we see similar tendencies. According to (Miralles et al. 2016) GLEAM, MOD16 and other products in their study show divergences in conditions of water stress and drought. Considering large parts of Africa are potentially under water stress due to the semi-arid and arid climate (IPCC 2019; World Bank 2018), this can explain the low ranking of GLEAM and MOD16 in this study. The RMSE and biases found in our study for Africa are comparable with those found by (Vinukollu, Wood, et al. 2011) at the global scale, however comparing different products to that of this study. The range is higher in this study for Africa than the range found at the global scale. In their study, (Trambauer et al. 2014) found GLEAM to underestimate ET in terms of their multi-product mean. This is again consistent with our finding where biases in GLEAM showed large underestimations across the basins in Africa with respected to the calculated ETWB. We used the LandFlux-EVAL benchmark product as an ensemble product without calculating the multi-product mean of the products being used in this study, as it was developed using a large range of ET products. LandFlux-EVAL, with the coarsest spatial resolution, ranked fourth in the final ranking only outranked by the products with the three highest spatial resolutions in this study, CMRSET, SSEBop and WaPOR. Therefore, LandFlux-EVAL performs well overall regardless of it's coarse

resolution and is interesting due to being an ensemble product. Therefore, continuation or commencement of a similar initiative to develop a benchmark product using a range of ET data sets including high resolution products ranked within this study may improve the ensemble product for future use."

Reviewer 2, Specific Comment 4 – Stylistic and structural

Nothing is mentioned about the Budyko result in Discussion or Conclusions.

Authors Response

We have now included conclusions regarding the Budyko analyses in the conclusion section. (P20,L7)

Authors Changes in Manuscript

"According to the comparison of the ETWB with ETBudyko, we see that ETWB follows the Budyko curve and has an overall low bias across the basins. This indicates the calculated ETWB is a sound reference condition to use for analyses."

Reviewer 2, Specific Comment 5 – Stylistic and structural

Nothing is mentioned about the differences between using three precipitation products.

Authors Response

We now use the average of the three precipitation products and not evaluations based on the individual products and thus do not mention this.

Reviewer 2, Specific Comment 6 – Stylistic and structural

Many abbreviations are not opened. Add abbreviation table to the paper. Use less abbreviations, i.e. if possible leave it open. Very hard to follow.

Authors Response

We have now opened all abbreviations but have not added an abbreviation table to the

paper as all abbreviations are opened.

Reviewer 2, Specific Comment 7 – Stylistic and structural

Explanation of the results in Figures are often not complete. Not clear which products were used, which reference, etc. Figures are too small.

Authors Response

Figures have mostly all been changed being high resolution and easier to read. Explanation of results of these figures tries to be more complete.

Authors Changes in Manuscript

Changes throughout the entirety of the manuscript

Reviewer 2, Specific Comment 8 – Stylistic and structural

Lots of sentences are too wake, "Based on the elements being analysed. . . p21, L30" What is meant? Be more concrete.

Authors Response

We have tried to be stronger in our sentences. Instead of using 'based on the elements being analysed' we have used 'based on the selected land cover elements being analysed' for example.

Authors Changes in Manuscript

Changes throughout the entirety of the manuscript

Reviewer 2, Specific Comment 9 – Stylistic and structural

Titles of the sections should be reconsidered.

Authors Response

Titles of the sections have been reconsidered and changed. Previous headers and
figure captions:

1. Introduction

2. Data

2.1. Remotely Sensed ET products

2.1.1. GLEAM

2.1.2. WaPOR

2.1.3. MOD16

2.1.4. SSEBop

2.1.5. WECANN

2.1.6. FLUXNET-MTE

2.1.7. ETMonitor

2.1.8. CMRSET

2.1.9. Multi-Product Mean

2.2. Precipitation data

2.2.1. EWEMBI

2.2.2. CHIRPS

2.2.3. MSWEP

2.3. Discharge data

2.4. Reference potential evapotranspiration data

3. Methodology

3.1. Preprocessing and data analsysis

2.2.1. Catchment water balance evapotranspiration (ETWB)

2.2.2. Evaluation using the Budyko curve

2.2.3. Spatial variability assessment

2.2.4. Assessment of similarity

3. Results

3.1. Catchment water balance

3.1.1. Comparison of precipitaton and potential evapotranspiration products

3.1.2. Basins used in analyses

3.1.3. Catchment water balance comparison

3.2. Evaluation using the Budyko curve

3.3. Spatial variability assessment

3.4. Product similarity assessment

3.5. Ranking of products

4. Discussion

5. Conclusion

. . ."

[Figure]

[Figure]

**Fig. 1.** Figure 5: Comparison of EWEMBI, MSWEP, CHIRPS precipitation products on their prediction of mean P across the basins

**Table 1.** Characteristics of remotely sensed ET products

| Product | Temporal Coverage | Spatial Coverage | Temporal Resolution | Spatial Resolution | Estimation Approach | Input Data Source | Reference |
|---|---|---|---|---|---|---|---|
| CMRSET (v20140423) | 2000-2013 | Global | 8-daily | 0.0022° × 0.0022° | P-T Equation, relationship between EVI and GVMI | MODIS | (Guerschman et al., 2009) |
| Access: http://remote-sensing.nci.org.au/u39/public/html/wirada/index.shtml | | | | | | | |
| ETMonitor | 2008-2013 | Global | daily | 0.005° × 0.005° | P-M, Gash model, Shuttleworth-Wallace | MODIS | Zheng et al. (2016) |
| Access: email first author in reference | | | | | | | |
| GLEAM (v3.2a) | 1980-2016 | Global | Daily | 0.25° × 0.25° | P-T Equation, soil stress factor | AMSR-E, LPRM, TRMM | Martens et al. (2017); Miralles et al. (2011) |
| Access: www.gleam.eu | | | | | | | |
| LandFlux-EVAL | 1989-2005 | Global | Monthly | 1° × 1° | Ensemble Approach | See reference | Mueller et al. (2013b) |
| Access: https://iac.ethz.ch/group/land-climate-dynamics/research/landflux-eval.html | | | | | | | |
| MOD16 (vA3) | 2000-2014 | Global | Monthly | 0.0083° × 0.0083° | P-M Equation, surface conductance model | MODIS | Mu et al. (2011, 2007) |
| Access: https://modis.gsfc.nasa.gov/data/dataprod/mod16.php | | | | | | | |
| MTE (vMay12) | 1982-2012 | Global | Monthly | 0.5° × 0.5° | MTE approach, training using in-situ observations, flux tower data | Eddy Covariance, in-situ | Jung et al. (2011) |
| Access: https://climatedataguide.ucar.edu/climate-data/fluxnet-mte-multi-tree-ensemble | | | | | | | |
| SSEBop (v4) | 2003-2017 | Global | Monthly | 0.0096° × 0.0096° | P-M Equation, ET fractions from $T_s$ estimates | MODIS | Senay et al. (2013) |
| Access: https://earlywarning.usgs.gov/fews/search | | | | | | | |
| WaPOR (v1.1) | 2009-2017 | Africa | Dekadal | 0.0022° × 0.0022° | P-M Equation, calculates E, T and I separately | MODIS, GEOS-5/MERRA | FAO (2018) |
| Access: https://wapor.apps.fao.org/home/1 | | | | | | | |
| WECANN (v1.0) | 2007-2015 | Global | Monthly | 1° × 1° | ANN approach, training using observations and model based LE | GOME-2 | Alemohammad et al. (2017) |
| Access: https://avdc.gsfc.nasa.gov/pub/data/project/WECANN/ | | | | | | | |

**Fig. 2.** Table 1: Characteristics of Remotely sensed ET products

**Table 2.** Characteristics of precipitation products

| Product | Temporal Coverage | Spatial Coverage | Temporal Resolution | Spatial Resolution | Input Data Source | Reference |
|---|---|---|---|---|---|---|
| EWEMBI (v1.1) | 1979-2016 | Global | daily | 0.5° × 0.5° | ERA-Interim, WFDEI: (Weedon et al., 2014), E2OBS: (Calton et al., 2016) | (Lange, 2019) |
| | Access: http://doi.org/10.5880/pik.2019.004 | | | | | |
| CHIRPS (v2.0) | 1981-2019 | quasi-Global | daily | 0.05° × 0.05° | in situ precipitation gauges, TRMM: (Huffman et al., 2007), CMORPH: (NCAR, 2017) | Funk et al. (2015) |
| | Access: https://chc.ucsb.edu/data/chirps | | | | | |
| MSWEP (v2.2) | 1979-2017 | Global | 3-hourly | 0.1° × 0.1° | in situ precipitation gauges, CMORPH, TRMM, GSMaP: (JAXA, 2009), IRA-Interim | Bai and Liu (2018) |
| | Access: http://www.gloh2o.org/ | | | | | |

**Fig. 3.** Table 2: Characteristics of precipitation products

**Table 3.** Characteristics of discharge data

| Product | Temporal Coverage | Spatial Coverage | Temporal Resolution | Spatial Resolution | Input Data Source | Reference |
|---|---|---|---|---|---|---|
| GRDC (v1.1) | 1806-2019 | Global | daily | point data | in situ discharge gauges | |
| | Access: https://www.bafg.de/GRDC/EN/Home/ | | | | | |
| HYDR-VUB | 1932-2018 | Global | daily | point data | in situ discharge gauges | |
| | Access: on request to http://www.hydr.vub.ac.be/ | | | | | |

**Fig. 4.** Table 3: Characteristics of discharge data

**Table 4.** Characteristics of potential evapotranspiration products

| Product | Temporal Coverage | Spatial Coverage | Temporal Resolution | Spatial Resolution | Input Data Source | | Reference | |
|---|---|---|---|---|---|---|---|---|
| Hargreaves | 1979-2012 | Global | daily | 0.05° × 0.05° | WFDEI, DEM | SRTM | (Sperna et al., 2015) | Weiland |
| | Access: https://wci.earth2observe.eu/ | | | | | | | |
| Penman-Montieth | 1979-2012 | Global | daily | 0.05° × 0.05° | WFDEI, DEM | SRTM | (Sperna et al., 2015) | Weiland |
| | Access: https://wci.earth2observe.eu/ | | | | | | | |
| Priestly-Taylor | 1979-2012 | Global | daily | 0.05° × 0.05° | WFDEI, DEM | SRTM | (Sperna et al., 2015) | Weiland |
| | Access: https://wci.earth2observe.eu/ | | | | | | | |

**Fig. 5.** Table 4: Characteristics of portential evapotranspiration products

| Basin | Variable | Data Availability | Trend | hypothesis | p-value | z-value | no. of samples |
|---|---|---|---|---|---|---|---|
| Awash | ET | 1990-2004 | no trend | false | 0.2496 | -1.1514 | 14 |
| | P | 1979-2016 | | | | | 38 |
| | Q | 1990-2004 | MK test not conducted, no trend found in ET | | | | 15 |
| Bandama | ET | 1979-1996 | no trend | false | 0.7619 | -0.3030 | 18 |
| | P | 1979-2016 | | | | | 38 |
| | Q | 1970-1996 | MK test not conducted, no trend found in ET | | | | 27 |
| Blue Nile | ET | not enough ET data points to conduct MK test on calculated ET | | | | | 4 |
| | P | 1979-2016 | no trend | false | 0.6875 | -0.4023 | 38 |
| | Q | 1900-1982 | decreasing | true | 0.0009 | -3.3271 | 83 |
| Buzi | ET | not enough ET data points to conduct MK test on calculated ET | | | | | 5 |
| | P | 1979-2016 | no trend | false | 0.4210 | -0.8046 | 38 |
| | Q | 1957-1983 | no trend | false | 1.0 | 0.0 | 23 |
| Cavally | ET | 1979-1996 | no trend | false | 0.54449 | -0.6060 | 18 |
| | P | 1979-2016 | | | | | 38 |
| | Q | 1970-1996 | MK test not conducted, no trend found in ET | | | | 27 |
| Congo | ET | 1979-2010 | no trend | false | 0.0830 | -1.7336 | 31 |
| | P | 1979-2016 | | | | | 38 |
| | Q | 1903-2010 | MK test not conducted, no trend found in ET | | | | 108 |
| Cunene | ET | 1980-2015 | increasing | true | 0.0003 | 3.5823 | 36 |
| | P | 1979-2016 | | | | | 38 |
| | Q | 1980-2015 | MK test not conducted, no trend found in ET | | | | 36 |
| Gambia | ET | not enough ET data points to conduct MK test on calculated ET | | | | | 5 |
| | P | 1979-2016 | no trend | false | 0.2579 | 1.1315 | 38 |
| | Q | 1979, 1981-82, 1984,1988 | no trend | false | 0.8065 | 0.2449 | 5 |
| Groot | ET | 1979-2015 | no trend | false | 0.1697 | 1.3733 | 37 |
| | P | 1979-2016 | | | | | 38 |
| | Q | 1964-2014 | MK test not conducted, no trend found in ET | | | | 51 |
| Kamoe | ET | 1979-1996 | no trend | false | 0.3633 | -0.9091 | 18 |
| | P | 1979-2016 | | | | | 38 |
| | Q | 1970-1996 | MK test not conducted, no trend found in ET | | | | 27 |
| Lake Chad | ET | not enough ET data points to conduct MK test on calculated ET | | | | | 4 |
| | P | 1979-2016 | increasing | true | 0.0194 | 2.3384 | 38 |
| | Q | 1983-1986 | no trend | false | 0.3081 | -1.0190 | 4 |
| Maputo | ET | not enough ET data points to conduct MK test on calculated ET | | | | | 5 |
| | P | 1979-2016 | no trend | false | 0.3393 | -0.9555 | 38 |
| | Q | 1953-1983 | no trend | false | 0.1261 | -1.5297 | 31 |
| Mono | ET | 1979-2007 | no trend | false | 0.5115 | -0.6565 | 29 |
| | P | 1979-2016 | | | | | 38 |
| | Q | 1944-2007 | MK test not conducted, no trend found in ET | | | | 64 |
| Niger | ET | 1979-2006 | no trend | false | 0.6214 | 0.4939 | 28 |
| | P | 1979-2016 | | | | | 38 |
| | Q | 1970-2006 | MK test not conducted, no trend found in ET | | | | 37 |
| Nile | ET | not enough ET data points to conduct MK test on calculated ET | | | | | 6 |
| | P | 1979-2016 | no trend | false | 0.2909 | 1.0560 | 38 |
| | Q | 1912-1984 | no trend | false | 0.0693 | 1.8164 | 56 |
| Okavango | ET | 1979-2014 | increasing | true | 0.0127 | 2.4926 | 36 |
| | P | 1979-2016 | | | | | 38 |
| | Q | 1950-2014 | MK test not conducted, no trend found in ET | | | | 65 |
| Olifant | ET | 1979-2014 | no trend | false | 0.9457 | 0.0681 | 36 |
| | P | 1979-2016 | | | | | 38 |
| | | | MK test not conducted, no trend found in ET | | | | |
| | Q | 1927-2014 | | | | | 88 |
| Orange | ET | 1979-2016 | no trend | false | 0.6691 | 0.4274 | 38 |
| | P | 1979-2016 | | | | | 38 |
| | Q | 1936-2014 | MK test not conducted, no trend found in ET | | | | 79 |
| Queme | ET | 1979-80, 1982-84, 1990-2005, 2007 | no trend | false | 0.3377 | 0.9587 | 22 |
| | P | 1979-2016 | | | | | 38 |
| | Q | 1948-2007 | MK test not conducted, no trend found in ET | | | | 60 |
| Rufiji | ET | not enough ET data points to conduct MK test on calculated ET | | | | | 0 |
| | P | 1979-2016 | no trend | False | 0.6508 | -0.4526 | 38 |
| | Q | 1954-1978 | no trend | False | 0.9741 | -0.0324 | 20 |
| Sassandra | ET | 1979-1996 | no trend | false | 0.8796 | 0.1515 | 18 |
| | P | 1979-2016 | | | | | 38 |
| | Q | 1970-1996 | MK test not conducted, no trend found in ET | | | | 27 |
| Save | ET | not enough ET data points to conduct MK test on calculated ET | | | | | 3 |
| | P | 1979-2016 | no trend | False | 0.8801 | 0.1509 | 38 |
| | Q | 1968-1981 | increasing | True | 0.0118 | 2.5183 | 14 |
| Senegal | ET | 1979-1989 | no trend | false | 0.2129 | 1.2456 | 11 |
| | P | 1979-2016 | | | | | 38 |
| | Q | 1979-1989 | MK test not conducted, no trend found in ET | | | | 11 |
| Tana | ET | not enough ET data points to conduct MK test on calculated ET | | | | | 0 |
| | P | 1979-2016 | decreasing | True | 0.0006 | -3.4447 | 38 |
| | Q | 1975-1978 | no trend | False | 0.7341 | -0.3397 | 4 |
| Upper Blue Nile | ET | not enough ET data points to conduct MK test on calculated ET | | | | | 8 |
| | P | 1979-2016 | no trend | False | 0.6875 | -0.4023 | 38 |
| | Q | 1961-1983 | no trend | False | 0.1339 | -1.4988 | 26 |
| Void | ET | not enough ET data points to conduct MK test on calculated ET | | | | | 3 |
| | P | 1979-2016 | no trend | False | 0.1251 | -1.5338 | 38 |
| | Q | 1979-1981 | increasing | True | 0.0483 | 1.9748 | 7 |
| Zambezi | ET | 1979-1990 | no trend | false | 0.5371 | 0.6172 | 12 |
| | P | 1979-2016 | | | | | 38 |
| | Q | 1960-1990 | MK test not conducted, no trend found in ET | | | | 31 |

**Fig. 6.** Table A1: Mann-Kendall test results for all basins on evapotranspiration, precipitation and discharge

**Table 5.** Calculated statistics, bias, $bias_{aw}$, RMSE, $RMSE_{aw}$ and r, for the comparison of the long-term annual average $ET_{WB}$ versus $ET_{RS}$

|  | CMRSET | ETMonitor | GLEAM | LandFlux-EVAL | MOD16 | MTE | SSEBop | WaPOR | WECANN |
|---|---|---|---|---|---|---|---|---|---|
| bias | -19 | 156 | 254 | 115 | 131 | 146 | 12 | -3 | 139 |
| $bias_{aw}$ | -18 | 237 | 313 | 148 | 266 | 183 | 30 | -46 | 223 |
| RMSE | 113 | 211 | 273 | 152 | 199 | 184 | 163 | 104 | 189 |
| $RMSE_{aw}$ | 187 | 502 | 594 | 304 | 590 | 424 | 123 | 165 | 520 |
| r | 0.94 | 0.91 | 0.97 | 0.97 | 0.91 | 0.95 | 0.89 | 0.96 | 0.95 |

**Fig. 7.** Table 5: Calculated statistics bias, bias_aw, RMSE, RMSE_aw and r for the comparison of the lon-term annual average ET_WB versus ET_RS

**Fig. 8.** Figure 10: Spatial assessment across Africa of each ET product based on selected land cover elements, forest, irrigated areas and water bodies

---

## Author Response (AR1)

**Can we trust remote sensing ET products across Africa?**

Imeshi WEERASINGHE[a,*], Ann van GRIENSVEN[ab], Wim BASTIAANSSEN[b], Marloes MUL[b], Li JIA[c]

celray.chawanda@vub.be

[a]Vrije Universiteit Brussel (VUB)
[b]IHE Delft Institute for Water Education
[c]Chinese Academy of Sciences
[d]Joint Center for Global Change Studies

**Overview**

We would like to thank the editor for his dedication in reviewing the manuscript and responses to the reviewers. We are also thankful for his thoughtful and constructive suggestions and comments. We have addressed all the comments raised by the editor and the manuscript has improved from the proposed changes.

**Contents**

**Editor Comments**

**Editor, Comment 1**

As per reviewer #2 one of the main weakness of the original version of the manuscript is lack of details and clarity on methods, results etc. I would strongly encourage you to have the revised manuscript reviewed by a colleague outside of the authors list to make sure that they find methods etc easily comprehensible.

**Authors Response**

1. The manuscript was read by two separate reviewers outside of the co-author list. Both gave useful and helpful comments on the manuscript to help with the changes.
2. Reviewer 1: Had mostly small comments regarding the overall manuscript with small suggestions on how to improve the manuscript which were mostly taken on board. In general, the comment was made that it was an "Very interesting and fun to read"
3. Reviewer 2: Had some suggestions on improvement of the methodology which were incorporated in the methodology section. He also had three main points which were 1. Contextualising the paper which was added to the discussion section, 2. The ranking system which we did not change and 3. The crop coefficient method which he did not agree with so we took it out of the paper as we also agreed. We also identified the shortcomings in the discussion section. He found the discussion and conclusion sections of the paper well written and gave a clear understanding of the paper.

**Authors Changes in Manuscript**

Changes were all throughout the manuscript due to the entire restructuring of the original manuscript.

**Editor, Comment 2**

I think reviewer #2's comment on the downscaling of coarse resolution data is valid and should be discussed in the manuscript. Interpolation of coarse resolution data can introduced further uncertainties and some of the differences can occur simply due to statistical interpolation.

Authors Response

This point was discussed in detailed as a response to author two. Subsequent to the editor's request, this was also discussed with Wim Thierry, a global climate modeller who understands this issue well. In terms of cropping these products using the catchment boundary shapefiles, he told us that it makes sense that we resample to the highest resolution in order to get the correct (or as correct as possible) boundaries. We also implemented some discussion about the reasons for this downscaling in the manuscript P7L13-16. As mentioned to reviewer #2 we calculated $ET_{WB}$ with the original resolution and resampled resolution and found negligible differences.

Authors Changes in Manuscript

[revised manuscript text omitted]
 0.0003 3.5823 36 5 P 1979-2016 no trend false 0.2579 1.1315 38 Q 19791981-8219841988 no trend false 0.8065 0.2449 5 Groot1979-2015 no trend false 0.1697 1.3733 37 Kamoe~~$_{WB}$ versus $ET_{RS}$

**Table 7.** Bias between the $ET_{Budyko}$ and $ET_{EB}$

| mm/year | $ET_{WB}$ | CMRSET | ETMonitor | GLEAM | LandFlux-EVAL | MOD16 | MTE | SSEBop | WaPOR | WECANN |
|---------|-----------|--------|-----------|-------|---------------|-------|-----|--------|-------|--------|
| bias    | 42        | 101    | 202       | 284   | 152           | 185   | 177 | 140    | 86    | 180    |

**Table 8.** Ranking of the RS products based on the different evaluation steps of the proposed methodology

| | CMRSET | ETMonitor | GLEAM | LandFlux-EVAL | MOD16 | MTE | SS... |
|---|---|---|---|---|---|---|---|
|  height    **Catchment water balance ranking (CWB)** | | | | | | | |
| bias | 3 | 8 | 9 | 4 | 5 | 7 | 4... |
| bias$_{aw}$ | 1 | 7 | 9 | 4 | 8 | 5 | |
|  RMSE | 2 | 8 | 9 | 3 | 7 | 5 | |
| RMSE$_{aw}$ | 3 | 6 | 9 | 4 | 8 | 5 | |
|  r | 6 | 7 | 1 | 1 | 7 | 4 | |
| **Overall CWB ranking** | 2 | 8 | 9 | 3 | 7 | 5 | |
| **Budyko ranking** | | | | | | | |
| Budyko | 2 | 8 | 9 | 4 | 7 | 5 | |
| **Spatial variability ranking** | | | | | | | |
|     **Visual Inspection (VI) - land cover elements** | | | | | | | |
| Forest | 2 | 2 | 6 | 8 | 1 | 7 | |
| Irrigated Area | 1 | 4 | 8 | 9 | 5 | 6 | |
| Water Bodies | 1 | 1 | n/a | n/a | n/a | n/a | |
| **Overall VI spatial ranking** | 1 | 2 | 7 | 9 | 5 | 6 | |
| **Quantitative Inspection (QI) - land cover elements** | | | | | | | |
|  Forest | 1 | 6 | 7 | 4 | 8 | 5 | 3... |
| Water Bodies | 2 | 1 | n/a | n/a | n/a | n/a | |
|  **Overall QI spatial ranking** | 1 | 4 | 9 | 5 | 7 | 6 | |
| **Overall spatial ranking** | 1 | 3 | 8 | 7 | 5 | 6 | |
|     **Final ranking** | | | | | | | |
| With visual inspection | 1 | 5 | 9 | 4 | 7 | 6 | |
|  Without visual inspection | 2 | 7 | 9 | 4 | 8 | 5 | 4... |

**Table 9.** Differences in mean WB ET estimations for varying RS product periods

| | Total | CMRSET | ETMonitor | GLEAM | Period
MOD16 | MTE | SSEBop | WaPOR | WECANN | Average |
|---|---|---|---|---|---|---|---|---|---|---|
| **Congo** | | | | | | | | | | |
| | 1979-2010 | 2000-2010 | 2008-2010 | 1980-2010 | 2000-2010 | 1982-2010 | 2003-2010 | 2009-2010 | 2007-2010 | |
| ET mm/year | 1186 | 1203 | 1159 | 1196 | 1203 | 1194 | 1194 | 1168 | 1193 | 1189 |
| Bias mm/year | | 17 | 27 | 10 | 17 | 8 | 8 | 18 | 7 | 14 |
| % bias | | 1.4 | 2.3 | 0.8 | 1.4 | 0.7 | 0.7 | 1.5 | 0.6 | 1.2 |
| **Groot** | | | | | | | | | | |
| | 1979-2015 | 2000-2013 | 2008-2013 | 1980-2015 | 2000-2014 | 1982-2012 | 2003-2015 | 2009-2015 | 2007-2015 | |
| ET mm/year | 373 | 390 | 381 | 377 | 390 | 371 | 387 | 396 | 392 | 386 |
| Bias mm/year | | 17 | 8 | 4 | 17 | 2 | 14 | 23 | 19 | 13 |
| % bias | | 4.4 | 2.1 | 1.1 | 4.4 | 0.5 | 3.6 | 5.8 | 4.9 | 3.4 |
| **Olifant** | | | | | | | | | | |
| | 1979-2014 | 2000-2013 | 2008-2013 | 1980-2014 | 2000-2014 | 1982-2012 | 2003-2014 | 2009-2014 | 2007-2014 | |
| ET mm/year | 278 | 279 | 293 | 284 | 278 | 286 | 272 | 275 | 296 | 283 |
| Bias mm/year | | 1 | 15 | 6 | 0 | 8 | 6 | 3 | 18 | 7 |
| % bias | | 0.4 | 5.1 | 2.1 | 0.0 | 2.8 | 2.2 | 1.1 | 6.1 | 2.5 |
| **Orange** | | | | | | | | | | |
| | 1979-2015 | 2000-2013 | 2008-2013 | 1980-2015 | 2000-2014 | 1982-2012 | 2003-2015 | 2009-2015 | 2007-2015 | |
| ET mm/year | 349 | 377 | 376 | 356 | 374 | 362 | 351 | 350 | 351 | 362 |
| Bias mm/year | | 28 | 27 | 7 | 25 | 13 | 2 | 1 | 2 | 13 |
| % bias | | 7.4 | 7.2 | 2.0 | 6.7 | 3.6 | 0.6 | 0.3 | 0.6 | 3.6 |

---

## Author Response (AR2)

**Can we trust remote sensing ET products across Africa?**

Imeshi WEERASINGHE[a,*], Ann van GRIENSVEN[ab], Wim BASTIAANSSEN[b], Marloes MUL[b], Li JIA[c]

celray.chawanda@vub.be

[a]Vrije Universiteit Brussel (VUB)
[b]IHE Delft Institute for Water Education
[c]Chinese Academy of Sciences
[d]Joint Center for Global Change Studies

**Overview**

We would like to thank the editor for his dedication in reviewing the manuscript and responses to the reviewers. We are also thankful for his thoughtful and constructive suggestions and comments. We have addressed all the comments raised by the editor and the manuscript has improved from the proposed changes.

**Contents**

**Editor Comments**

**Editor, Comment 1**

Title of table 1 should be corrected. You are listing ETRS products here not precipitation.

Authors Response

1. The title of table 1 was corrected to say 'evapotranspiration' instead of 'precipitation'.

Authors Changes in Manuscript

"Characteristics of evapotranspiration products"

**Editor, Comment 2**

Page 10 Line 19: Consider using PET for "potential evapotranspiration".

Authors Response

"potential evapotranspiration" was changed to "PET"

Authors Changes in Manuscript

"Precipitation and PET were taken as the average of three products."

**Editor, Comment 3**

Figure 9: The yellow color used for SSEBop is hardly visible. Please consider using a different color.

**Authors Response**

SSEBop colour in graph was changed to black

Authors Changes in Manuscript

[Figure]

**Editor, Comment 4**

Figure 10: Please consider mentioning in the caption what each of the rectangles indicate. Also the color bars should be revised and enlarged. I assume that each of the colors are indicating a range of ET values not a discreet ET value?.

**Authors Response**

We have included the description of the rectangles/boxes in the description. Yes each of the colours indicated a range of ET values and not a discreet value which has been amended in the figure.

Authors Changes in Manuscript

[Figure]

Editor, Comment 5

Figure 11: Please provide the source of forest ET found from literature.

Authors Response

This was already included in the text. Do you need more references here?

Authors Changes in Manuscript

"For forested areas, the average ET was taken from literature where estimations for the Congo forest, the forested area being evaluated, were between 1200-1500 mm/year (Otto2013,Reynolds1988)."

**Editor, Comment 6**

Table 2: Please also consider mentioning the latency of each of these datasets. This information would be helpful for readers and potential users of these datasets.

Authors Response

Unfortunately this information is not available for all datasets as I could find so I did not include this in the manuscript.

Authors Changes in Manuscript
No changes

**Editor, Comment 6**

Table 7: I think the title should be "Bias between the ETBudyko and, ETWB and ETRS".

Authors Response

The title of table 7 was changes to "Bias between the $ET_{Budyko}$ and, $ET_{WB}$ and $WT_{RS}$

Authors Changes in Manuscript
"Bias between the $ET_{Budyko}$ and, $ET_{WB}$ and $WT_{RS}$